# Coreset for Robust Geometric Median: Eliminating Size Dependency on Outliers

**Ziyi Fang**
Nanjing University

**Lingxiao Huang**[*]
Nanjing University

**Runkai Yang**
Nanjing University

## Abstract

We study the robust geometric median problem in Euclidean space $\mathbb{R}^d$, with a focus on coreset construction. A coreset is a compact summary of a dataset $P$ of size $n$ that approximates the robust cost for all centers $c$ within a multiplicative error $\varepsilon$. Given an outlier count $m$, we construct a coreset of size $\tilde{O}(\varepsilon^{-2} \cdot \min\{\varepsilon^{-2}, d\})$ when $n \geq 4m$, eliminating the $O(m)$ dependency present in prior work [39, 40]. For the special case of $d = 1$, we achieve an optimal coreset size of $\tilde{\Theta}(\varepsilon^{-1/2} + \frac{m}{n}\varepsilon^{-1})$, revealing a clear separation from the vanilla case studied in [40, 1]. Our results further extend to robust $(k, z)$-clustering in various metric spaces, eliminating the $m$-dependence under mild data assumptions. The key technical contribution is a novel non-component-wise error analysis, enabling substantial reduction of outlier influence, unlike prior methods that retain them. Empirically, our algorithms consistently outperform existing baselines in terms of size-accuracy tradeoffs and runtime, even when data assumptions are violated across a wide range of datasets.

## 1  Introduction

Geometric median, also known as the Fermat-Weber problem, is a foundational problem in computational geometry, whose objective is to identify a center $c \in \mathbb{R}^d$ for a given dataset $P \subset \mathbb{R}^d$ of size $n$ that minimizes the sum of Euclidean distances from each data point to $c$. The given objective function, while simple and user-friendly, suffers from significant robustness problems when exposed to noisy or adversarial data [13, 15, 14, 32, 24]. For example, an adversary could add a few distant noisy outliers to the main cluster. These points could deceive the geometric median algorithm into incorrectly positioning the center closer to these outliers to minimize the cost function. Such susceptibility to outliers has been a considerable obstacle in data science and machine learning, prompting a substantial amount of algorithmic research on the subject [15, 32, 29, 53, 49].

**Robust geometric median.** We consider *robust* versions of the geometric median problem, specifically a widely-used variant that introduces outliers [13]. Formally, given an integer $m \geq 0$, the goal of robust geometric median is to find a center $c \in \mathbb{R}^d$ that minimizes the objective function:

$$\text{cost}^{(m)}(P, c) := \min_{L \subset P : |L| = m} \sum_{p \in P \setminus L} \text{dist}(p, c), \tag{1}$$

where $L$ represents the set of $m$ outliers w.r.t. $c$ and $\text{dist}(p, c) = \|p - c\|_2$ denotes the Euclidean distance from $p$ to center $c$. Intuitively, outliers capture the points that are furthest away and these are typically considered to be noise. When the number of outliers $m = 0$, the robust geometric median problem simplifies to the vanilla geometric median. This problem (together with its generalization: robust $(k, z)$-clustering) has been well studied in the literature [15, 7, 29, 2, 23, 53]. However, the presence of outliers introduces significant computational challenges, particularly in the context of large-scale datasets [52, 50, 51, 53]. For example, the approximation algorithm for robust geometric

---

[*] Alphabetical order. Correspondence to: `huanglingxiao@nju.edu.cn`. LH's affiliations are the State Key Laboratory of Novel Software Technology and the New Cornerstone Science Laboratory.

39th Conference on Neural Information Processing Systems (NeurIPS 2025).

median proposed by [52] requires $\tilde{O}(n^{d+1}(n-m)d)^2$; and [2] presents a fixed-parameter tractable (FPT) algorithm with $f(\varepsilon, m) \cdot n^{O(1)}$ time. These challenges have driven extensive research on data reduction algorithms designed for limited hardware and time constraints.

**Coreset.** To tackle the computational challenge, we study *coresets*, which are (weighted) subsets $S \subseteq P$ such that the clustering cost $\text{cost}^{(m)}(S, c)$ approximates $\text{cost}^{(m)}(P, c)$ within a factor of $(1 \pm \varepsilon)$ for all center sets $c \in \mathbb{R}^d$, where $\varepsilon > 0$ is a given error parameter.[3] A coreset preserves key geometric information while substantially reducing the dataset size, thus serving as a compact proxy for the original dataset $P$. Consequently, applying existing approximation algorithms to the coreset significantly improves computational efficiency. Moreover, coresets can be reused in future analyses of $P$, reducing redundant computation and saving storage resources. Finally, the size of the coreset reflects the intrinsic complexity of the problem, making the study of optimal coreset size a research topic of independent interest.

Coreset for vanilla geometric median ($m = 0$) has been extensively studied across different dimensions $d$ [28, 16, 18, 19, 1, 22] (see Appendix A.2 for details). When $d = 1$, [37] proposed a coreset of size $\tilde{\Theta}(\varepsilon^{-1/2})$. Subsequently, [1] extended this result by constructing an optimal coreset of size $\tilde{\Theta}(\varepsilon^{-d/(d+1)})$ for dimension $d = O(1)$. Moreover, for high dimensions $d > \varepsilon^{-2}$, the optimal coreset size is $\tilde{\Theta}(\varepsilon^{-2})$ [18, 21]. In contrast, the study of coreset for robust geometric median is far from optimal, even in the simplest case of $d = 1$. Early work either required an exponentially large coreset size [27] or involved relaxing the outlier constraint [39]. A notable recent advancement by [39] overcame these limitations, proposing a coreset of size $O(m) + \tilde{O}(\varepsilon^{-3} \cdot \min\{\varepsilon^{-2}, d\})$, using a hierarchical sampling framework based on [9]. Further improvements reduced the coreset size to $O(m) + \tilde{O}(\varepsilon^{-2} \cdot \min\{\varepsilon^{-2}, d\})$ [40]. A recent paper [42] presented a coreset size $O(m\varepsilon^{-1} + \text{Vanilla size})$ via a novel reduction from the robust case to the vanilla case.

All previous coreset sizes for robust geometric median contain the factor $m$. However, the outlier number $m$ could approach $\Omega(n)$ in real-world scenarios [25, 12, 30, 10]. For instance, the PageBlocks dataset has $n = 5473$ points with $m \approx 0.1n$ outliers [10]. Such $m$ results in an inefficient coreset size, raising a natural question: *Can we eliminate the $O(m)$ term from the coreset size?*

At first glance, one might assume the answer is negative, as [39] establishes a coreset lower bound of $\Omega(m)$. However, their worst-case instance critically relies on an extremely large number of outliers, specifically with $m = n - 1$. This requirement can be relaxed to $n - m = o(n)$ (see Theorem 1.1). Motivated by this, we investigate the possibility of eliminating the $O(m)$ dependency when $n - m = \Omega(n)$, and show that this condition is both necessary and sufficient. Under this setting, we obtain a coreset of size $\tilde{O}(\varepsilon^{-2} \cdot \min\{\varepsilon^{-2}, d\})$ (see Theorem 1.3).

Nevertheless, this size is substantially larger than that of the vanilla case across all dimensions. For instance, when $d = 1$, our coreset size is $\tilde{O}(\varepsilon^{-2})$, whereas the vanilla case achieves a size of only $\tilde{\Theta}(\varepsilon^{-1/2})$ [37]; thus, our bound is likely not optimal. On the other hand, our lower bound in Theorem 1.1 suggests that the optimal coreset size should indeed depend on $m$. This raises a natural question: *What is the optimal coreset size for robust geometric median, and how does it vary with $m$?* Answering this question sheds light on when the complexity of robust geometric median fundamentally diverges from that of the vanilla case.

## 1.1 Our contributions

In this paper, we study (optimal) coreset sizes for the robust geometric median problem. See Table 3 in the appendix for a summary. We begin with a lower bound result (proof in Section C).

**Theorem 1.1** (Coreset lower bound for robust geometric median). *Let $0 < \varepsilon < 0.5$ and $n > m \geq 1$. There exists a dataset $P \subset \mathbb{R}$ of size $n$ such that any $\varepsilon$-coreset of $P$ for the robust geometric median problem must have size $\Omega(\frac{m}{n-m})$.*

---

[2]Here, $\tilde{O}(n)$ denotes $O(n \cdot \text{polylog}(n))$, hiding logarithmic factors.

[3]Here, the computation of $\text{cost}^{(m)}(S, c)$ excludes points of total weight $m$; see Appendix A.1 for the formal definition.

This theorem indicates that when $n - m = o(n)$, which implies $m = \Theta(n)$, the coreset size is at least $\Omega\left(\frac{m}{n-m}\right) = \frac{m}{o(m)}$, which depends on $m$. This extends the previous result in [39] to general $m$. Thus, $n - m = \Omega(n)$ is a necessary condition for eliminating the $O(m)$ term from the coreset size.

Accordingly, we assume $n \geq 4m$ for the algorithmic results, where the constant 4 ensures the inlier number $n - m$ is sufficiently larger than the outlier number $m$. We first state our result when $d = 1$.

**Theorem 1.2** (Optimal coreset for robust 1D geometric median). *Let $n, m \geq 1$ be integers and $\varepsilon \in (0, 1)$. Assume $n \geq 4m$. There is a linear algorithm that given dataset $P \subset \mathbb{R}$ of size $n$, outputs an $\varepsilon$-coreset for robust 1D geometric median of size $\tilde{O}(\varepsilon^{-\frac{1}{2}} + \frac{m}{n}\varepsilon^{-1})$ in $O(n)$ time. Moreover, there exists a dataset $X \subset \mathbb{R}$ of size $n$ such that any $\varepsilon$-coreset must have size $\Omega(\varepsilon^{-\frac{1}{2}} + \frac{m}{n}\varepsilon^{-1})$.*

This theorem provides the optimal coreset size for the robust 1D geometric median problem. Since $m \leq n$, the coreset size is at most $\tilde{O}(\varepsilon^{-1})$, which is independent of $m$. Thus, we eliminate the $O(m)$ dependency in the coreset size compared to previous bounds [40, 42]. In particular, relative to the bound $O(m) + \tilde{O}(\varepsilon^{-2})$ in [40], our result also improves the $\varepsilon$-dependence from $\varepsilon^{-2}$ to at most $\varepsilon^{-1}$.

The theorem also shows how the coreset size increases as $m$ grows. When $m \leq \sqrt{\varepsilon}n$, our coreset size is dominated by $\tilde{O}(\varepsilon^{-1/2})$ since $\frac{m}{n}\varepsilon^{-1} \leq \varepsilon^{-1/2}$. This size matches the coreset size $\tilde{O}(\varepsilon^{-1/2})$ for vanilla 1D geometric median as given in [40]. As $m$ increases from $\sqrt{\varepsilon}n$ to $\frac{n}{4}$, our coreset size is dominated by the term $\tilde{O}(\frac{m}{n}\varepsilon^{-1})$, which is a linear function of $m$ growing from $\tilde{O}(\varepsilon^{-1/2})$ to $\tilde{O}(\varepsilon^{-1})$. This size suggests that the complexity of robust 1D geometric median is higher than vanilla 1D geometric median in this range of outlier number $m$.

We remark that the size lower bound $\Omega(\varepsilon^{-\frac{1}{2}} + \frac{m}{n}\varepsilon^{-1})$ holds for any dimension $d \geq 1$. In contrast, for $d = O(1)$, the tight coreset size for the vanilla geometric median is $\tilde{\Theta}(\varepsilon^{-d/(d+1)})$ [1]. This implies that when $m \geq \Omega(n \cdot \varepsilon^{d/(d+1)})$, our coreset size lower bound exceeds $\Omega(\varepsilon^{-d/(d+1)})$, resulting in a gap between the coreset sizes for the robust and vanilla versions of geometric median.

**Theorem 1.3** (Coreset for robust geometric median in $\mathbb{R}^d$). *Let $n, m, d \geq 1$ be integers and $\varepsilon \in (0, 1)$. Assume $n \geq 4m$. There exists a randomized algorithm that given a dataset $P \subset \mathbb{R}^d$ of size $n$, outputs an $\varepsilon$-coreset for the robust geometric median problem of size $\tilde{O}(\varepsilon^{-2} \min\{\varepsilon^{-2}, d\})$ in $O(nd)$ time.*

Compared to previous bounds $O(m) + \tilde{O}(\varepsilon^{-2} \min\{\varepsilon^{-2}, d\})$ [40] and $O(m\varepsilon^{-1}) + \tilde{O}(\varepsilon^{-2})$ [42], this theorem eliminates the $O(m)$ term in the coreset size when $n \geq 4m$. This result can be extended to various metric spaces (Section E.2).

Finally, we extend this theorem to handle the robust $(k, z)$-clustering problem (Definition F.1), which encompasses robust $k$-median ($z = 1$) and robust $k$-means ($z = 2$). To capture the additional complexity introduced by $k$, we propose the following geometric assumption.

**Assumption 1.4** (Assumptions for robust $(k, z)$-clustering). Given a dataset $P \subset \mathbb{R}^d$ of $n$ points and an $\varepsilon$-approximate center set $C^\star \subset \mathbb{R}^d$ of $P$ for robust $(k, z)$-clustering. Define $P_I^\star := \arg\min_{P_I \subset P, |P_I| = n - m} \sum_{p \in P_I} \text{dist}(p, C^\star)$ to be the inlier points w.r.t. $C^\star$, where $\text{dist}(p, C^\star) := \min_{c \in C^\star} \text{dist}(p, c)$. Let $\{P_1^\star, \ldots, P_l^\star\} \subset P_I^\star$ denote the $k$ inlier clusters induced by $C^\star$, where each $P_i^\star$ contains points in $P_I^\star$ whose closest center is $c_i^\star$. We assume the followings: 1) $\min_{i \in [k]} |P_i^\star| \geq 4m$; 2) $\max_{p \in P_I^\star} \text{dist}(p, C^\star)^z \leq 4k \sum_{p \in P_I^\star} \text{dist}(p, C^\star)^z / |P_I^\star|$.

The first condition directly generalizes the assumption $n \geq 4m$, which requires that the size of each inlier cluster is more than the outlier number $m$. The second excludes "remote inlier points" to $C^\star$, which could play a similar role as outliers. This condition holds for several real-world datasets (see Table 4), demonstrating its practicality. Now we propose the following result.

**Theorem 1.5** (Coreset for robust $(k, z)$-clustering). *Let $n, k, d \geq 1$ be integers and $\varepsilon \in (0, 1)$. There exists a randomized algorithm that given a dataset $P \subset \mathbb{R}^d$ of size $n$ satisfying Assumption 1.4, outputs an $\varepsilon$-coreset for robust $(k, z)$-clustering of size $\tilde{O}(k^2\varepsilon^{-2z} \min\{\varepsilon^{-2}, d\})$ in $O(nkd)$ time.*

It improves upon the previous result $O(m) + \tilde{O}(k^2\varepsilon^{-2z} \min\{\varepsilon^{-2}, d\})$ in [39, 40] by eliminating the $O(m)$ term. Furthermore, the result can also be extended to various metric spaces (Section F.3).

Empirically, in Section 4, we evaluate the performance of our coreset algorithms on six real-world datasets. We compare the size-error tradeoff against baselines [39, 40], and across all tested sizes, our

algorithms consistently achieve lower empirical error. For instance, for robust geometric median on the **Census1990** dataset, our method produces a coreset of size 1000 with an empirical error of 0.012, while the baseline produces a coreset of size 2300 with an empirical error slightly higher than 0.013 (Figure 2). Moreover, our algorithms provide a speedup compared to baselines for achieving the same level of empirical error (see e.g., Table 2). Additionally, we show that our algorithms remain practically effective, regardless of which data assumption in the theoretical results is violated, further demonstrating the practical utility of our algorithms (Section G).

## 1.2 Technical overview

We outline the technical ideas and novelty for Theorems 1.2 and 1.3. Our approach introduces a novel non-component-wise error analysis for coreset construction, enabling a substantial reduction in the number of outlier points rather than preserving them all. In contrast, previous algorithms divide the dataset $P$ into multiple components, and their approach requires aligning the number of outliers between $P$ and $S$ in every component. This idea, as we will show, inherently introduces an $\Omega(m)$-sized coreset. Furthermore, for the 1D case, our new algorithm offers a more refined partitioning of inlier points, enabling an adaptation of the vanilla coreset construction and leading to the optimal coreset size.

### 1.2.1 Overview for Theorem 1.2 ($d = 1$)

**Revisiting coreset construction for vanilla 1D geometric median.** Recall that [37] first partitions dataset $P = \{p_1, \ldots, p_n\} \subset \mathbb{R}$ into $\tilde{O}(\varepsilon^{-1/2})$ buckets, where each bucket $B_i$ is a consecutive subsequence $\{p_l, p_{l+1} \ldots, p_r\}$ (see Definition 2.1). They select a mean point $\mu(B)$, assign a weight $|B|$ for each bucket $B$, and output their union as a coreset such that $\forall c \in \mathbb{R}$,

$$\sum_{i \in [T]} |\mathrm{cost}(B_i, c) - |B_i| \cdot \mathrm{dist}(\mu(B_i), c)| \leq \varepsilon \cdot \mathrm{cost}(P, c). \tag{2}$$

This follows that only the bucket containing $c$ contributes a non-zero error at most $\varepsilon \cdot \mathrm{cost}(P, c)$ (see Lemma B.1). To handle the additional outliers for the robust case, a natural idea is to extend Inequality (2) in the following manner: for each center $c \in \mathbb{R}$ and each tuple $(m_1, \ldots, m_T) \in \mathbb{Z}_{\geq 0}^T$ of outlier numbers per bucket (with $\sum_{i \in [T]} m_i = m$),

$$\sum_{i \in [T]} |\mathrm{cost}^{(m_i)}(B_i, c) - (|B_i| - m_i) \cdot \mathrm{dist}(\mu(B_i), c)| \leq \varepsilon \cdot \mathrm{cost}^{(m)}(P, c). \tag{3}$$

This bounds the induced error $|\mathrm{cost}^{(m_i)}(B_i, c) - (|B_i| - m_i) \cdot \mathrm{dist}(\mu(B_i), c)|$ of each bucket $B_i$ when aligning the outlier numbers within this bucket for dataset $P$ and coreset $S$.

Prior work [39, 40] utilized this component-wise error analysis for robust coreset construction. They show that at most three buckets $B_i$ could induce a non-zero error, including a bucket that contains $c$ and two "partially intersected" buckets with $0 < m_i < |B_i|$ that contain both inlier and outlier points relative to $c$. Thus, to prove Inequality (3), it suffices to ensure that the maximum induced error $|\mathrm{cost}^{(m_i)}(B_i, c) - (|B_i| - m_i) \cdot \mathrm{dist}(\mu(B_i), c)|$ of these three buckets is at most $\varepsilon \cdot \mathrm{cost}^{(m)}(P, c)/3$. However, it is possible that $\mathrm{cost}^{(m)}(P, c) \ll \mathrm{cost}(P, c)$, which presents a significant challenge for this guarantee compared to the vanilla case. To overcome this obstacle, prior work [39, 40] includes the "outmost" $m$ points of $P$ into the coreset to ensure zero induced error from them, resulting in an $O(m)$ size dependency.

**First attempt to eliminate the $O(m)$ dependency.** We first observe that a subset $P_M = \{p_{m+1}, \ldots, p_{n-m}\} \subset P$ of size $n - 2m$ acts as inliers w.r.t. any center $c \in \mathbb{R}$. The condition $n \geq 4m$ guarantees the existence of this $P_M$ with $|P_M| = n - 2m \geq 2m$. Since $\mathrm{cost}^{(m)}(P, c) \geq \mathrm{cost}(P_M, c)$ by the construction of $P_M$, $\mathrm{cost}^{(m)}(P, c)$ is intuitively not "too small", which is useful for eliminating the $O(m)$ dependency in analysis. A natural idea is to apply the vanilla coreset construction method given by [37] to $P_M$ and also include $P - P_M$ in the coreset $S$. This attempt yields a coreset of size $2m + \tilde{O}(\varepsilon^{-1/2})$, which already improves the previous bound of $O(m) + \tilde{O}(\varepsilon^{-2})$ by [40]. To eliminate the $O(m)$ term, it is essential to significantly reduce points in $P - P_M$.

Our first attempt is to further decompose $P - P_M$ into buckets and utilize the component-wise error analysis in Inequality (3). As discussed earlier, the critical step is to ensure $|\mathrm{cost}^{(m_i)}(B_i, c) - (|B_i| - m_i) \cdot \mathrm{dist}(\mu(B_i), c)| \leq \frac{\varepsilon}{3} \cdot \mathrm{cost}^{(m)}(P, c)$ for two partially intersecting buckets induced by $c$ with $0 < m_i < |B_i|$. To achieve this, we study the relative location of such buckets with respect to $c$

(Lemma 2.3), enabling us to partition the buckets based on the scale of $\text{cost}^{(m)}(P, c)$, similar to the vanilla method (see Lines 1-5 of Algorithm 1). However, this partitioning approach is applicable only for $c \in P_M$, where the scale of $\text{cost}^{(m)}(P, c)$ is well-controlled (see Lemma D.1). Consequently, the challenge remains to control the induced error of these buckets for $c \notin P_M$.

**Obstacle for $c \notin P_M$.** Unfortunately, to ensure Inequality (3) holds for $c \notin P_M$, the number of constructed buckets in $P - P_M$ must be at least $\Omega(m)$. To see this, we provide an illustrative example in Section A.3, where there is a collection $P_1$ of $(n - m)$ points condensed into a small interval, and a collection $P_2$ of $m$ points that are exponentially far from each other. We show that for any bucket containing two points $p, q \in P_2$ with $p < q$, the induced error of this bucket could be much larger than $\text{cost}^{(m)}(P, c)$. Thus, to ensure Inequality (3) holds, all points in $P_2$ must be included in the coreset, leading to a size of $\Omega(m)$. Therefore, a new error analysis beyond Inequality (3) is required.

**Key approach.** W.L.O.G., we assume $c > p_{n-m}$, i.e., $c$ is to the right of $P_M$. The key idea is to develop a novel non-component-wise error analysis beyond Inequality (3): we regard $|\text{cost}^{(m)}(P, c) - \text{cost}^{(m)}(S, c)|$ as a single entity and study how it changes as $c$ shifts to the right from $p_{n-m}$. Let $f'_P(c)$ and $f'_S(c)$ denote the derivative of $\text{cost}^{(m)}(P, c)$ and $\text{cost}^{(m)}(S, c)$ respectively. Then the induced error can be rewritten as $|\text{cost}^{(m)}(P, c) - \text{cost}^{(m)}(S, c)| \le |\text{cost}^{(m)}(P, p_{n-m}) - \text{cost}^{(m)}(S, p_{n-m})| + \int_{p_{n-m}}^{c} |f'_P(x) - f'_S(x)|dx$. We can ensure $\text{cost}^{(m)}(P, p_{n-m}) = \text{cost}^{(m)}(S, p_{n-m})$ by an extra bucket-partitioning step (see Line 8 of Algorithm 1). Thus, it suffices to ensure $\int_{p_{n-m}}^{c} |f'_P(x) - f'_S(x)|dx \le \varepsilon \cdot \text{cost}^{(m)}(P, c)$.

A key geometric observation is that $f'_P(c)$ equals the difference between the number of inliers in $P$ located to the left of $c$ and those to the right of $c$; a similar relationship holds for $f'_S(c)$. For $c$, let $m_i$ and $m'_i$ denote the number of outliers in bucket $B_i$ relative to dataset $P$ and coreset $S$, respectively. By the geometric observation, we conclude that $|f'_P(c) - f'_S(c)| \le \sum_i |m_i - m'_i| + 2|B_c|$, where $B_c$ denotes the bucket containing $c$. Therefore, $\int_{p_{n-m}}^{c} |f'_P(x) - f'_S(x)|dx \le (\sum_i |m_i - m'_i| + 2|B_c|) \cdot \text{dist}(p_{n-m}, c)$ (see Lemma D.3). Thus, it suffices to ensure $(\sum_i |m_i - m'_i| + 2|B_c|) \cdot \text{dist}(p_{n-m}, c) \le \varepsilon \cdot \text{cost}^{(m)}(P, c)$. This desired property can be achieved by limiting the size of each bucket in $P - P_M$ to be at most $O(\varepsilon n)$, which ensures $\sum_i |m_i - m'_i| \le O(\varepsilon n)$ for all $c \in P_M$ (see Lemma D.2).

We remark that, due to the misalignment of outlier counts, a single bucket may induce an arbitrarily large error in our analysis. However, these bucket-level errors can cancel out, and the overall error remains well controlled. To the best of our knowledge, this represents the first non-component-wise error analysis in the coreset literature, which may be of independent research interest. To demonstrate the power of our new error analysis, we apply it to the aforementioned obstacle instance—a case that previous component-wise analyses cannot solve (Appendix A.3).

**Coreset size analysis.** Note that the coreset size equals the number of buckets. We partition $P_M$ into $\tilde{O}(\varepsilon^{-1/2})$ buckets using the vanilla coreset construction method [37]. The analysis for $c \in P_M$ and $c \notin P_M$ results in at most $\tilde{O}(\varepsilon^{-1/2})$ buckets for the former and $O\left(\frac{m}{n}\varepsilon^{-1}\right)$ buckets for the latter. Therefore, the total coreset size is $\tilde{O}(\varepsilon^{-1/2} + \frac{m}{n}\varepsilon^{-1})$ (see Lemma 2.4).

This coreset size is shown to be tight. To see this, we construct a worst-case example in Section D.6, where the $m$ outliers are partitioned into $\frac{m}{n}\varepsilon^{-1}$ intervals, each containing $\varepsilon n$ points, with the interval scales increasing exponentially. If the coreset omits all points from any such interval, each point within it contributes an error of $\frac{2 \cdot \text{cost}^{(m)}(P, c)}{n}$, resulting in a total error of $2\varepsilon \cdot \text{cost}^{(m)}(P, c)$—which is unacceptably large. Thus, to control the error, the coreset must include at least one point from each interval, yielding a lower bound of $\frac{m}{n}\varepsilon^{-1}$ on the coreset size.

### 1.2.2 Overview for Theorem 1.3 (general $d$)

The obstacle of the traditional component-wise analysis remains in the high-dimensional case, motivating us to adopt the non-component-wise analysis framework. However, due to the increased complexity of candidate center distribution, a straightforward extension of the 1D analysis would involve using an $\varepsilon$-net to partition the center space, as in [35], but this introduces an $O(\varepsilon^{-d})$ factor in the coreset size. To overcome this, we leverage the concept of the ball range space, as explored in previous works [9, 39], which allows us to effectively describe high-dimensional spaces.

Our Algorithm 2 takes a uniform sample $S_O$ of size $\tilde{O}(\varepsilon^{-2} \min\{\varepsilon^{-2}, d\})$ from the "outmost" $m$ points of $P$ to include in the coreset, in contrast to including them all as in [39, 38]. To analyze the error induced by $S_O$, we examine errors caused by "outlier-misaligned" points—those that act as outliers (or inliers) in $P$ but as inliers (or outliers) in $S$ with respect to a fixed $c$. The induced error for each such point is bounded by $\frac{\text{cost}^{(m)}(P,c)}{m}$ when $n \geq 4m$ (see Lemmas E.3, E.4, and E.5). To ensure the total error remains within $O(\varepsilon) \cdot \text{cost}^{(m)}(P, c)$, it suffices for the number of such outlier-misaligned points to be $O(\varepsilon m)$. The key geometric insight is that this condition holds when $S_O$ serves as an $\varepsilon$-approximation for the ball range space on $L^\star$ (see Lemma E.6). This approximation is guaranteed by ensuring $|S_O| = \tilde{O}(\varepsilon^{-2} \min\{\varepsilon^{-2}, d\})$, as established in Lemma E.2.

To our knowledge, this analysis is the first to apply the range space argument to outlier points. The range space argument leverages the fact that the VC dimension of $\mathbb{R}^d$ is at most $O(d)$ and can be further reduced to $\tilde{O}(\varepsilon^{-2})$ by dimension reduction. This enables us to generalize results to various metric spaces via the notion of VC dimension (or doubling dimension), as well as to robust $(k, z)$-clustering; see Appendix E.2 and F.

## 2 Optimal coreset size when $d = 1$

Let $P = \{p_1, \ldots, p_n\} \subset \mathbb{R}$ with $p_1 < p_2 < \ldots < p_n$. Denote $L^{(c)} := \arg\min_{L \subseteq P, |L| = m} \text{cost}(P - L, c)$ to be the set of outliers of $P$ w.r.t. a center $c$, and $P_I^{(c)} = P - L^{(c)}$ to be the set of inliers. Let $c^\star := \arg\min_{c \in \mathbb{R}} \text{cost}^{(m)}(P, c)$ be an optimal center. Let $L^\star = L^{(c^\star)}$ and $P_I^\star = P_I^{(c^\star)}$.

**Buckets and cumulative error.** We introduce the definition of bucket proposed by [35] associated with related notions, which is useful for coreset construction when $d = 1$ [34, 37].

**Definition 2.1** (Bucket and associated statistics). A bucket $B$ is a continuous subset $\{p_l, p_{l+1}, \ldots, p_r\}$ of $P$ for some $1 \leq l \leq r \leq n$. Let $N(B) := r - l + 1$ represents the number of points within $B$, $\mu(B) := \frac{\sum_{p \in B} p}{N(B)}$ represents the mean point of $B$, and $\delta(B) := \sum_{p \in B} |p - \mu(B)|$ represents the cumulative error of $B$.

A basic idea for vanilla coreset construction in 1D case is to partition $P$ into multiple buckets $B$ and then retain a point $\mu(B)$ with weight $N(B)$ as the representative point of $B$ in coreset. This idea works since each bucket induces an error at most $\delta(B)$; see Lemma B.1. Thus, we have the following theorem that provides the optimal coreset for vanilla 1D geometric median [37]

**Theorem 2.2** (Coreset for vanilla 1D geometric median [37]). *There exists algorithm $\mathcal{A}$, that given an input data set $P \subset \mathbb{R}$ and $\varepsilon \in (0, 1)$, $\mathcal{A}(P, \varepsilon)$ outputs an $\varepsilon$-coreset of $P$ for vanilla 1D geometric median with size $\tilde{O}(\varepsilon^{-\frac{1}{2}})$ in $O(|P|)$ time.*

We will apply $\mathcal{A}$ to the "inlier subset" of $P$, say the set of middle $n - 2m$ points $P_M = \{p_{m+1}, \ldots, p_{n-m}\}$. Let $P_L = \{p_1, \ldots, p_m\}$ and $P_R = \{p_{n-m+1}, \ldots, p_n\}$. Any continuous subsequence of length $n - m$ of $P$ must contain $P_M$, implying that all points in $P_M$ must be inliers w.r.t. any center $c \in \mathbb{R}$. This motivates us to apply the vanilla method $\mathcal{A}$ to $P_M$.

As discussed in Section 1.2, we then partition $P_L$ an $P_R$ into buckets, which requires an understanding of the relative position between the inlier subset $P_I^{(c)}$ and $c$. Define $r_{\max} := \max_{p \in P_I^\star} |p - c^\star|$, $c_L := c^\star - r_{\max}$ and $c_R := c^\star + r_{\max}$. We present the following lemma (proof in Appendix B.2).

**Lemma 2.3** (Location of $P_I^{(c)}$). *Let $c \in P_M$ be a center with $c < c^\star$ and $P_I^{(c)} \neq P_I^\star$. Let $p_l$ be the leftmost point of $P_I^{(c)}$. Then $\text{dist}(p_l, c_L) \leq 2 \cdot \text{dist}(c, c^\star)$.*

A symmetric observation can be made for $c > c^\star$. Note that $\text{cost}^{(m)}(P, c) > \text{cost}(P_M, c) \geq O(n) \cdot \text{dist}(c, c^\star) \geq O(n) \cdot \text{dist}(p_l, c_L)$. This lower bound for $\text{cost}^{(m)}(P, c)$ motivates us to select the cumulative error bound for the bucket containing point $p_l$ as $\varepsilon n \cdot \text{dist}(p_l, c_L)$. Thus, we partition $P_L \cup P_R$ into disjoint *blocks* according to points' distance to $c_L$ and $c_R$, a concept inspired by [37]. Concretely, we partition the four collections $P_L \cap (-\infty, c_L)$, $P_L \cap [c_L, \infty)$, $P_R \cap (-\infty, c_R]$, $P_R \cap (c_R, \infty)$ into blocks, respectively. Due to symmetry, we only define the blocks in $P_L \cap (-\infty, c_L)$ below. Blocks for other parts follow similarly and are provided in Appendix B.3; see Figure 1 for a visualization.

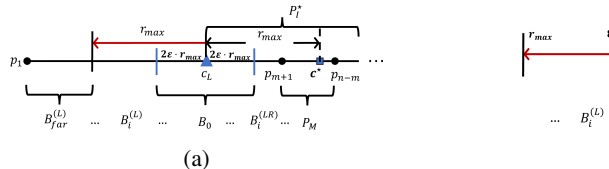

(a)                                                                    (b)

Figure 1: Illustration of the block partition. The blue square marks the optimal solution $c^\star$, and the blue triangle $c_L$ denotes the left boundary of the inlier set $P_I^\star$, with distance $r_{\max} = \text{dist}(c_L, c^\star)$. Figure 1(a) partitions the one-dimensional space left of $P_M$ into disjoint blocks based on each point's position relative to $c_L$: points farther than $r_{\max}$ form $B_{\text{far}}$, and those within $2\varepsilon r_{\max}$ form $B_0$. Figure 1(b) shows the logarithmic subdivision of inner blocks $B_i^{(L)}$ within distance $r_{\max}$ from $c_L$.

*Outer blocks ($B_{far}$):* Define $B_{far}^{(L)}$ as the set of points that are far from $c_L$, where

$$B_{far}^{(L)} := \{p \in P_L \cap (-\infty, c_L) \mid \text{dist}(p, c_L) \geq r_{\max}\}. \tag{4}$$

*Inner blocks ($B_i$):* Define $B_0^{(L)}$ as the set of points that are close to $c_L$, where

$$B_0^{(L)} := \{p \in P_L \cap (-\infty, c_L) \mid \text{dist}(p, c_L) < 2\varepsilon r_{\max}\}. \tag{5}$$

For the remaining points, partition them into blocks $B_i^{(L)}$ based on exponentially increasing distance ranges for $i = 1, \ldots, \lceil \log_2(\varepsilon^{-1}) \rceil$, where

$$B_i^{(L)} := \{p \in P_L \cap (-\infty, c_L) \mid 2^i \varepsilon r_{\max} \leq \text{dist}(p, c_L) < 2^{i+1} \varepsilon r_{\max}\}. \tag{6}$$

**The algorithm.** Our algorithm (Algorithm 1) consists of three stages. In Stage 1, we construct a coreset $S_M$ for $P_M$ using Algorithm $\mathcal{A}$ for vanilla 1D geometric median (Theorem 2.2), ensuring that $\text{cost}(S_M, c) \in (1 \pm \varepsilon) \cdot \text{cost}(P_M, c)$ for any center $c \in \mathbb{R}$. In Stage 2, we divide sets $P_L$ and $P_R$ into outer and inner blocks by Equations (4)-(6), and greedily partition these blocks into disjoint buckets $B$ with bounded $\delta(B)$ and $N(B)$ in Lines 3-6. In Stage 3, we ensure that $\text{cost}^{(m)}(P, p_{n-m}) = \text{cost}^{(m)}(S, p_{n-m})$ and $\text{cost}^{(m)}(P, p_{m+1}) = \text{cost}^{(m)}(S, p_{m+1})$ to control induced error when $c \notin P_M$. Finally, we add the mean point $\mu(B)$ of each bucket $B$ with weight $N(B)$ into $S_O$, and return $S_O \cup S_M$ as the coreset of $P$.

By construction, the coreset size $|S|$ is exactly the number of buckets. Therefore, we have the following lemma that proves the coreset size in Theorem 1.2. Its proof can be found in Section D.2.

**Lemma 2.4** (Number of buckets). *Algorithm 1 constructs at most $\tilde{O}(\varepsilon^{-\frac{1}{2}} + \frac{m}{n}\varepsilon^{-1})$ buckets.*

**Key proof idea of Theorem 1.2.** Using Lemma 2.3, we can control the error from partially intersected buckets when $c \in P_M$ (Lemma D.1). For $c \notin P_M$, let $m_i$ and $m_i'$ be the number of outliers in bucket $B_i$ for $P$ and $S$, respectively. We show that the total misaligned outliers satisfy $\sum_{i \in [q]} |m_i - m_i'| \leq \frac{\varepsilon n}{4}$ (Lemma D.2). In this case, the error is also bounded by $|\text{cost}^{(m)}(P, c) - \text{cost}^{(m)}(S, c)| \leq (\sum_{i \in [q]} |m_i - m_i'| + \frac{\varepsilon n}{8}) \cdot \text{dist}(c, P_M) \leq \varepsilon \cdot \text{cost}^{(m)}(P, c)$ (Lemma D.3). The complete proof can be found in Appendix D.

## 3   Improved coreset sizes for general $d \geq 1$

We present Algorithm 2 for Theorem 1.3. In Line 1, we construct $L^\star$ as the set of outliers of $P$ w.r.t. $c^\star$ and $P_I^\star = P - L^\star$ as the set of inliers. We construct coresets for $P_I^\star$ and $L^\star$ separately. In Line 2, we take a uniform sample $S_O$ from $L^\star$ as the coreset of $L^\star$. This step is the key for eliminating the $O(m)$ dependency in the coreset size. In Line 3, we use the following theorem by [40] to construct a coreset $S_I$ for $P_I^\star$. The coreset $S$ is the union of $S_O$ and $S_I$ (Line 5). In Section F, we show how to generalize this algorithm to robust $(k, z)$-clustering (Algorithm 3).

**Theorem 3.1** (Restatement of corollary 5.4 in [40]). *There exists a randomized algorithm $\mathcal{A}_d$ that in $O(nd)$ time constructs a weighted subset $S_I \subseteq P_I^\star$ of size $\tilde{O}(\varepsilon^{-2} \min \{\varepsilon^{-2}, d\})$, such that for every dataset $P_O$ of size $m$, every integer $0 \leq t \leq m$ and every center $c \in \mathbb{R}^d$, $|\text{cost}^{(t)}(P_O \cup P_I^\star, c) - \text{cost}^{(t)}(P_O \cup S_I, c)| \leq \varepsilon \cdot \text{cost}^{(t)}(P_O \cup P_I^\star, c) + 2\varepsilon \cdot \text{cost}(P_I^\star, c^\star).$*

---
**Algorithm 1** Coreset construction for 1D
---
**Input:** A dataset $P = \{p_1, \ldots, p_n\} \subset \mathbb{R}$ with $p_1 < \ldots < p_n$, and $\varepsilon \in (0, 1)$.
**Output:** An $\varepsilon$-coreset $S$
1: Set $P_M \leftarrow \{p_{m+1}, \ldots, p_{n-m}\}$, $P_L \leftarrow \{p_1, \ldots, p_m\}$, $P_R \leftarrow \{p_{n-m+1}, \ldots, p_n\}$.
2: Construct $S_M \leftarrow \mathcal{A}(P_M, \frac{\varepsilon}{3})$ by Theorem 2.2.
3: Compute an optimal center $c^\star$ of $P$ for robust 1D geometric median. Let $P_I^\star$ be the set of inliers w.r.t. $c^\star$, and $r_{\max} := \max_{p \in P_I^\star} \mathrm{dist}(p, c^\star)$, $c_L \leftarrow c^\star - r_{\max}$, $c_R \leftarrow c^\star + r_{\max}$.
4: Given $c_L$ and $c_R$, divide $P_L$ and $P_R$ into outer blocks $B_{far}^{(L)}$ and $B_{far}^{(R)}$ by Equation (8), and inner blocks $B_i^{(L)}$, $B_i^{(LR)}$, $B_i^{(RL)}$ and $B_i^{(R)}$ ($0 \leq i \leq \lceil \log_2(\varepsilon^{-1}) \rceil$) by Equations (9)-(10).
5: For each non-empty inner block $B_i$, divide $B_i$ into disjoint buckets $\{B_{i,j}\}_{j \geq 0}$ in a greedy way: each bucket $B_{i,j}$ is a maximal set with $\delta(B_{i,j}) \leq \frac{2^i \cdot \varepsilon^2 n r_{\max}}{288}$ and $N(B_{i,j}) \leq \frac{\varepsilon n}{16}$.
6: If $B_{far}^{(L)}$ is non-empty, divide $B_{far}$ into disjoint buckets $\{B_{far,j}^{(L)}\}_{j \geq 0}$ in a greedy way: each bucket $B_{far,j}^{(L)}$ is a maximal set with $N(B_{far,j}^{(L)}) \leq \frac{\varepsilon n}{16}$. The same for $B_{far}^{(R)}$.
7: Compute the inlier set $P_I^{(p_{m+1})}$ with respect to center $c = p_{m+1}$ and the inlier set $P_I^{(p_{n-m})}$ with respect to center $c = p_{n-m}$ respectively.
8: If there exists some bucket $B$ such that both $B \cap P_I^{(p_{m+1})}$ and $B \setminus P_I^{(p_{m+1})}$ are non-empty, divide $B$ into two buckets $B \cap P_I^{(p_{m+1})}$ and $B \setminus P_I^{(p_{m+1})}$. Do the same thing for $P_I^{(p_{n-m})}$.
9: For every $B$, add $\mu(B)$ with weight $N(B)$ into $S_O$.
10: Return $S \leftarrow S_O \cup S_M$.
---

---
**Algorithm 2** Coreset Construction for General $d$
---
**Input:** A dataset $P \subset \mathbb{R}^d$, $\varepsilon \in (0, 1)$ and an $O(1)$-approximate center $c^\star \in \mathbb{R}^d$
**Output:** An $\varepsilon$-coreset $S$
1: $L^\star \leftarrow \arg\min_{L:|L|=m} \mathrm{cost}(P - L, c^\star)$, $P_I^\star \leftarrow P - L^\star$
2: Uniformly sample $S_O \subseteq L^\star$ of size $\tilde{O}(\varepsilon^{-2} \min\{\varepsilon^{-2}, d\})$. Set $\forall p \in S_O$, $w_O(p) \leftarrow \frac{m}{|S_O|}$.
3: Construct $(S_I, w_I) \leftarrow \mathcal{A}_d(P_I^\star)$ by Theorem 3.1.
4: For any $p \in S_O$, define $w(p) = w_O(p)$ and for any $p \in S_I$, define $w(p) = w_I(p)$.
5: Return $S \leftarrow S_O \cup S_I$ and $w$;
---

The theorem tells that $S_I$ serves as an $\varepsilon$-coreset for the combination of $P_I^\star$ and any possible set of outliers $P_O$. The flexible choice of $P_O$ is useful for our analysis. To estimate the error induced by $S_O$, we introduce the key lemma below, whose proof can be found in Section E.1.

**Lemma 3.2** (Induced error of $S_O$). *For any center $c \in \mathbb{R}^d$, we have $|\mathrm{cost}^{(m)}(P, c) - \mathrm{cost}^{(m)}(S_O \cup P_I^\star, c)| \leq O(\varepsilon) \cdot \mathrm{cost}^{(m)}(P, c)$.*

*Proof of Theorem 1.3.* Fix a center $c \in \mathbb{R}^d$. Let $P_O = S_O$ and $t = m$ in Theorem 3.1, we have $|\mathrm{cost}^{(m)}(S_O \cup P_I^\star, c) - \mathrm{cost}^{(m)}(S_O \cup S_I, c)| \leq \varepsilon \cdot \mathrm{cost}^{(m)}(S_O \cup P_I^\star, c) + 2\varepsilon \cdot \mathrm{cost}(P_I^\star, c^\star)$. By Lemma 3.2, we have $|\mathrm{cost}^{(m)}(P, c) - \mathrm{cost}^{(m)}(S_O \cup P_I^\star, c)| \leq O(\varepsilon) \cdot \mathrm{cost}^{(m)}(P, c)$. Adding the two inequalities above, we have $|\mathrm{cost}^{(m)}(P, c) - \mathrm{cost}^{(m)}(S, c)| \leq O(\varepsilon) \cdot \mathrm{cost}^{(m)}(P, c)$.

The runtime is dominated by Line 1 and Line 3 that costs $O(nd)$ time by Theorem 3.1, making the total overhead $O(nd)$. This completes the proof of Theorem 1.3. $\qquad\square$

## 4  Empirical results

We implement our coreset construction algorithm and compare its performance to several baselines. All experiments are conducted on a PC with an Intel Core i9 CPU and 16GB of memory, and the algorithms are implemented in C++ 11.

**Baselines.** We compare our algorithm with two baselines: 1) Method **HJLW23** proposed by [39], which directly includes $L^\star$ in the coreset and samples points from $P_I^\star$. 2) Method **HLLW25** proposed by [40], improves the sample size from $P_I^\star$ in [39].

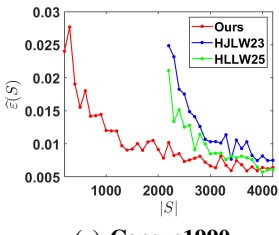
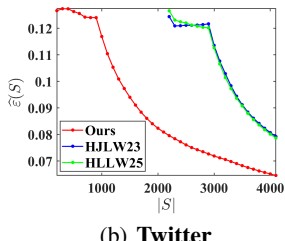
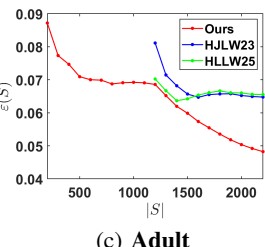

| (a) **Census1990** | (b) **Twitter** | (c) **Adult** |

Figure 2: Tradeoff between coreset size $|S|$ and empirical error $\widehat{\varepsilon}(S)$.

**Setup.** We conduct experiments on six datasets from diverse domains, including social networks, demographics, and disease statistics, with sample sizes ranging from $(10^4)$ to $(10^5)$ and feature dimensions from 2 to 68, as summarized in Table 4. These datasets cover those used in baseline [39], ensuring fair comparison. In each dataset, numerical features are extracted to create a vector for each record and the outlier number is set to $2\%$ of the dataset size. To simplify computation, we subsample $10^5$ points from the **Twitter** and **Census1990** datasets, and $10^4$ points from the **Athlete** and **Diabetes** datasets, respectively. We use $k$-means++ to compute an approximate center $c^\star$.

**Size-error tradeoff.** We evaluate the tradeoff between coreset size and empirical error. Given a (weighted) subset $S \subseteq P$ and a center $c \subset \mathbb{R}^d$, we define the empirical error $\widehat{\varepsilon}(S, c) := \frac{|\text{cost}^{(m)}(P,c) - \text{cost}^{(m)}(S,c)|}{\text{cost}^{(m)}(P,c)}$, where lower values indicate better coreset performance for $c$. It is difficult to estimate the performance for every center. So we sample 500 centers $c_i \in \mathbb{R}^d$, where each $c_i$ is drawn uniformly from $P$ without replacement. Like in the literature [39], we evaluate the empirical error $\widehat{\varepsilon}(S) := \max_{i \in [500]} \widehat{\varepsilon}(S, c_i)$. We vary the coreset size $|S|$ from $m$ to $2m$, and compute the empirical error $\widehat{\varepsilon}(S)$. For each size and each algorithm, we run the algorithm 10 times, compute their empirical errors $\widehat{\varepsilon}(S)$, and report the average of 10 empirical errors.

Figure 2 presents the empirical results illustrating the size-error tradeoff on the **Census1990**, **Twitter** and **Adult** datasets. As shown in Figures 2, our coreset algorithm consistently achieves the lowest empirical error among all methods. Moreover, unlike the baselines, which require a coreset of size at least $m$, our method attains the same level of error with a coreset size **smaller than** $m$. For example, with the **Census1990** dataset, our method yields a coreset of size 1000 with an empirical error of 0.012, while the best baseline needs size 2300 to achieve a worse error of 0.013. Results for other datasets are presented in Section G. We also perform statistical tests across six real-world datasets by comparing the ratio of empirical errors between our algorithm and baselines, which further demonstrates that our algorithm consistently outperforms the baselines; see Table 1. The results show that both $\widehat{\varepsilon}(S_2)/\widehat{\varepsilon}(S_1)$ and $\widehat{\varepsilon}(S_3)/\widehat{\varepsilon}(S_1)$ are consistently bigger than 1, demonstrating that our coreset consistently yields lower empirical error than the baselines. This confirms the applicability of our coreset across real-world datasets.

**Speed-up baselines.** In this experiment, we compare the coreset of size $2m$ constructed by the **HLLW25** baseline and coreset of size $m$ constructed by Algorithm 2. We repeat the experiment 10 times and report the averages. The result is listed in Table 2. The construction time of our coreset is similar to that of the baseline **HLLW25**. However, our algorithm achieves a speed-up over **HLLW25** (a $2\times$ reduction in the running time on the coreset), while achieving the same level of empirical error.

**Additional experiment.** In Section G.1, we demonstrate the robustness of our Algorithm 2 when the assumption $n \geq 4m$ is violated or dataset is noisy. In Sections G.2, G.3, and G.4, we implement our algorithms for the 1D case, robust $k$-median, and $k$-means, respectively, and conduct similar experiments. The results show improved performance on real-world datasets, even when the theoretical data assumptions are violated, further highlighting the practical robustness of our algorithms.

## 5 Conclusion and future work

We investigate coreset construction for robust geometric median problem, successfully eliminating the size dependency on the number of outliers. Specifically, for the 1D Euclidean case, we achieve the first optimal coreset size. Furthermore, our results generalize to robust clustering applications. Empirically, our algorithms achieve a superior size-error balance and a runtime acceleration.

Table 1: Statistical comparison of different coreset construction methods for robust geometric median. The coreset $S_1$ represents our coreset, $S_2$ represents the coreset constructed by the baseline **HJLW23**, and $S_3$ the coreset constructed by baseline **HLLW25**. For each empirical error ratio $\widehat{\varepsilon}(S_2)/\widehat{\varepsilon}(S_1)$ and $\widehat{\varepsilon}(S_3)/\widehat{\varepsilon}(S_1)$, we report the mean value over 20 runs, with the subscript indicating the standard deviation.

| Coreset Size | Census1990 | | Twitter | |
|---|---|---|---|---|
| | $\widehat{\varepsilon}(S_2)/\widehat{\varepsilon}(S_1)$ | $\widehat{\varepsilon}(S_3)/\widehat{\varepsilon}(S_1)$ | $\widehat{\varepsilon}(S_2)/\widehat{\varepsilon}(S_1)$ | $\widehat{\varepsilon}(S_3)/\widehat{\varepsilon}(S_1)$ |
| 2200 | $3.253_{2.063}$ | $2.645_{1.458}$ | $1.793_{0.644}$ | $1.667_{0.479}$ |
| 3200 | $1.257_{0.842}$ | $1.251_{0.632}$ | $1.343_{0.234}$ | $1.283_{0.197}$ |
| 4200 | $1.303_{0.692}$ | $1.168_{0.739}$ | $1.244_{0.152}$ | $1.246_{0.148}$ |

| Coreset Size | Bank | | Adult | |
|---|---|---|---|---|
| | $\widehat{\varepsilon}(S_2)/\widehat{\varepsilon}(S_1)$ | $\widehat{\varepsilon}(S_3)/\widehat{\varepsilon}(S_1)$ | $\widehat{\varepsilon}(S_2)/\widehat{\varepsilon}(S_1)$ | $\widehat{\varepsilon}(S_3)/\widehat{\varepsilon}(S_1)$ |
| 1200 | $1.647_{0.972}$ | $1.360_{1.018}$ | $1.467_{0.287}$ | $1.094_{0.542}$ |
| 1700 | $1.010_{0.654}$ | $1.028_{0.574}$ | $2.149_{0.884}$ | $2.416_{1.002}$ |
| 2200 | $1.010_{0.654}$ | $1.026_{0.674}$ | $1.089_{0.360}$ | $1.172_{0.537}$ |

| Coreset Size | Athlete | | Diabetes | |
|---|---|---|---|---|
| | $\widehat{\varepsilon}(S_2)/\widehat{\varepsilon}(S_1)$ | $\widehat{\varepsilon}(S_3)/\widehat{\varepsilon}(S_1)$ | $\widehat{\varepsilon}(S_2)/\widehat{\varepsilon}(S_1)$ | $\widehat{\varepsilon}(S_3)/\widehat{\varepsilon}(S_1)$ |
| 210 | $5.172_{3.634}$ | $4.200_{1.944}$ | $5.700_{3.303}$ | $5.868_{2.952}$ |
| 310 | $2.467_{1.564}$ | $1.427_{0.660}$ | $1.567_{0.800}$ | $1.332_{0.653}$ |
| 410 | $1.658_{0.881}$ | $1.045_{0.449}$ | $1.360_{1.103}$ | $1.216_{0.943}$ |

This work opens several intriguing research directions. One immediate problem is to optimize the coreset size for $d > 1$ or $k > 1$, particularly in cases where the size diverges from that of the vanilla setting. Extending robust coresets to the streaming model is a valuable but challenging direction. The primary obstacle is their lack of mergeability, as outlier interactions across different data chunks prevent the compositional updates essential for streaming algorithms. It is also interesting to explore whether our non-component-wise analysis can be applied to other robust machine learning problems, such as robust regression and robust PCA.

Table 2: Comparison of runtime between our Algorithm 2 and baseline **HLLW25**. For each dataset, the coreset size of baseline **HLLW25** is $2m$ and the coreset size of ours is $m$. We use Lloyd algorithm given by [7] to compute approximate solutions $c_P$ and $c_S$ for both the original dataset $P$ and coreset $S$, respectively. "COST$_P$" denotes $\mathrm{cost}^{(m)}(P, c_P)$ on the original dataset $P$. "COST$_S$" denotes $\mathrm{cost}^{(m)}(P, c_S)$ on the coreset constructed by METHOD. $T_X$ is the running time on the original dataset. $T_S$ is the running time on coreset. $T_C$ is the construction time of the coreset.

| DATASET | COST$_P$ | METHOD | COST$_S$ | $T_X$ | $T_C$ | $T_S$ |
|---|---|---|---|---|---|---|
| CENSUS1990 | $5.099{\times}10^6$ | OURS | $5.100{\times}10^6$ | 63.425 | 6.876 | 1.284 |
| | | HLLW25 | $5.099{\times}10^6$ | | 6.950 | 2.629 |
| TWITTER | $7.307{\times}10^6$ | OURS | $7.310{\times}10^6$ | 41.816 | 3.233 | 0.633 |
| | | HLLW25 | $7.307{\times}10^6$ | | 3.259 | 1.278 |
| BANK | $7.815{\times}10^6$ | OURS | $7.760{\times}10^6$ | 16.555 | 1.427 | 0.308 |
| | | HLLW25 | $7.765{\times}10^6$ | | 1.477 | 0.677 |
| ADULT | $3.418{\times}10^9$ | OURS | $3.412{\times}10^9$ | 16.907 | 1.632 | 0.310 |
| | | HLLW25 | $3.411{\times}10^9$ | | 1.596 | 0.597 |
| ATHLETE | $1.460{\times}10^5$ | OURS | $1.467{\times}10^5$ | 2.92 | 0.320 | 0.055 |
| | | HLLW25 | $1.463{\times}10^5$ | | 0.321 | 0.104 |
| DIABETES | $1.781{\times}10^5$ | OURS | $1.786{\times}10^5$ | 3.977 | 0.366 | 0.062 |
| | | HLLW25 | $1.788{\times}10^5$ | | 0.360 | 0.135 |

## Acknowledgment

LH acknowledges support from the New Cornerstone Science Laboratory, and NSFC Grant No. 625707396.

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

# Contents

Table 3: Comparison of the state-of-the-art coreset size and our results for robust geometric median and robust $k$-median in $\mathbb{R}^d$. Robust $k$-median is a generalization of robust geometric median; see Definition F.1 when $z = 1$.

| PARAMETERS $d,k$ | PRIOR RESULTS | OUR RESULTS (ASSUMING $n \geq 4m$) |
|---|---|---|
| $d = 1$ $\quad$ $k = 1$ | $O(m) + \tilde{O}(\varepsilon^{-2})$[40] 
 $\tilde{O}(m\varepsilon^{-1})$ + VANILLA SIZE[42] 
 $\Omega(m)$ [39] | $\tilde{O}(\varepsilon^{-\frac{1}{2}} + \frac{m}{n}\varepsilon^{-1})$ (THEOREM 1.2) 
 $\Omega(\varepsilon^{-\frac{1}{2}} + \frac{m}{n}\varepsilon^{-1})$ (THEOREM 1.2) |
| $d > 1$ $\quad$ $k = 1$ | $O(m) + \tilde{O}(\varepsilon^{-2} \cdot \min\{\varepsilon^{-2}, d\})$[40] 
 $\tilde{O}(m\varepsilon^{-1})$ + VANILLA SIZE[42] 
 $\Omega(m)$ [39] | $\tilde{O}(\varepsilon^{-2} \cdot \min\{\varepsilon^{-2}, d\})$ (THEOREM 1.3) 
 $\Omega(\varepsilon^{-\frac{1}{2}} + \frac{m}{n}\varepsilon^{-1})$ (THEOREM 1.2) |
| $d > 1$ $\quad$ $k > 1$ | $O(m) + \tilde{O}(k^2\varepsilon^{-2} \cdot \min\{\varepsilon^{-2}, d\})$[40] 
 $\tilde{O}(\min\{km\varepsilon^{-1}, m\varepsilon^{-2}\})$ + VANILLA SIZE[42] 
 $\Omega(m)$[39] | $\tilde{O}(k^2\varepsilon^{-2} \cdot \min\{\varepsilon^{-2}, d\})$ 
 (THEOREM 1.5, UNDER ASSUMPTION 1.4) |

# A Omitted details in Section 1

## A.1 Formal definition of coreset

In this section, we define the coreset for robust geometric median. For preparation, we first generalize the cost function $\text{cost}^{(m)}$ to handle weighted datasets.

**Definition A.1** (Generalized cost function). Let $m$ be an integer. Let $S \subseteq \mathbb{R}^d$ be a weighted dataset with weights $w(p)$ for each point $p \in S$. Let $w(S) := \sum_{p \in S} w(p)$. Define a collection of weight functions $\mathcal{W} := \{w' : S \to \mathbb{R}^+ \mid \sum_{p \in S} w'(p) = w(S) - m \wedge \forall p \in S, w'(p) \leq w(p)\}$. Moreover, we define the following cost function on $S$: $\forall c \in \mathbb{R}^d$, $\text{cost}^{(m)}(S, c) := \min_{w' \in \mathcal{W}} \sum_{p \in S} w'(p) \cdot \text{dist}(p, c)$.

Intuitively, to compute $\text{cost}^{(m)}(S, c)$, we find a weighted subset $S'$ of $S$ with total weight $w(S) - m$ that minimizes its vanilla cost to $c$. Thus, this $\text{cost}^{(m)}(S, c)$ serves as the cost function for robust geometric median on a weighted set. Note that for the unweighted case where $w(p) = 1$ for all $p \in S$, this cost function reduces to that in Equation (1).

Next, we define the notion of coreset for robust geometric median.

**Definition A.2** (Coreset for robust geometric median). Given a point set $P \subset \mathbb{R}^d$ of size $n \geq 1$, integer $m \geq 1$ and $\varepsilon \in (0, 1)$, we say a weighted subset $S \subseteq P$ together with a weight function $w : S \to \mathbb{R}^+$ is an $\varepsilon$-coreset of $P$ for robust geometric median if $w(S) = n$ and for any center $c \in \mathbb{R}^d$, $\text{cost}^{(m)}(S, c) \in (1 \pm \varepsilon) \cdot \text{cost}^{(m)}(P, c)$.

This formulation ensures that the weighted coreset $S$ provides an accurate approximation of the original dataset $P$'s cost for all centers $c$, within a tolerance specified by $\varepsilon$.

## A.2 Other related work

**Coreset for robust clustering.** A natural extension of robust geometric median is called robust $(k, z)$-Clustering, attracting considerable interest for its coreset construction techniques in the literature [27, 38, 39, 41, 53]. In early work, [27] proposed a coreset construction method for the robust $k$-median problem, which requires an exponentially large size $(k + m)^{O(k+m)}(\varepsilon^{-1}d\log n)^2$. Recently, [39] improved the coreset size to $O(m) + \tilde{O}(k^3\varepsilon^{-3z-2})$ via a hierarchical sampling framework proposed by [9]. Following this, [40] further improved the size to $O(m) + \tilde{O}(k^2\varepsilon^{-2z-2})$. More recently, [42] proposed a new coreset of size $O(\min\{km\varepsilon^{-1}, m\varepsilon^{-2z}\})$ + Vanilla size. We give Table 3 to compare our theoretical results with prior work.

**Coreset for other clustering problems.** Coreset construction for other variants of $(k, z)$-Clustering problems has also been extensively studied, including vanilla clustering [35, 16, 26, 9, 18, 19, 37, 41], capitalized clustering [9, 20], fair clustering [17, 5] and fault-tolerant clustering [44, 33]. Specifically, for vanilla $(k, z)$-clustering recent advancements by [19, 18, 41], produced a coreset

of size $\tilde{O}(\min\left\{k\varepsilon^{-z-2}, k^{\frac{2z+2}{z+2}}\varepsilon^{-2}\right\})$. When $\varepsilon \geq k^{-\frac{1}{z+2}}$, the coreset upper bound $k\varepsilon^{-z-2}$ is shown to be optimal by a recent breakthrough [41]. Furthermore, a recent study [37] has investigated the coreset bounds when $d$ is small.

### A.3 An illustrative example for the obstacle of Inequality (3)

Recall that we partition the input dataset $P \subset \mathbb{R}$ into disjoint buckets $\{B_1, \ldots, B_T\}$, constructs a representative point $\mu(B_i)$ with weight $|B_i|$ for each bucket and takes their union as a coreset $S$ of $P$ for robust 1D geometric median. Also, recall that prior work [39, 40] use Inequality (3) for error analysis, i.e., for any center $c \in \mathbb{R}$ and any tuple of outlier numbers $(m_1, \ldots, m_T)$ with $\sum_{i \in [T]} m_i = m$,

$$\sum_{i \in [T]} |\text{cost}^{(m_i)}(B_i, c) - (|B_i| - m_i)\text{dist}(\mu(B_i), c)| < \varepsilon \cdot \text{cost}^{(m)}(P, c). \tag{7}$$

In this section, we provide an example in which this inequality only holds if $|S| = \Omega(m)$.

Construct the dataset $P$ as follows: for $i = 1, \ldots, n - m$, $p_i = \frac{i}{n}$; for $n - m + 1 \leq i \leq n$, $p_i = n^{3(i-n+m)}$. We show that the Inequality (7) only holds if each $p_j$ forms its own isolated bucket. Assume, for the sake of contradiction, that there exists a bucket $B_q = \{p_i, \ldots, p_j\}$ such that $|B_q| > 1$ and $j > n - m$.

**Case 1:** If $i \leq n - m$, let $c = 0$. Then we have that the inlier set with respect to $c$ is $P_I^{(c)} = \{p_1, \ldots, p_{n-m}\}$. Thus,

$$\text{cost}^{(m)}(P, c) = \text{cost}(P_I^{(c)}, c) = \sum_{t=1}^{n-m} \frac{t}{n} < n.$$

Let $m_1 = \ldots = m_{q-1} = 0$, $m_q = j - n + m$ and $m_t = |B_t|$ for $q + 1 \leq t \leq T$. We have

$$\sum_{i \in [T]} |\text{cost}^{(m_i)}(B_i, c) - (|B_i| - m_i)\text{dist}(\mu(B_i), c)|$$

$$\begin{aligned}
&= & |\text{cost}^{(m_q)}(B_q, c) - (|B_q| - m_q)\text{dist}(\mu(B_q), c)| & \\
&> & (|B_q| - m_q) \cdot \mu(B_q) - n & (\text{cost}^{(m_q)}(B_q, c) \leq \text{cost}^{(m)}(P, c) < n) \\
&\geq & \mu(B_q) - n & (|B_q| = j - i + 1, i \leq n - m) \\
&= & \frac{\sum_{p \in B_q} p}{|B_q|} - n & \\
&> & n^2 - n & \\
&> & \varepsilon \cdot \text{cost}^{(m)}(P, c), & (\text{cost}^{(m)}(P, c) < n)
\end{aligned}$$

which leads to a contradiction with Inequality (7).

**Case 2:** If $i > n - m$, let $c = p_i$. Then $P_I^{(c)} = \{p_{i-n+m+1}, \ldots, p_i\}$. In this case, we have

$$\text{cost}^{(m)}(P, c) = \sum_{p \in P_I^{(c)}} d(p, p_i) \leq (n - m) \cdot n^{3(i-n+m)}.$$

Note that $|B_q \cap P_I^{(c)}| = 1$, which means $|B_q| - m_q = 1$, then we have

$$\sum_{i \in [T]} |\text{cost}^{(m_i)}(B_i, c) - (|B_i| - m_i)\text{dist}(\mu(B_i), c)|$$

$$\begin{aligned}
&\geq & |\text{cost}^{(m_q)}(B_q, c) - (|B_q| - m_q)\text{dist}(\mu(B_q), c)| & \\
&= & \text{dist}(\mu(B_q), c) & (\text{cost}^{(m_q)}(B_q, c) = \text{dist}(p_i, c) = 0) \\
&= & \frac{\sum_{p \in B_q} p}{|B_q|} - p_i & \\
&> & n^{3(i-n-m)} \cdot (n^2 - 1) & \\
&> & \varepsilon \cdot \text{cost}^{(m)}(P, c), & (\text{cost}^{(m)}(P, c) \leq (n - m) \cdot n^{3(i-n+m)})
\end{aligned}$$

which leads to a contradiction with Inequality (7). Overall, Inequality (7) does not hold if $\{p_j\}$ does not form a separated bucket.

**Non-component-wise analysis breaks the obstacle.** We then show how to adapt our new non-component-wise analysis to this example, which allows a more careful bucket decomposition. The key is that in our analysis, the induced error of the aforementioned bucket $B_q = \{p_i, \ldots, p_j\}$ with $j > n - m$ is 0, due to allowing the misaligned outlier numbers in each bucket. Below illustrate the above two cases.

**Case 1:** If $i \leq n - m$, $c = 0$, then only the bucket $B_q$ may induce an error. Since $j > n - m$, we have $p_j \geq n^3$, thus

$$\mu(B_q) = \frac{\sum_{p \in B_q} p}{|B_q|} > \frac{n^3}{n} = n^2 > \max_{p \in P_I^{(c)}} \mathrm{dist}(p, c).$$

Therefore, $B_q$ will be totally regarded as an outlier which induces 0 error, thus $\mathrm{cost}^{(m)}(S, c) = \mathrm{cost}^{(m)}(P, c)$.

**Case 2:** If $i > n - m$, $c = p_i$, then $P_I^{(c)} = \{p_{i-n+m+1}, \ldots, p_i\}$. Let the bucket $B_L = \{p_l, \ldots, p_r\}$ with $l \leq i - n + m + 1$, $r \geq i - n + m + 1$ denote the leftmost bucket containing at least one inlier. Therefore, only the buckets $B_L$ and $B_q$ may induce an error. For $B_q$, we have $\mathrm{dist}(\mu(B_q), c) > (n^2 - 1)\mathrm{dist}(p_1, c)$, thus $B_q$ induces 0 error. For $B_L$, the induced error is bounded by $\varepsilon n \cdot n^{3(i-n-m)} \leq \varepsilon \cdot \mathrm{cost}^{(m)}(P, c)$, since, in our framework, the size of each bucket in $P - P_M$ is restricted to at most $O(\varepsilon n)$. In summary, the total error is bounded by $\varepsilon \cdot \mathrm{cost}^{(m)}(P, c)$.

Thus, non-component-wise analysis significantly reduces bucket-wise errors in this example, which is crucial for proving $S$ is a coreset.

# B  Omitted details in Section 2

## B.1  Property of buckets

In Section 2, we introduce a useful notion called bucket. The following lemma shows that the coreset error on each bucket $B$ is bounded by $\delta(B)$. Recall that $\mathrm{cost}(B, c) = \sum_{p \in B} \mathrm{dist}(p, c)$ for any point set $B \subset \mathbb{R}$ and center $c \in \mathbb{R}$.

**Lemma B.1** (Error analysis for buckets [34]). *Let $B = \{p_l, \ldots, p_r\} \subseteq P$ for $1 \leq l \leq r \leq n$ be a bucket and $c \in \mathbb{R}$ be a center. We have*

*1. if $c \in (p_l, p_r)$, $|\mathrm{cost}(B, c) - N(B) \cdot \mathrm{dist}(\mu(B), c)| \leq \delta(B)$;*

*2. if $c \notin (p_l, p_r)$, $|\mathrm{cost}(B, c) - N(B) \cdot \mathrm{dist}(\mu(B), c)| = 0$.*

## B.2  Proof of Lemma 2.3

*Proof of Lemma 2.3.* We prove the lemma by contradiction. Suppose there exists a center $c \in P_M$ satisfying that $c < c^\star$ and $P_I^{(c)} \neq P_I$, and the leftmost point $p_l$ of $P_I^{(c)}$ satisfies $\mathrm{dist}(p_l, c_L) > 2\mathrm{dist}(c, c^\star)$. Then by the assumption of $c$, we have $p_l < c_L$, thus $p_l \notin P_I^\star$. For any points $p$ in $P_I^\star$, we have

$$
\begin{aligned}
\mathrm{dist}(p, c) &\leq \mathrm{dist}(p, c^\star) + \mathrm{dist}(c, c^\star) &&\text{(Triangle Inequality)} \\
&\leq r_{\max} + \mathrm{dist}(c, c^\star) &&\text{(Definition of } r_{\max}) \\
&< r_{\max} + \mathrm{dist}(p_l, c_L) - \mathrm{dist}(c, c^\star) &&(\mathrm{dist}(p_l, c_L) > 2\mathrm{dist}(c, c^\star)) \\
&= \mathrm{dist}(p_l, c)
\end{aligned}
$$

This implies that any point in $P_I^\star$ must be in $P_I^{(c)}$. However, since $|P_I^\star| = |P_I^{(c)}| = n - m$, $P_I^{(c)}$ cannot contain any other point not in $P_I^\star$. This contradicts $p_l \notin P_I^\star$. Thus, $\mathrm{dist}(p_l, c_L) \leq 2\mathrm{dist}(c, c^\star)$ holds. $\qquad \square$

### B.3 Complete definition of blocks

Now we provide the complete definition of blocks, which are used in our algorithm 1. Recall that $r_{\max} = \max_{p \in P_I^\star} |p - c^\star|$, $c_L = c^\star - r_{\max}$ and $c_R = c^\star + r_{\max}$. Then we divide the sets $P_L$ and $P_R$ into disjoint blocks as follows:

**Outer blocks** ($B_{far}$): Define $B_{far}^{(L)}$ and $B_{far}^{(R)}$ as the set of points that are far from $c_L$ and $c_R$, where

$$
\begin{aligned}
B_{far}^{(L)} &:= \{\, p \in P_L \cap (-\infty, c_L) \mid \mathrm{dist}(p, c_L) \geq r_{\max} \,\}, \\
B_{far}^{(R)} &:= \{\, p \in P_R \cap (c_R, \infty) \mid \mathrm{dist}(p, c_R) \geq r_{\max} \,\}.
\end{aligned}
\tag{8}
$$

**Inner blocks** ($B_i$): Define $B_0^{(L)}$, $B_0^{(LR)}$, $B_0^{(R)}$, and $B_0^{(RL)}$ as the set of points that are close to $c_L$ or $c_R$, where

$$
\begin{aligned}
B_0^{(L)} &:= \{\, p \in P_L \cap (-\infty, c_L) \mid \mathrm{dist}(p, c_L) < 2\varepsilon r_{\max} \,\}, \\
B_0^{(LR)} &:= \{\, p \in P_L \cap [c_L, \infty) \mid \mathrm{dist}(p, c_L) < 2\varepsilon r_{\max} \,\}, \\
B_0^{(R)} &:= \{\, p \in P_R \cap (c_R, \infty) \mid \mathrm{dist}(p, c_R) < 2\varepsilon r_{\max} \,\}, \\
B_0^{(RL)} &:= \{\, p \in P_R \cap (-\infty, c_R] \mid \mathrm{dist}(p, c_R) < 2\varepsilon r_{\max} \,\}.
\end{aligned}
\tag{9}
$$

For the remaining points, partition them into blocks $B_i^{(L)}$, $B_i^{(LR)}$, $B_i^{(R)}$ and $B_i^{(RL)}$ based on exponentially increasing distance ranges for $i = 1, \ldots, \lceil \log_2(\varepsilon^{-1}) \rceil$, where

$$
\begin{aligned}
B_i^{(L)} &:= \{\, p \in P_L \cap (-\infty, c_L) \mid 2^i \varepsilon r_{\max} \leq \mathrm{dist}(p, c_L) < 2^{i+1} \varepsilon r_{\max} \,\}, \\
B_i^{(LR)} &:= \{\, p \in P_L \cap [c_L, \infty) \mid 2^i \varepsilon r_{\max} \leq \mathrm{dist}(p, c_L) < 2^{i+1} \varepsilon r_{\max} \,\}, \\
B_i^{(R)} &:= \{\, p \in P_R \cap (c_R, \infty) \mid 2^i \varepsilon r_{\max} \leq \mathrm{dist}(p, c_R) < 2^{i+1} \varepsilon r_{\max} \,\}, \\
B_i^{(RL)} &:= \{\, p \in P_R \cap (-\infty, c_R] \mid 2^i \varepsilon r_{\max} \leq \mathrm{dist}(p, c_R) < 2^{i+1} \varepsilon r_{\max} \,\}.
\end{aligned}
\tag{10}
$$

### B.4 Justifying the selection of the optimal center

In previous coreset constructions [39], it is hard to obtain the exact value of optimal center $c^\star$, so an approximate center is generally used instead. However, when $d = 1$, $P_I^\star$ is always a continuous subsequence of $P$. Suppose $P_I^\star = \{p_i, \ldots, p_j\}$, we have $c^\star = p_{\lfloor \frac{i+i}{2} \rfloor}$. This indicates that $P_I^\star$ and $c^\star$ can be computed in polynomial time in robust 1D geometric median. Additionally, computing an $O(1)$-approximation of $c^\star$ is also sufficient for our algorithm to remain valid, which only results in a factor of $O(1)$ difference in the coreset size.

## C Proof of Theorem 1.1: coreset lower bound for robust geometric median

We first construct a bad instance $P := \{p_1, \ldots, p_n\} \subset \mathbb{R}$, where $p_i = i$ for $i \in [m]$ and $p_j = x$ for $m < j \leq n$, $x \to \infty$.

W.l.o.g, we assume $n - m$ is an even number and $n - m \leq (m-1)/2$. Suppose $(S, w)$ is an $\varepsilon$-coreset of size $|S| < \frac{m}{2(n-m)+1}$ for robust geometric median on $P$. Define $S_I := S \cap \{p_{m+1}, \ldots, p_n\}$ and $S_O := S \cap \{p_1, \ldots, p_m\}$. There exists $2(n-m)$ consecutive points $p_{i+1}, \ldots, p_{i+2(n-m)}$ not in $S$ for some $i \in [m - 2(n-m)]$. Let the center $c = i + (n-m) + \frac{1}{2}$. Then we have

$$
\begin{aligned}
\mathrm{cost}^{(m)}(P, c) = \quad & 2\left(\left(\frac{1}{2}\right) + \left(\frac{1}{2} + 1\right) + \ldots + \left(\frac{1}{2} + \frac{n-m}{2} - 1\right)\right) \\
\leq \quad & \frac{(n-m)^2}{2}.
\end{aligned}
\tag{11}
$$

We claim that $w(S_O) + w(S_I) - m \geq (1 - \varepsilon) \cdot (n - m)$. Fix a center $c' = x + 1$. If $w(S_O) > m$, we have $\mathrm{cost}^{(m)}(P, c') = n - m$ and $\mathrm{cost}^{(m)}(S, c') \to \infty$, leading to an unbounded error. Therefore

we only need to consider the case $w(S_O) \leq m$ and our claim can be verified by as follow:

$$\frac{\text{cost}^{(m)}(S, c')}{\text{cost}^{(m)}(P, c')} = \frac{w(S_O) - m + w(S_I)}{n - m} \to (1 \pm \varepsilon). \tag{12}$$

By this claim, we have

$$\begin{aligned}
\text{cost}^{(m)}(S, c) &\geq & (n - m + 0.5) \cdot (w(S_O) + w(S_I) - m) & \\
&\geq & (n - m + 0.5) \cdot (1 - \varepsilon) \cdot (n - m) & \text{(Inequality (12))} \\
&> & (1 + \varepsilon) \cdot (\frac{(n - m)^2}{2}) & (0 < \varepsilon < 0.5) \\
&\geq & (1 + \varepsilon) \cdot \text{cost}^{(m)}(P, c) & \text{(Inequality (11))}.
\end{aligned}$$

In summary, we have $\text{cost}^{(m)}(S, c) \geq (1 + \varepsilon) \cdot \text{cost}^{(m)}(P, c)$, which contradicts the definition of $\varepsilon$-coreset. We conclude that, each $\varepsilon$-coreset of $P$ is of size $\Omega(\frac{m}{n-m})$.

## D  Proof of Theorem 1.2: optimal coreset size for robust 1D geometric median

In this section, we provide the complete proof of Theorem 1.2 for both upper and lower bounds.

### D.1  Proof of the upper bound in Theorem 1.2

Now we show that our coreset $S$ obtained by Algorithm 1 is an $\varepsilon$-coreset of $P$.

In the following discussion, we define $S_I^{(c)}$ and $S_I^{\star}$ as follows. Given a weighted set $S$ of total weight $n$ and a center $c \in \mathbb{R}$, let $S_I^{(c)} := \arg\min_{S_I^{(c)} \subseteq S, \sum_{s \in S_I^{(c)}} w(s) = n - m} \text{cost}(S_I^{(c)}, c)$ be the set of inliers of $S$, where $w(s)$ represents the weight of $s$ in $S_I^{(c)}$ and is at most the weight of $s$ in $S$. Moreover, let $S_I^{\star}$ represents $S_I^{(c^{\star})}$, which exactly contains every $\mu(B)$ of each bucket $B$ in $P_I^{\star}$ with weight $|B|$.

First, we consider the case that $c \in P_M$. Note that regardless of the position of $c$, at most three buckets can induce the error: the bucket containing $c$, and the buckets containing the endpoints of $P_I^{(c)}$ on either side. Actually, the coreset error in $P_M$ is already controlled by the vanilla coreset construction algorithm $\mathcal{A}$. Thus, we only have to consider the cumulative error of the buckets that partially intersect with $P_I^{(c)}$ on either side. We will show that this error is controlled by the carefully selected cumulative error bound of each inner block. Moreover, we adapt the analysis for the vanilla case in [37] to the robust case, as shown in Lemma D.1. This allows us to avoid considering the error caused by the misaligned outliers in $P$ and $S$, which still suffice to ensure that the coreset error is bounded. The proof of Lemma D.1 is deferred to Section D.3.

**Lemma D.1** (Error analysis for $c \in P_M$). *When center $c \in (p_{m+1}, p_{n-m})$, $|\text{cost}^{(m)}(P, c) - \text{cost}^{(m)}(S, c)| \leq \varepsilon \cdot \text{cost}^{(m)}(P, c)$.*

Now we analyze the case that $c$ is not in $P_M$. Assume $P$ is divided into disjoint buckets $B_1, \ldots, B_q$, from left to right. Fix a center $c \in \mathbb{R}$, for each bucket $B_i$ ($i \in [q]$), define $m_i := |B_i \setminus P_I^{(c)}|$ and $m_i' := |B_i \setminus S_I^{(c)}|$, which represent the number of outliers in $B_i$ with respect to $c$. Obviously we have $\sum_i m_i = \sum_i m_i' = m$. The following lemma shows that the number of inliers in each bucket for $P_I^{(c)}$ and $S_I^{(c)}$ remains roughly consistent, which indicates that $\text{cost}^{(m)}(P, c)$ and $\text{cost}^{(m)}(S, c)$ increase almost equally.

**Lemma D.2.** *For any center $c \in \mathbb{R}$, $\sum_{i \in [q]} |m_i - m_i'| \leq \frac{\varepsilon n}{4}$.*

Let $\Gamma := \sup_{c \in \mathbb{R}} \sum_{i \in [q]} |m_i - m_i'|$, then we have $\Gamma \leq \frac{\varepsilon n}{4}$. For $c \notin P_M$, we consider the derivative of the cost value with respect to $c$, and gives an upper bound of the induced error in Lemma D.3. Combined with the upper bound of $\Gamma$, we conclude that the induced error is $O(\varepsilon n \cdot \text{dist}(c, c^{\star}))$, which is bounded by $O(\varepsilon) \cdot \text{cost}^{(m)}(P, c)$ obviously. The main idea is similar to that of the proof of Theorem 2.1 in [37]. We defer the proof of Lemma D.2 and D.3 to Sec D.4 and D.5, respectively.

**Lemma D.3** (Error analysis for $c \notin P_M$). *When the center $c \leq p_{m+1}$ or $c \geq p_{n-m}$, $|\text{cost}^{(m)}(P,c) - \text{cost}^{(m)}(S,c)| \leq (\Gamma + \frac{\varepsilon n}{8}) \cdot \text{dist}(c, P_M) \leq \varepsilon \cdot \text{cost}^{(m)}(P,c)$.*

Now we are ready to prove the upper bound in Theorem 1.2.

*Proof of Theorem 1.2 (upper bound).* Given a dataset $P$ of size $n$, we apply Algorithm 1 to $P$ and obtain the output weighted set $S$. By Lemma 2.4, Algorithm 1 divides $P$ into $\tilde{O}(\varepsilon^{-\frac{1}{2}} + \frac{m}{n}\varepsilon^{-1})$ buckets. Note that $S$ contains only the mean point $\mu(B)$ of each bucket $B$. Thus we have $|S| = \tilde{O}(\varepsilon^{-\frac{1}{2}} + \frac{m}{n}\varepsilon^{-1})$. Combined with Lemma D.1 and D.3, we conclude that $S$ is an $\varepsilon$-coreset of $P$.

For the runtime, Line 3 cost $O(n)$ time to obtain the optimal center. Recall that $P_I^\star = \{p_i, \ldots, p_j\}$ is a continuous subsequence of $P$ and $c^\star = p_{\lfloor \frac{j+i}{2} \rfloor}$, since we only consider the one-dimensional case. Thus, we can compute $P_I^\star$ and $c^\star$ in $O(n)$ time, since we can sequentially replace the leftmost point in the current inliers with the next point from the outliers, and the resulting cost difference can be computed in $O(1)$ time. Lines 4-6 cost $O(n)$ time, since we can sequentially check whether the next point can be added to the current bucket, otherwise, we place it into a new bucket. Obviously Lines 7-8 also cost $O(n)$ time. This completes the proof. $\qquad\square$

## D.2 Proof of Lemma 2.4: Number of buckets

*Proof of Lemma 2.4.* Note that in Line 2, the number of buckets is $\tilde{O}(\varepsilon^{-\frac{1}{2}})$ in $S_M$ by Theorem 2.2. For Line 4, there are at most $O(\log(\frac{1}{\varepsilon}))$ non-empty blocks. For Line 6, the constraint on $N(B_{i,j})$ generates at most $O(\frac{m}{n}\varepsilon^{-1})$ buckets. For Lines 7-8, there are $O(1)$ new one-point buckets.

What remains is to show that each non-empty block contains $\tilde{O}(\varepsilon^{-\frac{1}{2}} + \frac{m}{n}\varepsilon^{-1})$ buckets in Line 5. Note that each block contains at most $m$ points, thus the constraint on $N(B_{i,j})$ generates at most $O(\frac{m}{n}\varepsilon^{-1})$ buckets. Thus we only have to consider the constraint on $\delta(B_{i,j})$.

Suppose we divide an inner block $B_i$ into $t$ buckets $\{B_{i,j}\}_{1 \leq j \leq t}$ due to controlling $\delta(B_{i,j})$. Since each $B_{ij}$ is the maximal bucket with $\delta(B_{i,j}) \leq \frac{2^i \cdot \varepsilon^2 n r_{\max}}{288}$, we have $\delta(B_{i,j} \cup B_{i+1,j}) > \frac{2^i \cdot \varepsilon^2 n r_{\max}}{288}$. Denote $B_{i,2j-1} \cup B_{i,2j}$ by $C_j$ for $j \in \{1, \ldots, \lfloor \frac{t}{2} \rfloor\}$. Let $len(B) := \max_{p \in B} p - \min_{p \in B} p$ be the length of $B$. Note that $\delta(B) \leq N(B) \cdot len(B)$ holds for every bucket $B$. Thus we have:

$$
\begin{aligned}
m2^i \varepsilon r_{\max} &\geq N(B_i) \cdot len(B_i) \qquad (N(B_i) < m, len(B_i) < 2^i \varepsilon r_{\max}) \\
&\geq \sum_{j=1}^{\lfloor \frac{t}{2} \rfloor} N(C_j) \sum_{j=1}^{\lfloor \frac{t}{2} \rfloor} |C_j| \\
&\geq (\sum_{j=1}^{\lfloor \frac{t}{2} \rfloor} N(C_j)^{\frac{1}{2}} |C_j|^{\frac{1}{2}})^2 \qquad \text{(Cauchy-Schwarz Inequality)} \\
&\geq (\sum_{j=1}^{\lfloor \frac{t}{2} \rfloor} \delta(C_j)^{\frac{1}{2}})^2 \\
&> (\lfloor \frac{t}{2} \rfloor)^2 \cdot \frac{2^i \cdot \varepsilon^2 n r_{\max}}{288}.
\end{aligned}
\tag{13}
$$

So we have $(\lfloor \frac{t}{2} \rfloor)^2 \cdot \frac{2^i \cdot \varepsilon^2 n r_{\max}}{288} < m2^i \varepsilon r_{\max}$, which implies $t \leq O(\varepsilon^{-\frac{1}{2}})$. The proof above is similar to Lemma 2.8 in [40]. Similarly, it is trivial to prove that $B_0$ also satisfies the above inequality. Since there are $O(\log(\varepsilon^{-1}))$ non-empty blocks, the constraint on $\delta(B_{i,j})$ generates at most $\tilde{O}(\varepsilon^{-\frac{1}{2}})$ buckets. Thus Lemma 2.4 holds. $\qquad\square$

## D.3 Proof of Lemma D.1: Error analysis for $c \in P_M$

*Proof of Lemma D.1.* Let $L_O := P_L \cap L^\star$, $R_O := P_R \cap L^\star$. Recall that $L^\star$ denote the set of outliers w.r.t. the optimal center $c^\star$. W.L.O.G, assume $c > c^\star$, thus $P_I^{(c)} \cap L_O = \emptyset$. Next, we analyze the induced error in two cases, based on the scale of $\text{dist}(c, c^\star)$. When $\varepsilon r_{\max} > \text{dist}(c, c^\star)$, the center

$c$ is sufficiently close to $c^\star$, so $\text{cost}^{(m)}(P,c) \approx \text{cost}^{(m)}(P,c^\star)$, resulting in a small error. When $\varepsilon r_{\max} \leq \text{dist}(c,c^\star)$, we have $\text{cost}^{(m)}(P,c) > \Omega(n \cdot \text{dist}(c,c^\star)) > \Omega(\varepsilon n r_{\max})$, which matches the error from any outlier-misaligned bucket $B$, whose error is at most $O(\varepsilon n)(\text{dist}(c,c^\star) + \varepsilon r_{\max})$ by Lemma 2.3.

**Case 1:** $\text{dist}(c,c^\star) > \frac{\varepsilon}{6} \cdot r_{\max}$.

If $P_I^{(c)} \cap R_O = \emptyset$, then we have $P_I^{(c)} = P_I^\star$, $S_I^{(c)} = S_I^\star$. In this case, we directly have $\text{cost}^{(m)}(S,c) \in (1 \pm \varepsilon)\text{cost}^{(m)}(P,c)$ by Theorem 2.2.

Next we assume $P_I^{(c)} \cap R_O \neq \emptyset$. Same as the above lemma, denote the leftmost and rightmost buckets intersecting $P_I^{(c)}$ as $B_L$, $B_R$, respectively. Recall that the coreset constructed by $\mathcal{A}(P_M, \frac{\varepsilon}{3})$ is $S_M$. By Theorem 2.2, Algorithm $\mathcal{A}$ ensures that

$$|\text{cost}(S_M,c) - \text{cost}(P_M,c)| \leq \frac{\varepsilon}{3} \cdot \text{cost}(P_M,c) < \frac{\varepsilon}{3} \cdot \text{cost}^{(m)}(P,c).$$

Next, we bound the cumulative error of $B_L$ and $B_R$. Define $\gamma := \max(0, \lceil \log(\frac{\text{dist}(c,c^\star)}{2\varepsilon \cdot r_{\max}}) \rceil)$. Obviously $\frac{\varepsilon}{6} r_{\max} \leq \text{dist}(c,c^\star) < r_{\max}$, thus $\gamma \in [0, \lceil \log(\varepsilon^{-1}) \rceil - 1]$. By the definition of $\gamma$, we have $\text{dist}(c,c^\star) \leq 2^{\gamma+1} \varepsilon r_{\max}$. Denote the rightmost point of $B_R$ as $p_r$, then it follows that

$$
\begin{aligned}
\text{dist}(p_r, c_R) &\leq 2\text{dist}(c,c^\star) & \text{(Lemma 2.3)} \\
&\leq 2^{\gamma+2} \varepsilon r_{\max}. & (\text{dist}(c,c^\star) \leq 2^{\gamma+1}\varepsilon r_{\max})
\end{aligned}
$$

Recall that block $B_i^{(R)} = \{ p \in P_R \mid p > c_R, 2^i \varepsilon r_{\max} \leq \text{dist}(p,c_R) < 2^{i+1} \varepsilon r_{\max} \}$. So the block $B_i$ which contains bucket $B_R$ satisfies $i \leq \gamma + 1$. By Line 5 in Algorithm 1, any bucket $B_{i,j}$ in inner block $B_i$ satisfies $\delta(B_{i,j}) \leq \frac{2^i \cdot \varepsilon^2 n r_{\max}}{288}$. Thus,

$$\delta(B_R) \leq \frac{2^{\gamma+1} \cdot \varepsilon^2 n r_{\max}}{288}.$$

Similarly,

$$\delta(B_L) \leq \frac{2^{\gamma+1} \cdot \varepsilon^2 n r_{\max}}{288}.$$

Note that at least $\lfloor \frac{n-m}{2} \rfloor$ points are on the left of $c^\star$, among which there are at least $(\lfloor \frac{n-m}{2} \rfloor - m)$ inliers w.r.t center $c$. Moreover, when $\gamma = 0$, we simply have $\text{dist}(c,c^\star) > \frac{2^\gamma \varepsilon r_{\max}}{6}$ by the assumption; when $\gamma > 0$, we have $\text{dist}(c,c^\star) > 2^\gamma \varepsilon r_{\max}$ by the definition of $\gamma$. Thus,

$$
\begin{aligned}
\text{cost}^{(m)}(P,c) &\geq (\lfloor \frac{n-m}{2} \rfloor - m) \cdot \text{dist}(c,c^\star) \\
&> \frac{n}{8} \cdot \text{dist}(c,c^\star) & (n \geq 4m) \qquad (14) \\
&> \frac{2^\gamma \varepsilon n r_{\max}}{48}. & (\text{dist}(c,c^\star) > \frac{2^\gamma \varepsilon r_{\max}}{6})
\end{aligned}
$$

Now we are ready to prove our goal $|\text{cost}^{(m)}(P,c) - \sum_{j \in [q]} (|B_j| - m_j) \cdot \text{dist}(\mu(B_j),c)| \leq \varepsilon \cdot \text{cost}^{(m)}(P,c)$. Recall that $m_j := |B_j \setminus P_I^{(c)}|$ for each bucket $B_j$. Thus for each bucket $B_j$ that is between $B_L$ and $B_R$, we have $m_j = 0$; for $B_L$ and $B_R$, we have $|B_L| - m_L = |B_L \cap P_I^{(c)}|$ and

$|B_R| - m_R = |B_R \cap P_I^{(c)}|$. Then we have

$$|\text{cost}^{(m)}(P, c) - \sum_{j \in [q]} (|B_j| - m_j) \cdot \text{dist}(\mu(B_j), c)|$$

$$\leq |\text{cost}(S_M, c) - \text{cost}(P_M, c)| + \sum_{p \in B_L \cap P_I^{(c)}} \text{dist}(p, \mu(B_L)) + \sum_{p \in B_R \cap P_I^{(c)}} \text{dist}(p, \mu(B_R))$$

$$\text{(Triangle Inequality)}$$

$$\leq |\text{cost}(S_M, c) - \text{cost}(P_M, c)| + \delta(B_L) + \delta(B_R) \qquad \text{(Definition of } \delta(B))$$

$$< \frac{\varepsilon}{3} \cdot \text{cost}^{(m)}(P, c) + \frac{2^\gamma \cdot \varepsilon^2 n r_{\max}}{72}$$

$$\leq \varepsilon \cdot \text{cost}^{(m)}(P, c). \qquad \text{(Inequality (14))}$$

Thus by the definition of $S_I^{(c)}$,

$$\text{cost}^{(m)}(S, c) \leq \sum_{j \in [q]} (|B_j| - m_j) \cdot \text{dist}(\mu(B_j), c) < (1 + \varepsilon)\text{cost}^{(m)}(P, c).$$

Moreover, it is easy to prove that the rightmost block $B_{i'}$ intersecting with $S_I^{(c)}$ still satisfies $i' \leq \gamma + 1$. Thus, similarly to the previous discussion, we have:

$$\text{cost}^{(m)}(P, c) < (1 + \varepsilon)\text{cost}^{(m)}(S, c).$$

Thus $\text{cost}^{(m)}(S, c) \in (1 \pm \varepsilon)\text{cost}^{(m)}(P, c)$.

**Case 2:** $\text{dist}(c, c^\star) \leq \frac{\varepsilon}{6} \cdot r_{\max}$.

Let $w_l := \sum_{p \in P_I^{(c)} \setminus P_I^\star} w(p)$. Consider the points in $P_I^{(c)} \setminus P_I^\star$, we have:

$$\text{cost}^{(m)}(P, c) > w_l(r_{\max} - \text{dist}(c, c^\star)) \geq (1 - \frac{\varepsilon}{6})w_l r_{\max} > \frac{5}{6}w_l r_{\max}. \qquad (15)$$

Thus,

$$\begin{aligned}
\text{cost}^{(m)}(P, c) &= \text{cost}(P_I^{(c)} \cap P_I^\star, c) + \text{cost}(P_I^{(c)} \setminus P_I^\star, c) \\
&\geq \text{cost}(P_I^\star, c) - 2w_l \cdot \text{dist}(c, c^\star) \qquad \text{(Definition of } w_l) \\
&\geq \text{cost}(P_I^\star, c) - \frac{\varepsilon}{3}w_l r_{\max} \qquad\qquad\qquad (16) \\
&> \text{cost}(P_I^\star, c) - \frac{2\varepsilon}{5} \cdot \text{cost}^{(m)}(P, c) \qquad \text{(Inequality (15))}.
\end{aligned}$$

By the definition of $\text{cost}^{(m)}(P, c)$, we have

$$\text{cost}^{(m)}(P, c) \leq \text{cost}(P_I^\star, c). \qquad (17)$$

Similarly, we have

$$\text{cost}(S_I^\star, c) - \frac{2\varepsilon}{5} \cdot \text{cost}^{(m)}(S, c) \leq \text{cost}^{(m)}(S, c) < \text{cost}(S_I^\star, c). \qquad (18)$$

By Theorem 2.2, Algorithm $\mathcal{A}$ ensures that $|\text{cost}(S_M, c) - \text{cost}(P_M, c)| < \frac{\varepsilon}{3} \cdot \text{cost}(P_M, c)$. According to the definition of the block $B_0$, we can obtain that there is no bucket that partially intersects with $P_I^\star$. Thus,

$$\begin{aligned}
|\text{cost}(S_I^\star, c) - \text{cost}(P_I^\star, c)| &= |\text{cost}(S_M, c) - \text{cost}(P_M, c)| \\
&< \frac{\varepsilon}{3} \cdot \text{cost}(P_M, c) \qquad\qquad (19) \\
&< \frac{\varepsilon}{3} \cdot \text{cost}(P_I^\star, c).
\end{aligned}$$

Combining the above equations, we have

$$\mathrm{cost}^{(m)}(S,c) < \mathrm{cost}(S_I^\star,c) \qquad\qquad \text{(Inequality (18))}$$
$$< (1+\frac{\varepsilon}{3})\mathrm{cost}(P_I^\star,c) \qquad\qquad \text{(Inequality (19))}$$
$$< (1+\frac{\varepsilon}{3})(1+\frac{2\varepsilon}{5})\mathrm{cost}^{(m)}(P,c) \qquad\qquad \text{(Inequality (16))}$$
$$< (1+\varepsilon)\mathrm{cost}^{(m)}(P,c).$$

Similarly we have $\mathrm{cost}^{(m)}(P,c) < (1+\varepsilon)\mathrm{cost}^{(m)}(S,c)$, thus $\mathrm{cost}^{(m)}(S,c) \in (1\pm\varepsilon)\cdot\mathrm{cost}^{(m)}(P,c)$. $\qquad\square$

## D.4 Proof of Lemma D.2: Number of misaligned outliers

*Proof of Lemma D.2.* Note that for any center $c$, $P_I^{(c)}$ is a continuous subset of size $n-m$. Let $P_I^{(c)} = \{p_l, \ldots, p_{l+n-m-1}\}$. Assume $p_l \in B_L$ and $p_{l+n-m-1} \in B_R$. Consider a point $p_j \in P_L$ that satisfies $\mathrm{dist}(p_j,c) < \mathrm{dist}(p_{j+n-m},c)$ and $\mathrm{dist}(\mu(B),c) > \mathrm{dist}(\mu(B'),c)$, where $p_j \in B$ and $p_{j+n-m} \in B'$. Next we show that the total number of points satisfying the above conditions is at most $O(\varepsilon n)$, which also provides an upper bound for $\sum_i |m_i - m_i'|$.

In this case $p_j \in P_I^{(c)}$. By the definition of $P_I^{(c)}$, we have $j \geq l$. If $B \neq B_L$ and $B' \neq B_R$, then by $j \geq l$, we have $\mu(B) > \max(B_L)$ and $\mu(B') > \max(B_R)$. Then we have

$$\mathrm{dist}(p_j,c) \geq \mathrm{dist}(\max(B_L),c) > \mathrm{dist}(\mu(B),c) > \mathrm{dist}(\mu(B'),c) > \mathrm{dist}(\max(B_R),c)$$
$$\geq \mathrm{dist}(p_{j+n-m},c),$$

which contradicts the inequality $\mathrm{dist}(p_j,c) < \mathrm{dist}(p_{j+n-m},c)$. Thus, either $B = B_L$ or $B' = B_R$ holds. This indicates that the total number of points satisfying the above conditions is at most $|B_L| + |B_R|$. Formally, denote $B^{(i)}$ as the bucket containing $p_i$, we have

$$\mathbb{I}_1 = \sum_{j\in[m]} \left| \mathbb{I}(p_j \in P_I^{(c)}, \mathrm{dist}(\mu(B^{(j)}),c) > \mathrm{dist}(\mu(B^{(j+n-m)}),c)) \right| \leq |B_L| + |B_R|. \qquad (20)$$

Symmetrically, consider the points in $P_R$, we have

$$\sum_{j\in[n-m+1,n]} \left| \mathbb{I}(p_j \in P_I^{(c)}, \mathrm{dist}(\mu(B^{(j-n+m)}),c) < \mathrm{dist}(\mu(B^{(j)}),c)) \right| \leq |B_L| + |B_R|. \qquad (21)$$

Since $|P_I^{(c)}| = n-m$, for any $i \in [1,m]$, exactly one point in $\{p_i, p_{i+n-m}\}$ is in $P_I^{(c)}$. This means that $p_i \in P_I^{(c)}$ if and only if $p_{i+n-m} \notin P_I^{(c)}$. Thus, Inequality (21) is equivalent to the following form:

$$\mathbb{I}_2 = \sum_{j\in[m]} \left| \mathbb{I}(p_j \notin P_I^{(c)}, \mathrm{dist}(\mu(B^{(j)}),c) < \mathrm{dist}(\mu(B^{(j+n-m)}),c)) \right| \leq |B_L| + |B_R|. \qquad (22)$$

Moreover, we'll show that $\mathbb{I}_1$ and $\mathbb{I}_2$ cannot both be greater than 0. Suppose $\mathbb{I}_1 > 0$. In this case, there exists a point $p_j \in P_L \cap P_I^{(c)}$ such that $\mathrm{dist}(\mu(B^{(j)}),c) > \mathrm{dist}(\mu(B^{(j+n-m)}),c)$. Consider a point $p_{j'} \in P_L \setminus P_I^{(c)}$, obviously $j' < j$. Then we have

$$\mathrm{dist}(\mu(B^{(j')}),c) \geq \mathrm{dist}(\mu(B^{(j)}),c) > \mathrm{dist}(\mu(B^{(j+n-m)}),c) \geq \mathrm{dist}(\mu(B^{(j'+n-m)}),c).$$

Thus $\mathbb{I}_2 = 0$. Conversely, it still holds due to symmetry. Based on the above analysis, we have

$$\mathbb{I}_1 + \mathbb{I}_2 \leq |B_L| + |B_R|. \qquad (23)$$

By $p_i \in P_I^{(c)} \iff p_{i+n-m} \notin P_I^{(c)}$, it is easy to prove that $\sum_i |m_i - m_i'|$ in $P_L$ and $P_R$ are exactly the same. Then it follows that

$$\sum_{i \in [q]} |m_i - m_i'| = 2 \sum_{i \in [q], B_i \subseteq P_L} |m_i - m_i'|$$

$$= 2 \sum_{i \in [q], B_i \subseteq P_L} \left| \sum_{p_j \in B_i} \mathbb{I}(p_j \in P_I^{(c)}) - \mathbb{I}(\text{dist}(\mu(B_i), c) < \text{dist}(\mu(B^{(j+n-m)}), c)) \right|$$

$$\leq 2 \sum_{j \in [m]} \left| \mathbb{I}(p_j \in P_I^{(c)}) - \mathbb{I}(\text{dist}(\mu(B^{(j)}), c) < \text{dist}(\mu(B^{(j+n-m)}), c)) \right|$$

$$\leq 2(\mathbb{I}_1 + \mathbb{I}_2)$$
$$\leq 2(|B_L| + |B_R|) \qquad\qquad (\text{Inequality (23)})$$
$$\leq \frac{\varepsilon n}{4}. \qquad\qquad (|B| \leq \frac{\varepsilon n}{16})$$

$\square$

## D.5  Proof of Lemma D.3: Error analysis for $c \notin P_M$

*Proof of Lemma D.3.* W.L.O.G., we assume that $c \geq p_{n-m}$. Recall that $P$ is divided into buckets $B_1, \ldots, B_q$, from left to right. Suppose $B_t$ ($t \in [q]$) contains the center $c$, i.e., $\min_{p \in B_t} p \leq c \leq \max_{p \in B_t} p$. We define function $f_P(c) = \text{cost}^{(m)}(P, c)$ and $f_S(c) = \text{cost}^{(m)}(S, c)$ for every $c \in \mathbb{R}$.

Note that $f_P(c) = f_P(p_{n-m}) + \int_{p_{n-m}}^{c} f_P'(x)$ and $f_S(c) = f_S(p_{n-m}) + \int_{p_{n-m}}^{c} f_S'(x)$ holds for any $c > p_{n-m}$. Next, we first show that $f_P(p_{n-m}) = f_S(p_{n-m})$ by Line 8 in Algorithm 1. Then we verify that $|f_P'(c) - f_S'(c)|$ is bounded by $\sum_{i \in [q]} |m_i - m_i'|$, which suffices to prove the lemma. The main idea is similar to that of the proof of Theorem 2.1 in [37].

By Line 8 in Algorithm 1, there is no bucket that partially intersect with $P_I^{(p_{n-m})}$. In this case, for each bucket $B \subset P_I^{(p_{n-m})}$, $\text{dist}(\mu(B), c) \leq \max_{p \in P_I^{(p_{n-m})}} \text{dist}(p, c)$; for each bucket $B \not\subset P_I^{(p_{n-m})}$, $\text{dist}(\mu(B), c) > \max_{p \in P_I^{(p_{n-m})}} \text{dist}(p, c)$. This indicates that $S_I^{(p_{n-m})}$ contains $\mu(B)$ with weight $|B|$ for each $B \subset P_I^{(p_{n-m})}$, which maintains consistent weights with $P_I^{(p_{n-m})}$. Then by Lemma B.1, we have $f_P(p_{n-m}) = f_S(p_{n-m})$. Note that $f_p(c)$ is a linear function of $c$, and we have

$$f_P'(c) = |\{p \mid p \in P_I^{(c)}, p \leq c\}| - |\{p \mid p \in P_I^{(c)}, p > c\}|$$
$$= \sum_{i<t}(|B_i| - m_i) + |B_t| \cap (-\infty, c]| - |B_t \cap (c, \infty)| - \sum_{i>t}(|B_i| - m_i).$$

Similarly, when $\mu(B_t) \leq c$, we have $f_S'(c) = \sum_{i<t}(|B_i| - m_i') + |B_t| - \sum_{i>t}(|B_i| - m_i')$; when $\mu(B_t) > c$, we have $f_S'(c) = \sum_{i<t}(|B_i| - m_i') - |B_t| - \sum_{i>t}(|B_i| - m_i')$. Thus

$$|f_P'(c) - f_S'(c)| \leq \sum_{i \in [q]} |m_i - m_i'| + 2|B_t|$$
$$\leq \Gamma + 2|B_t| \qquad\qquad (\text{Definition of } \Gamma)$$
$$\leq \Gamma + \frac{\varepsilon n}{8}. \qquad\qquad (|B| \leq \frac{\varepsilon n}{16})$$

$|T_0| = n - q\lfloor\frac{m}{q}\rfloor$ $|T_1| = \lfloor\frac{m}{q}\rfloor$ $|T_2| = \lfloor\frac{m}{q}\rfloor$ $\ldots\ldots$ $|T_q| = \lfloor\frac{m}{q}\rfloor$

$0$ $\quad$ $m^\alpha$ $\quad\quad$ $m^{2\alpha}$ $\quad\quad\quad\quad$ $m^{q\alpha}$

Figure 3: A case for demonstrating the coreset lower bound for robust 1D geometric median. $T_i$ contains $\lfloor\frac{m}{q}\rfloor$ points where each point $p \in T_i$ satisfies $p = m^{i\alpha}$. $T_0$ contains the remaining points where each point $p \in T_0$ satisfies $p = 0$.

Then by $f_P(c) = f_P(p_{n-m}) + \int_{p_{n-m}}^c f'_P(x)\,dx$ and $f_S(c) = f_S(p_{n-m}) + \int_{p_{n-m}}^c f'_S(x)\,dx$, we have

$$|f_P(c) - f_S(c)| = \left| f_P(p_{n-m}) + \int_{p_{n-m}}^c f'_P(x)\,dx - f_S(p_{n-m}) - \int_{p_{n-m}}^c f'_S(x)\,dx \right|$$

$$= \left| \int_{p_{n-m}}^c f'_P(x) - f'_S(x)\,dx \right| \qquad (f_P(p_{n-m}) = f_S(p_{n-m}))$$

$$\leq \int_{p_{n-m}}^c |f'_P(x) - f'_S(x)|\,dx$$

$$\leq \int_{p_{n-m}}^c (\Gamma + \frac{\varepsilon n}{8})\,dx$$

$$= (c - p_{n-m})(\frac{\varepsilon n}{8} + \Gamma)$$

$$= (\frac{\varepsilon n}{8} + \Gamma) \cdot \mathrm{dist}(c, P_M). \qquad (\text{Definition of } P_M)$$

Moreover, since $|P_R| = m$, at least $n - 2m$ inliers w.r.t. $c$ are on the left of $p_{n-m}$. These points satisfy that $\mathrm{dist}(p, c) \geq \mathrm{dist}(p_{n-m}, c)$, which implies that $\mathrm{cost}^{(m)}(P, c) \geq (n-2m)\cdot\mathrm{dist}(p_{n-m}, c)$. Then we have

$$|f_P(c) - f_S(c)| \leq (c - p_{n-m})(\frac{\varepsilon n}{8} + \Gamma)$$

$$\leq (\frac{3\varepsilon n}{8})(c - p_{n-m}) \qquad (\text{Lemma } D.2)$$

$$< \varepsilon(n - 2m)(c - p_{n-m}) \qquad (n \geq 4m)$$

$$\leq \varepsilon \cdot \mathrm{cost}^{(m)}(P, c).$$

This completes the proof. $\qquad\qquad\square$

### D.6 Proof of the lower bound in Theorem 1.2

Next we show that for $n \geq 4m$, the size lower bound of $\varepsilon$-coreset for robust 1D geometric median is $\Omega(\varepsilon^{-\frac{1}{2}} + \frac{m}{n}\varepsilon^{-1})$. This lower bound matches the upper bound in the above discussion, which completes the proof of Theorem 1.2. In the following discussion, we assume that the size of dataset $P$ is sufficiently large such that $\varepsilon n > 1$, which holds in nearly all practical scenarios.

*Proof of Theorem 1.2 (lower bound).* For vanilla 1D geometric median, [26] shows that the size lower bound of $\varepsilon$-coreset is $\Omega(\varepsilon^{-\frac{1}{2}})$, which is obviously also the coreset size lower bound for robust 1D geometric median. Thus, it remains to show the coreset size is $\Omega(\frac{m}{n}\varepsilon^{-1})$ when $\frac{m}{n} > \varepsilon^{\frac{1}{2}}$.

We first construct the dataset $P$ of size $n$. Let $q = \lfloor\frac{m}{2n\varepsilon}\rfloor$. The dataset $P$ is a union of $1+q$ disjoint sets $\{T_0, T_1, \ldots, T_q\}$. For each $i \in \{1, \ldots, q\}$, $T_i$ contains $\lfloor\frac{m}{q}\rfloor$ points, and every point $p \in T_i$ satisfies $p = m^{i\alpha}$, where $\alpha = 2 + \log_m(\varepsilon^{-2})$. $T_0$ contains $n - q\lfloor\frac{m}{q}\rfloor$ points where each point $p \in T_0$ satisfies $p = 0$. Correspondingly, define $q$ disjoint intervals $\{I_1, \ldots, I_q\}$, where $I_i = [m^{(i-1)\alpha+1}, m^{i\alpha+1}]$.

Suppose $S$ is a $\varepsilon$-coreset of $P$ with size $|S| < q$. Then by the pigeonhole principle, there exists an interval $I_j$ ($j \in [1, q]$) such that $S$ does not include any points located in $I_j$. This implies that for any point $p \in S$, we have $p \leq m^{(j-1)\alpha+1}$ or $p \geq m^{j\alpha+1}$. Fix the center $c = m^{j\alpha}$, then we have

$$
\begin{aligned}
\text{cost}^{(m)}(S, c) &\geq (n - m) \cdot \min_{p \in S} \text{dist}(p, c) \\
&\geq (n - m) \cdot (m^{j\alpha} - m^{(j-1)\alpha+1}) && (S \cap I_j = \emptyset) \\
&> [(1 - m^{1-\alpha})n - m]m^{j\alpha} \\
&> [(1 - \varepsilon^2)n - m]m^{j\alpha} && (\alpha = 2 + \log_m(\varepsilon^{-2})) \\
&> [(1 - \varepsilon - 2\varepsilon^2)n + \varepsilon n - m]m^{j\alpha} \\
&> (1 + \varepsilon)[(1 - 2\varepsilon)n - (m - 1)]m^{j\alpha}. && (\varepsilon > \tfrac{1}{n})
\end{aligned}
$$

Since $n \geq 4m$, we have $|T_0| = n - q \cdot \lfloor \frac{m}{q} \rfloor \geq n - m$. Consider the points in $T_0$ and $T_j$ as inliers, we have

$$
\begin{aligned}
\text{cost}^{(m)}(P, c) &\leq \text{cost}(T_j, c) + (n - m - |T_j|)m^{j\alpha} \\
&= (n - m - \lfloor \frac{m}{q} \rfloor)m^{j\alpha} && (\text{Definition of } T_j) \\
&\leq (n - m - \frac{m}{q} + 1)m^{j\alpha} \\
&\leq [(1 - 2\varepsilon)n - (m - 1)]m^{j\alpha}. && (\text{Definition of } q)
\end{aligned}
$$

It follows that $\text{cost}^{(m)}(S, c) > (1 + \varepsilon)\text{cost}^{(m)}(P, c)$, thus $S$ is not a $\varepsilon$-coreset of $P$, which leads to a contradiction. This implies that any $\varepsilon$-coreset of $P$ contains $\Omega(\frac{m}{n}\varepsilon^{-1})$ points when $\frac{m}{n} > \varepsilon^{\frac{1}{2}}$. Considering the discussion above, the lower bound of the coreset size is $\Omega(\varepsilon^{-\frac{1}{2}} + \frac{m}{n}\varepsilon^{-1})$.

$\square$

Note that the dataset $P$ we construct is a multiset, which is slightly different from the definition. However, this does not affect the proof, because we can move each point by a sufficiently small and distinct distance, making the cost value almost unchanged.

When $\varepsilon n < 1$, we can show that the coreset size is $\Omega(\varepsilon^{-\frac{1}{2}} + m)$ by the above discussion. Moreover, consider a trivial method that applies algorithm $\mathcal{A}$ on $P_M$ and keeps all points not in $P_M$. It's easy to prove that this method constructs an $\varepsilon$-coreset of the original dataset under the assumption. In this case, the coreset size is $\tilde{O}(\varepsilon^{-\frac{1}{2}} + m)$, which matches the above lower bound.

# E  Proofs of Theorem 1.3 $(d \geq 1)$ and extension to metric spaces

In this section, we list out the missing proof in Section 3 and show how to extend Theorem 1.3 to various metric spaces.

## E.1  Proof of Lemma 3.2: induced error of $S_O$

Below we briefly introduce the proof idea. We first observe that the induced error of $S_O$ is primarily caused by points that act as inliers in $P$ but outliers in $S$, or vice versa, as shown in Lemmas E.3 and E.4. The error from a single point is bounded by $O(\frac{\text{cost}^{(m)}(P,c)}{m})$ (see Lemma E.5). The next task is ensuring that there are $O(\varepsilon m)$ such points in $S_O$, which is guaranteed when $S_O$ provides an $\varepsilon$-approximation for the ball range space on $L^\star$ (Lemma E.6). To achieve this, we sample $\tilde{O}(d/\varepsilon^2)$ points for $S_O$ (Lemma E.2).

Fix a center $c \in \mathbb{R}^d$. Let $L^{(c)} := \arg\min_{L \subseteq P: |L| = m} \text{cost}(P - L, c)$ be the set of outliers of $P$ w.r.t. $c$ and $m_P := |L^\star \cap L^{(c)}|$ represent the number of these outliers contained in $L^\star$. For $S_O \cup P_I^\star$, we first define a family of weight functions $\mathcal{W} := \{w_S : S_O \cup P_I^\star \to \mathbb{R}^+ \mid w_S(p) \leq w(p), \forall p \in S_O; w_S(p) \leq 1, \forall p \in P_I^\star; \|w_S\|_1 = n - m\}$. Intuitively, $\mathcal{W}$ represents the collection of all possible

weight functions for $n - m$ inliers of the weighted dataset $S_O \cup P_I^\star$. Define a weight function $w'$ as follows:

$$w' := \arg \min_{w_S \in \mathcal{W}} \sum_{p \in S_O \cup P_I^\star} w_S(p) \cdot \text{dist}(p, c),$$

i.e., $(S_O \cup P_I^\star, w')$ represents the $n - m$ inliers of $S_O \cup P_I^\star$ with respect to center $c$. Let $m_S := \sum_{p \in S_O}(w(p) - w'(p))$ denote the number of outliers of $S_O \cup P_I^\star$ w.r.t. $c$ that are contained in $S_O$.

For preparation, we introduce the concept of ball range space, which facilitates a precise analysis of point distributions in $P$ and $S$.

**Definition E.1** (Approximation of ball range space, Definition F.2 in [39]). For a given dataset $P \subset \mathbb{R}^d$, the ball range space on $P$ is $(P, \mathcal{P})$ where $\mathcal{P} := \{P \cap \text{Ball}(c, u) \mid c \in \mathbb{R}^d, u > 0\}$ and $\text{Ball}(c, u) := \{p \in \mathbb{R}^d \mid \text{dist}(p, c) \leq u\}$. A subset $Y \subset P$ is called an $\varepsilon$-approximation of the ball range space $(P, \mathcal{P})$ if for every $c, u \in \mathbb{R}^d$,

$$\left| \frac{|P \cap \text{Ball}(c, u)|}{|P|} - \frac{|Y \cap \text{Ball}(c, u)|}{|Y|} \right| \leq \varepsilon.$$

Based on this definition, we have the following preparation lemma that measures the performance of $S_O$; which is refined from [39].

**Lemma E.2** (Refined from Lemma F.3 of [39]). *Given dataset $P_O \subset \mathbb{R}^d$. Let $S_O$ be a uniform sampling of size $\tilde{O}(\frac{d}{\varepsilon^2})$ from $P_O$, then with probability at least $1 - \frac{1}{poly(1/\varepsilon)}$, $S_O$ is an $\varepsilon$-approximation of the ball range space on $P_O$. Define a weight function $w$: $w(p) = \frac{|P_O|}{|S_O|}$, for any $p \in S_O$. Then for any $c \in \mathbb{R}^d$, $u \in \mathbb{R}^+$,*

$$||P_O \cap \text{Ball}(c, u)| - w(S_O \cap \text{Ball}(c, u))| \leq \varepsilon |P_O|.$$

By the iterative method introduced by [48], the factor $O(d)$ of coreset size can be replaced by $\tilde{O}(\varepsilon^{-2})$. Therefore, $S_O$ is an $\varepsilon$-approximation of the ball range space on $L^\star$.

Then we are ready to prove Lemma 3.2. We first analyze where the induced error comes from.

Recall that we fix a center $c$ in this section. Let $T_O := \min_{p \in L^\star} \text{dist}(p, c)$ denote the minimum distance from points in $L^\star$ to $c$. Let $T_I := \max_{p \in P_I^\star} \text{dist}(p, c)$ denote the maximum distance from points in $P_I^\star$ to $c$. We have the following Lemma.

**Lemma E.3** (Comparing $T_O$ and $T_I$). *When $m_P = m$, $|\text{cost}^{(m)}(P, c) - \text{cost}^{(m)}(S_O \cup P_I^\star, c)| = 0$ holds. When $m_P < m$, $T_O \leq T_I$ holds.*

*Proof.* If $m_P = m$, for any point $p \in L^\star$, we have $\text{dist}(p, c) \geq \max_{p \in P_I^\star} \text{dist}(p, c)$. Since $S_O$ is sampled from $L^\star$, we know that for any point $p \in S_O$, $\text{dist}(p, c) \geq \max_{p \in P_I^\star} \text{dist}(p, c)$. Therefore, we have $m_P = m_S = m$ and then $\text{cost}^{(m)}(P, c) = \text{cost}(P_I^\star, c) = \text{cost}^{(m)}(S_O \cup P_I^\star, c)$.

If $m_P < m$, then there exists a point $\hat{p} \in P_I^\star$ such that $\text{dist}(\hat{p}, c) \geq T_O$. By definition, we also know that $\text{dist}(\hat{p}, c) \leq T_I$. Therefore, combining these two conditions, we have $T_O \leq T_I$. □

It remains to analyze the case that $m_P < m$. In this setting, we present the following lemma, which provides an upper bound on the estimation error $|\text{cost}^{(m)}(P, c) - \text{cost}^{(m)}(S_O \cup P_I^\star, c)|$. Before stating the lemma, we introduce a notation that will also be used in subsequent lemmas. We sort all distances $\text{dist}(p, c)$ for each point $p \in P_I^\star$ in descending order, w.l.o.g. say $\text{dist}(p_1, c) > \ldots > \text{dist}(p_{|P_I^\star|}, c)$. Here, we can safely assume all distances $\text{dist}(p, c)$ are distinct, given that adding small values to the distances has only a subtle impact on the cost. Let $d_i := \text{dist}(p_i, c)$ for $i \in [m]$. Then $d_1, \ldots, d_m$ represent the distances from the $m$ furthest points in $P_I^\star$ to the center $c$. Now, we are ready to provide the following lemmas.

**Lemma E.4** (An upper bound of the induced error of $S_O$). *When $m_P < m$, suppose $||P_O \cap \text{Ball}(c, u)| - w(S_O \cap \text{Ball}(c, u))| \leq \Delta$ for any $u > 0$, the following holds: $|\text{cost}^{(m)}(P, c) - \text{cost}^{(m)}(S_O \cup P_I^\star, c)| \leq 2 \cdot (\Delta + |m_P - m_S|) \cdot (T_I - T_O)$.*

*Proof.* By Fact F.1 in [39], we have

$$\text{cost}^{(m_P)}(L^\star, c) = \int_0^\infty (m - m_P - |L^\star \cap \text{Ball}(c, u)|)^+ du \tag{24}$$

and

$$\text{cost}^{(m_S)}(S_O, c) = \int_0^\infty (m - m_S - w(S_O \cap \text{Ball}(c, u)))^+ du \tag{25}$$

where $(a)^+ = \max\{a, 0\}$.

Let $T := \max_{p \in L^\star - L^{(c)}} \text{dist}(p, c)$ and $T_S := \max_{p \in S_O : w'(p) > 0} \text{dist}(p, c)$. Then we know that $T_O \leq T$, $T_S \leq T_I$. By definition, for any distance $u > T$, we have $|L^\star \cap \text{Ball}(c, u)| \geq m - m_P$. Then we can transform the Inequality (24) to

$$\text{cost}^{(m_P)}(L^\star, c) = \int_0^T (m - m_P - |L^\star \cap \text{Ball}(c, u)|) du. \tag{26}$$

Similarly, for any distance $u > T_S$, we have $w(S_O \cap \text{Ball}(c, u)) \geq m - m_S$. We can transform the Inequality (25) to

$$\text{cost}^{(m_S)}(S_O, c) = \int_0^{T_S} (m - m_S - w(S_O \cap \text{Ball}(c, u))) du. \tag{27}$$

Based on the notation of $d_1, ..., d_m$, we know that each point $p \in P_I^\star$ satisfying $\text{dist}(p, c) \leq d_{m - m_P + 1}$ is an inlier of $P_I^\star \cup L^\star$, and each point $q \in P_I^\star$ with weight $w'(q) > 0$ satisfying $\text{dist}(q, c) \leq d_{m - \lfloor m_S \rfloor}$ is an inlier of $S_O \cup P_I^\star$. Let $m_S' := \lfloor m_S \rfloor$. Let $l := |m_P - m_S|$ denote the difference in the number of outliers. If $m_P > m_S$, we have

$$l \cdot d_{m - m_S'} \leq \text{cost}^{(m - m_P)}(P_I^\star, c) - \text{cost}^{(m - m_S)}(P_I^\star, c) \tag{28}$$

and

$$\text{cost}^{(m - m_P)}(P_I^\star, c) - \text{cost}^{(m - m_S)}(P_I^\star, c) \leq l \cdot d_{m - m_P + 1}. \tag{29}$$

If $m_P < m_S$, we have

$$l \cdot d_{m - m_P} \leq \text{cost}^{(m - m_S)}(P_I^\star, c) - \text{cost}^{(m - m_P)}(P_I^\star, c) \tag{30}$$

and

$$\text{cost}^{(m - m_S)}(P_I^\star, c) - \text{cost}^{(m - m_P)}(P_I^\star, c) \leq l \cdot d_{m - m_S'}. \tag{31}$$

If $m_P = m_S$, we have

$$\text{cost}^{(m - m_S)}(P_I^\star, c) - \text{cost}^{(m - m_P)}(P_I^\star, c) = 0. \tag{32}$$

By definition, we know that

$$d_m \leq d_{m-1} \leq \ldots \leq d_1 \leq T_I. \tag{33}$$

When $m_S \neq m$, we know the point $p \in P_I^\star$ that has distance $\text{dist}(p, c) = d_{m - m_S'}$ is an outlier on $P_I^\star \cup S_O$. Then there exists a point $q \in S_O$ such that $\text{dist}(q, c) \leq \text{dist}(p, c)$. Then we have

$$d_{m - m_S'} \geq T_O. \tag{34}$$

Similarly, the point $p \in P_I^\star$ with distance $\text{dist}(p, c) = d_{m - m_P}$ is an outlier on $P$, thus there exists a point $q \in L^\star$ such that $\text{dist}(q, c) \leq \text{dist}(p, c)$. Then we have

$$d_{m - m_P} \geq T_O. \tag{35}$$

By Lemma E.3, it suffices to discuss the following three cases based on the values of $m_P$ and $m_S$.

**Case 1:** $m_S = m$

Recall that $l = m_S - m_P$, we have

$$
\begin{aligned}
\text{cost}^{(m_P)}(L^\star, c) &= \int_0^T (m - m_P - |L^\star \cap \text{Ball}(c, u)|) du && \text{(Inequality (26))} \\
&= \int_0^T (l - |L^\star \cap \text{Ball}(c, u)|) du && (l = m_S - m_P) \\
&= l \cdot T - \int_{T_O}^T |L^\star \cap \text{Ball}(c, u)| du && (\forall p \in L^\star, \text{dist}(p, c) \geq T_O).
\end{aligned}
$$

Since $m_S = m$ and $m_P < m_S$, each point $p \in L^\star - L^{(c)}$ satisfies $\text{dist}(p, c) < \min_{q \in S_O} \text{dist}(q, c)$. For each $T_O \leq u \leq T$, we have $w(S_O \cap \text{Ball}(c, u)) = 0$. Since $|L^\star \cap \text{Ball}(c, u)| - w(S_O \cap \text{Ball}(c, u)) \leq \Delta$, we know $|L^\star \cap \text{Ball}(c, u)| \leq \Delta$ for $T_O \leq u \leq T$ and

$$l \cdot T - (T - T_O) \cdot \Delta \leq \text{cost}^{(m_P)}(L^\star, c) \leq l \cdot T.$$

Since $\text{cost}^{(m_S)}(S_O, c) = 0$, we have

$$\text{cost}^{(m_P)}(L^\star, c) - \text{cost}^{(m_S)}(S_O, c) \leq l \cdot T \tag{36}$$

and

$$\text{cost}^{(m_S)}(S_O, c) - \text{cost}^{(m_P)}(L^\star, c) \leq (T - T_O) \cdot \Delta - l \cdot T. \tag{37}$$

Adding Inequality (30) and Inequality (36), we have

$$
\begin{aligned}
\text{cost}^{(m)}(P, c) - \text{cost}^{(m)}(S_O \cup P_I^\star, c) \leq \quad & l \cdot T_I - l \cdot T_O && (T \leq T_I \text{ and Inequality (35)}) \\
\leq \quad & |m_P - m_S| \cdot (T_I - T_O) && (\text{Lemma E.6}).
\end{aligned}
$$

Adding Inequality (31) into Inequality (37), we have

$$
\begin{aligned}
\text{cost}^{(m)}(S_O \cup P_I^\star, c) - \text{cost}^{(m)}(P, c) \leq \quad & (T - T_O) \cdot \Delta + l \cdot (d_1 - T) \\
\leq \quad & (T_I - T_O) \cdot \Delta + l \cdot (T_I - T_O) \\
& (T_O \leq T \leq T_I \text{ and Inequality (33)}) \\
\leq \quad & (|m_P - m_S| + \Delta) \cdot (T_I - T_O).
\end{aligned}
$$

Then we complete the proof of Case 2.

**Case 2:** $m_S \neq m_P < m$

Without loss of generality, we assume that $m_P > m_S$ and $T \geq T_S$.

Firstly, we prove that $\text{cost}^{(m_P)}(P, c) - \text{cost}^{(m_S)}(S_O \cup P_I^\star, c) \leq 2(\Delta + |m_P - m_S|) \cdot (T_I - T_O)$. We have

$$
\begin{aligned}
& \text{cost}^{(m_P)}(L^\star, c) - \text{cost}^{(m_S)}(S_O, c) \\
&= \quad \int_0^T (m - m_P - |L^\star \cap \text{Ball}(c, u)|) du \\
&\quad - \int_0^{T_S} (m - m_S - w(S_O \cap \text{Ball}(c, u))) du && (\text{Inequalities (26) and (27)}) \\
&= \quad \int_0^{T_S} (m_S - m_P + w(S_O \cap \text{Ball}(c, u)) - |L^\star \cap \text{Ball}(c, u)|) du \\
&\quad + \int_{T_S}^T (m - m_P - |L^\star \cap \text{Ball}(c, u)|) du && (T \geq T_S) \\
&= \quad \int_{T_O}^{T_S} (w(S_O \cap \text{Ball}(c, u)) - |L^\star \cap \text{Ball}(c, u)|) du - l \cdot T_S \\
&\quad + \int_{T_S}^T (m - m_P - |L^\star \cap \text{Ball}(c, u)|) du \\
&\leq \quad \Delta \cdot (T_S - T_O) - l \cdot T_S + \int_{T_S}^T (m - m_P - |L^\star \cap \text{Ball}(c, u)|) du.
\end{aligned}
$$

For $T_S < u < T$, we have $|L^\star \cap \text{Ball}(c, u)| \geq m - \Delta$, thus

$$
\begin{aligned}
\text{cost}^{(m_P)}(L^\star, c) - \text{cost}^{(m_S)}(S_O, c) \\
\leq \quad & \Delta \cdot (T_S - T_O) + \int_{T_S}^T (\Delta - m_P) du - l \cdot T_S \\
\leq \quad & \Delta \cdot (T_S - T_O) + \Delta \cdot (T - T_S) - l \cdot T_S \\
\leq \quad & 2 \cdot \Delta \cdot (T_I - T_O) - l \cdot T_S && (T_O \leq T_S \leq T_I \text{ and } T \leq T_I).
\end{aligned}
$$

Adding Inequality (29) and the above inequality, we obtain

$$\text{cost}^{(m)}(P,c) - \text{cost}^{(m)}(S_O \cup P_I^\star, c)$$
$$\leq \quad 2 \cdot \Delta \cdot (T_I - T_O) + l \cdot (d_{m-m_P+1} - T_S)$$
$$\leq \quad 2 \cdot \Delta \cdot (T_I - T_O) + l \cdot (T_I - T_O) \qquad \text{(Inequality (33))}$$
$$\leq \quad (2 \cdot \Delta + |m_P - m_S|) \cdot (T_I - T_O).$$

Secondly, we prove that $\text{cost}^{(m_S)}(S_O \cup P_I^\star, c) - \text{cost}^{(m_P)}(P,c) \leq (\Delta + |m_P - m_S|) \cdot (T_I - T_O)$. By Inequality (26) and Inequality (27), we have

$$\text{cost}^{(m_S)}(S_O, c) - \text{cost}^{(m_P)}(L^\star, c)$$
$$= \quad \int_0^{T_S} (m - m_S - w(S_O \cap \text{Ball}(c, u))) du$$
$$\quad - \int_0^{T} (m - m_P - |L^\star \cap \text{Ball}(c, u)|) du$$
$$= \quad \int_0^{T_S} (m_P - m_S + |L^\star \cap \text{Ball}(c, u)| - w(S_O \cap \text{Ball}(c, u))) du$$
$$\quad - \int_{T_S}^{T} (m - m_P - |L^\star \cap \text{Ball}(c, u)|) du \qquad (T_S \leq T)$$
$$\leq \quad \int_0^{T_S} (m_P - m_S + |L^\star \cap \text{Ball}(c, u)| - w(S_O \cap \text{Ball}(c, u))) du$$
$$\leq \quad \int_{T_O}^{T_S} (|L^\star \cap \text{Ball}(c, u)| - w(S_O \cap \text{Ball}(c, u))) du + l \cdot T_S$$
$$\leq \quad \Delta \cdot (T_S - T_O) + l \cdot T_S.$$

Adding Inequality (28) and the above inequality, we have

$$\text{cost}^{(m)}(S_O \cup P_I^\star, c) - \text{cost}^{(m)}(P, c)$$
$$\leq \quad \Delta \cdot (T_S - T_O) + l \cdot (T_S - d_{m_S'})$$
$$\leq \quad \Delta \cdot (T_I - T_O) + l \cdot (T_I - T_O) \qquad \text{(Inequality (34) and } T_S \leq T_I)$$
$$\leq \quad (\Delta + |m_P - m_S|) \cdot (T_I - T_O).$$

In summary, we have

$$|\text{cost}^{(m)}(P, c) - \text{cost}^{(m)}(S_O \cup P_I^\star, c)| \leq 2 \cdot (\Delta + |m_P - m_S|) \cdot (T_I - T_O).$$

Similarly, we can get the same conclusion when $T < T_S$. Moreover, for the case that $m_P < m_S$, with the help of Inequalities (30)(31)(33)(35), the conclusion still holds.

**Case 3:** $m_P = m_S \neq m$

By Inequality (32), we have

$$|\text{cost}^{(m)}(S_O \cup P_I^\star, c) - \text{cost}^{(m)}(P, c)| = |\text{cost}^{(m_S)}(S_O, c) - \text{cost}^{(m_P)}(L^\star, c)|.$$

By a similar argument as in Case 2, we have

$$|\text{cost}^{(m_S)}(S_O \cup P_I^\star, c) - \text{cost}^{(m_P)}(P, c)| \leq 2 \cdot (\Delta + |m_P - m_S|) \cdot (T_I - T_O).$$

Overall, we complete the proof of Lemma E.4.

$\square$

The Lemma E.4 tells us that at most $2(\Delta + |m_P - m_S|)$ points contribute to the error, with each point inducing at most $T_I - T_O$ error. We now analyze the magnitude of $T_I - T_O$.

**Lemma E.5** (Bounding induced error of each point). $T_I - T_O \leq O(\text{cost}^{(m)}(P, c)/m)$

*Proof.* Let $r_{\max} := \max_{p \in P_I^\star} \mathrm{dist}(p, c^\star)$ denote the maximum distance from $P_I^\star$ to $c^\star$. Let $d_{\max} := \mathrm{dist}(c, c^\star)$ denote the distance between the approximate center $c^\star$ and the center $c$. Let $\bar{r} := \frac{\mathrm{cost}(P_I^\star, c^\star)}{n-m}$ denote the average distance from $P_I^\star$ to $c^\star$.

Utilizing the triangle inequality, for any point $p \in L^\star$, we can assert that $\mathrm{dist}(p, c) \geq \mathrm{dist}(p, c^\star) - \mathrm{dist}(c, c^\star) \geq r_{\max} - d_{\max}$, thus $T_O \geq r_{\max} - d_{\max}$. Similarly, for any point $p \in P_I^\star$, we have $\mathrm{dist}(p, c) \leq \mathrm{dist}(p, c^\star) + \mathrm{dist}(c, c^\star) = r_{\max} + d_{\max}$, thus $T_I \leq r_{\max} + d_{\max}$. Consequently, depending on whether $r_{\max}$ is greater than or less than $d_{\max}$, we derive different expressions for $T_I - T_O$: If $r_{\max} \geq d_{\max}$, then $T_I - T_O \leq 2 \cdot d_{\max}$; if $r_{\max} < d_{\max}$, given that $T_O \geq 0$, it follows that $T_I - T_O \leq r_{\max} + d_{\max} < 2 \cdot d_{\max}$. Therefore, we turn to prove $2 \cdot m \cdot d_{\max} \leq O(\mathrm{cost}^{(m)}(P, c))$. We discuss the relationship between $d_{\max}$ and $\bar{r}$ in two cases.

**Case 1:** $d_{\max} \leq 4\bar{r}$ By definition of $\bar{r}$, we have

$$
\begin{aligned}
4 \cdot \mathrm{cost}^{(m)}(P, c^\star) = \quad & 4 \cdot (n - m)\bar{r} \\
\geq \quad & (n - m)d_{\max} \\
\geq \quad & 2 \cdot m \cdot d_{\max} \qquad\qquad (n \geq 4m).
\end{aligned}
$$

since $c^\star$ is an $O(1)$-approximate solution for robust geometric median, we have

$$
O(\mathrm{cost}^{(m)}(P, c)) \geq 2 \cdot m \cdot d_{\max}
$$

.

**Case 2:** $d_{\max} > 4\bar{r}$ Let $\widehat{P} := \{p \in P_I^\star | \mathrm{dist}(p, c^\star) \leq 2\bar{r}\}$. By definition, we have $|\widehat{P}| \geq \frac{1}{2}|P_I^\star| = \frac{n-m}{2}$. Since $d_{\max} > 4\bar{r}$, we obtain

$$
\begin{aligned}
8 \cdot \mathrm{cost}^{(m)}(P, c) \geq \quad & 8 \cdot ((n - m)/2 - m) \min_{p \in \widehat{P}} \mathrm{dist}(p, c) \\
\geq \quad & 4 \cdot m \cdot \min_{p \in \widehat{P}} \mathrm{dist}(p, c) \qquad\qquad (n \geq 4m) \\
\geq \quad & 4 \cdot m \cdot \min_{p \in \widehat{P}}(d_{\max} - \mathrm{dist}(p, c^\star)) \qquad \text{(Triangle Inequality)} \\
\geq \quad & 4 \cdot m \cdot (d_{\max} - 2\bar{r}) \\
> \quad & 2 \cdot m \cdot d_{\max}.
\end{aligned}
$$

Overall, we have $T_I - T_O \leq O(\mathrm{cost}^{(m)}(P, c)/m)$, which completes the proof. $\qquad\square$

Combining Lemmas E.4 and E.5, we obtain the bound: $|\mathrm{cost}^{(m)}(P, c) - \mathrm{cost}^{(m)}(S_O \cup P_I^\star, c)| \leq O(1) \cdot (\Delta + |m_P - m_S|) \cdot \frac{\mathrm{cost}^{(m)}(P, c)}{m}$. To prove $|\mathrm{cost}^{(m)}(P, c) - \mathrm{cost}^{(m)}(S_O \cup P_I^\star, c)| \leq O(\varepsilon) \cdot \mathrm{cost}^{(m)}(P, c)$, it remains to ensure that $\Delta = O(\varepsilon m)$ and $|m_P - m_S| = O(\varepsilon m)$. We show that both conditions hold when $S_O$ is an $\varepsilon$-approximation for the ball range space on $L^\star$.

**Lemma E.6** (Bounding misaligned outlier count). *Suppose $S_O$ is an $\varepsilon$-approximation for the ball range space on $L^\star$, we have $|m_P - m_S| \leq 2 \cdot \varepsilon \cdot m$.*

Before proving this lemma, we first analyze the properties of $w'$. Given a point $p \in S_O \cup P_I^\star$, we say $p$ is of partial weight w.r.t. $c$ if $0 < w'(p) < w(p)$ when $p \in S_O$ and $0 < w'(p) < 1$ when $p \in P_I^\star$. Intuitively, such a point is partially an inlier and partially an outlier of $S_O \cup P_I^\star$ to $c$. The following claim indicates that the number of partial-weight points is at most one for any $c$.

*Claim 1* (Properties of partial-weight points). For every center $c \subset \mathbb{R}^d$, there exists at most one point $v \in S_O \cup P_I^\star$ of partial weight w.r.t. $c$. Moreover, for any other point $p \in S_O \cup P_I^\star$ with $w'(p) = w(p)$, we have $\mathrm{dist}(p, c) \leq \mathrm{dist}(v, c)$.

*Proof.* By contradiction, we assume that there exist two points $v, v' \in S_O \cup P_I^\star$ of partial weight w.r.t. $c$. W.l.o.g., suppose $\mathrm{dist}(v, c) \leq \mathrm{dist}(v', c)$. We note that transferring weight from $v$ to $v'$ increases $w'(v)$ by any constant $\delta > 0$ and decreases $w'(v')$ by $\delta$ does not increase the term $\mathrm{cost}^{(m)}(S_O \cup P_I^\star, c)$. Then by selecting a suitable $\delta$, we can make either $v$ or $v'$ no longer of partial weight. Hence, there exists at most one point $v \in S_O \cup P_I^\star$ of partial weight w.r.t. $c$.

For any other point $p \in S_O \cup P_I^\star$ with $w'(p) = w(p)$, suppose $\text{dist}(v, c) < \text{dist}(p, c)$. In this case, increasing $w'(v)$ by a small amount $\delta > 0$ and decreasing $w'(p)$ by $\delta$ decreases $\text{cost}^{(m)}(S_O \cup P_I^\star, c)$. This contradicts the definition of $w'$, which completes the proof. $\square$

Recall that $d_1, \ldots, d_m$ represent the distances from the $m$ furthest points in $P_I^\star$ to the center $c$. Now we are ready to prove Lemma E.6.

*Proof of Lemma E.6.* We only need to consider the case of $m_P > m_S$ and $m_P < m_S$. Without loss of generality, we assume that $m_P > m_S$. Let $l_1 := m - m_P + 1$. Based on the definition of $d_1, ..., d_m$, each inlier point $p$ satisfies $\text{dist}(p, c) \leq d_{l_1 - 1}$. There are $m - m_P$ inlier points in $L^\star$, so we have:

$$|L^\star \cap \text{Ball}(c, d_{l_1})| \leq m - m_P. \tag{38}$$

Let $l_2 := m - \lfloor m_S \rfloor + 1$. Since $m_P > m_S$, we have $m_P \geq \lfloor m_S \rfloor + 1$, then we get the inequality

$$l_2 - 1 \geq l_1. \tag{39}$$

Moreover, we claim that

$$m - w(S_O \cap \text{Ball}(c, d_{l_2 - 1})) \leq \lfloor m_S \rfloor + 1. \tag{40}$$

By Claim 1, at most one point $v \in S_O \cup P_I^\star$ of partial weight w.r.t. $c$ exists. If $v \in P_I^\star$, we have $\text{dist}(v, c) = d_{l_2}$. Any point $v' \in P_I^\star$ with $\text{dist}(v', c) \geq d_{l_2 - 1}$ is an outlier of $S_O \cup P_I^\star$. Then there are $m - \lfloor m_S \rfloor - 1$ points in $P_I^\star$ that have distance to $C$ at least $d_{l_2 - 1}$. By contradiction, we assume that $m - w(S_O \cap \text{Balls}(c, d_{l_2 - 1})) > \lfloor m_S \rfloor + 1$. Then there are no less than $m$ points in $P$ that have a distance to $c$ greater than $d_{l_2 - 1}$, which contradicts the fact that $v'$ is an outlier of $S_O \cup P_I^\star$. Hence, Inequality (40) holds. For the case that $v \in S_O$ or there is no point of partial weight w.r.t. $c$, the argument is similar.

Since $m_P > m_S$, we have $w(S_O \cap \text{Ball}(c, d_{l_1})) > m - m_P$. Combining with Inequality (38), we conclude that

$$\begin{aligned} |L^\star \cap \text{Ball}(c, d_{l_1})| &\leq & m - m_P \\ &<& w(S_O \cap \text{Ball}(c, d_{l_1})). \end{aligned} \tag{41}$$

Moreover,

$$\begin{aligned} w(S_O \cap \text{Ball}(c, d_{l_1})) &\geq& w(S_O \cap \text{Ball}(c, d_{l_2 - 1})) \\ &>& m - \lfloor m_S \rfloor - 1. \end{aligned} \tag{42}$$

Then, we have

$$\begin{aligned} ||L^\star \cap \text{Ball}(c, d_{l_1})| &- w(S_O \cap \text{Ball}(c, d_{l_1}))| \\ &= & w(S_O \cap \text{Ball}(c, d_{l_1})) - |L^\star \cap \text{Ball}(c, d_{l_1})| & \text{(Inequality (41))} \\ &\geq& m_P - \lfloor m_S \rfloor - 1 & \text{(Inequality (38) and (42))} \\ &\geq& m_P - m_S - 1. \end{aligned}$$

Since $S_O$ is an $\varepsilon$-approximation for the ball range space on $L^\star$, by Lemma E.2, we have

$$||L^\star \cap \text{Ball}(c, d_{l_1})| - w(S_O \cap \text{Ball}(c, d_{l_1}))| \leq \varepsilon \cdot m.$$

Combining the above two inequalities, we have

$$|m_P - m_S| \leq \varepsilon \cdot m + 1 \leq 2 \cdot \varepsilon \cdot m.$$

Similarly, we can get the same conclusion when $m_P < m_S$, which completes the proof of Lemma E.6. $\square$

Now we are ready to prove Lemma 3.2.

*Proof of Lemma 3.2.* Based on Lemmas E.3 and E.4, we have

$$\begin{aligned} |\text{cost}^{(m)}(P, c) &- \text{cost}^{(m)}(P_I^\star \cup S_O, c)| \\ &\leq& 2 \cdot (\Delta + |m_P - m_S|) \cdot (T_I - T_O) \\ &\leq& O(1) \cdot (\Delta + |m_P - m_S|) \cdot \text{cost}^{(m)}(P, c)/m & \text{(Lemma E.5)} \\ &\leq& O(1) \cdot (\varepsilon m + |m_P - m_S|) \cdot \text{cost}^{(m)}(P, c)/m & \text{(Lemma E.8)} \\ &\leq& O(\varepsilon) \cdot \text{cost}^{(m)}(P, c), & \text{(Lemma E.2)} \end{aligned}$$

which completes the proof. $\square$

## E.2 Extension to other metric spaces

In this section, we explore the extension of Algorithm 2, designed for the robust geometric median in Euclidean space, to the metric spaces by leveraging the notions of VC dimension and doubling dimension since the VC dimension is known for various metric spaces [8, 9, 18, 40].

Let $(\mathcal{X}, \text{dist})$ denote the metric space, where $\mathcal{X}$ is the set under consideration, and $\text{dist} : \mathcal{X} \times \mathcal{X} \to \mathbb{R}_{\geq 0}$ is a function that measures the distance between points in $\mathcal{X}$, satisfying the triangle inequality. Specifically, in the Euclidean metric, $\mathcal{X}$ represents Euclidean space $\mathbb{R}^d$ and $\text{dist}$ is the Euclidean distance.

Similar to the analysis for robust geometric median on Euclidean space, we discuss the induced error of $S_I$ and $S_O$ separately. We first introduce how to bound the error induced by $S_O$. Since the $\text{dist}$ function satisfies the triangle inequality, our Lemmas E.3-E.4 hold, ensuring that each point in $S_O$ induces at most $O(\text{cost}^{(m)}(P, c)/m)$ error. To ensure the number of points in $S_O$ inducing error is $O(\varepsilon m)$, it suffices for $S_O$ to be an $\varepsilon$-approximation for the ball range space on $L^\star$ (as shown in Lemma E.6). Next, we illustrate how this condition can be satisfied using the notions: 1) VC dimension; 2) doubling dimension.

**VC dimension.** We begin by introducing the concept of the VC dimension of the ball range space, which serves as a measure of the complexity of this ball range space.

**Definition E.7** (VC dimension of ball range space). Let $M = (\mathcal{X}, \text{dist})$ be the metric space and define $\text{Balls}(\mathcal{X}) := \{\text{Ball}(c, u) \mid c \in \mathcal{X}, u > 0\}$ as the collection of balls in the space. The VC dimension of the ball range space $(\mathcal{X}, \text{Balls}(\mathcal{X}))$, denoted by $d_{VC}(\mathcal{X})$, is the maximum $|P|, P \subseteq \mathcal{X}$ such that $|P \cap \text{Balls}(\mathcal{X})| = 2^{|P|}$, where $P \cap \text{Balls}(\mathcal{X}) := \{P \cap \text{Ball}(c, u) \mid \text{Ball}(c, u) \in \text{Balls}(\mathcal{X})\}$.

The VC dimension of $(\mathcal{X}, \text{Balls}(\mathcal{X}))$ aligns to the pseudo-dimension of $(\mathcal{X}, \text{Balls}(\mathcal{X}))$ used by [45]. Based on this observation, we give out a refined lemma from [45].

**Lemma E.8** (Refined from [45]). *Let $M = (\mathcal{X}, \text{dist})$ be the metric space with VC dimension $d_{VC}(\mathcal{X})$. Given dataset $P_O \subseteq \mathcal{X}$, assume $S_O$ is a uniform sample of size $\tilde{O}(\frac{d_{VC}(\mathcal{X})}{\varepsilon^2})$ from $P_O$, then with probability $1 - 1/poly(1/\varepsilon)$, $S_O$ is an $\varepsilon$-approximation of the ball range space on $P_O$.*

The Lemma E.2, used earlier for the Euclidean space case, is a direct corollary, since in Euclidean space $\mathbb{R}^d$, we have $d_{VC}(\mathcal{X}) = O(d)$. This lemma illustrates the number of samples required from $L^\star$.

**Doubling dimension.** Another notion to measure the complexity of the ball range space is the doubling dimension.

**Definition E.9** (Doubling dimension [31, 3]). The doubling dimension $\text{ddim}(\mathcal{X})$ of a metric space $(\mathcal{X}, \text{dist})$ is the least integer $t$ such that for any $\text{Ball}(c, u)$ with $c \in \mathcal{X}, u > 0$, it can be covered by $2^t$ balls of radius $u/2$.

We denote the metric space with bounded doubling dimension as **doubling metric**. Based on [38], it is known that the VC dimension of a doubling metric $(\mathcal{X}, \text{dist})$ may not be bounded. Therefore, the previous Lemma E.8 does not apply for an effective bound. We want to directly relate the ball range space bound to the doubling dimension. This goal can be achieved by the following refined lemma.

**Lemma E.10** (Ball range space approximation in doubling metrics [38]). *Let $M = (\mathcal{X}, \text{dist})$ be the metric space with doubling dimension $\text{ddim}(\mathcal{X})$. Given $P_O \subseteq \mathcal{X}$, assume $S_O$ is a uniform sample of size $\tilde{O}(\text{ddim}(\mathcal{X})\varepsilon^{-2})$ from $P_O$, then with probability $1 - 1/poly(1/\varepsilon)$, $S_O$ is an $\varepsilon$-approximation of the ball range space on $P_O$.*

*Proof.* Suppose the distorted distance function $\text{dist}' : \mathcal{X} \times \mathcal{X} \to \mathbb{R}_{\geq 0}$ satisfies: for any $x, y \in \mathcal{X}$, we have $(1 - \varepsilon)\text{dist}(x, y) \leq \text{dist}'(x, y) \leq (1 + \varepsilon)\text{dist}(x, y)$. By Lemma 3.1 of [38], we know that with probability $1 - poly(1/\varepsilon)$, $S_O$ is an $\varepsilon$-ball range space approximation of $P_O$ w.r.t. $\text{dist}'$, when $S_O = \tilde{O}(\text{ddim}(\mathcal{X})\varepsilon^{-2})$. By setting of $\text{dist}'$, we know $S_O$ is an $O(\varepsilon)$-ball range space approximation of $P_O$ w.r.t. $\text{dist}$, which completes the proof. $\square$

This lemma illustrates the number of points needed to be sampled in order to bound the induced error of $S_O$ when the metric space is a doubling metric.

The remaining problem is to bound the induced error of $S_I$. We give out a generalized version of Theorem 3.1 as below.

**Theorem E.11** (Refined from Corollary 5.4 in [40]). *Let $M = (\mathcal{X}, \text{dist})$ be the metric space. There exists a randomized algorithm that in $O(n)$ time constructs a weighted subset $S_I \subseteq P_I^\star$ of size:*

- *$\tilde{O}(d_{VC}(\mathcal{X}) \cdot \varepsilon^{-4})$, w.r.t. the VC dimension $d_{VC}(\mathcal{X})$,*

- *$\tilde{O}(\text{ddim}(\mathcal{X}) \cdot \varepsilon^{-2})$, w.r.t. the doubling dimension $\text{ddim}(\mathcal{X})$,*

*such that for every dataset $P_O$ of size $m$, every integer $0 \leq t \leq m$ and every center $c \in \mathbb{R}^d$, $|\text{cost}^{(t)}(P_O \cup P_I^\star, c) - \text{cost}^{(t)}(P_O \cup S_I, c)| \leq \varepsilon \cdot \text{cost}^{(t)}(P_O \cup P_I^\star, c) + 2\varepsilon \cdot \text{cost}(P_I^\star, c^\star)$.*

By combining the discussions of the errors induced by $S_O$ and $S_I$, we present the main theorem for constructing a coreset for the robust geometric median across various metric spaces. We study the shortest-path metric $(\mathcal{X}, \text{dist})$, where $\mathcal{X}$ is the vertex set of a graph $G = (V, E)$, and $\text{dist}(\cdot, \cdot)$ measures the shortest distance between two points in the graph. The treewidth of a graph measures how "tree-like" the graph is; see Definition 2.1 of [4] for formal definition. A graph excluding a fixed minor is one that does not contain a particular substructure, known as the minor.

**Theorem E.12** (Coreset size for robust geometric median in various metric spaces). *Let $\varepsilon \in (0, 1)$. For a metric space $M = (\mathcal{X}, \text{dist})$ and a dataset $X \subset \mathcal{X}$ of size $n \geq 4m$, let $S = S_O \cup S_I$ be a sampled set of size*

- *$\tilde{O}(\log(|\mathcal{X}|) \cdot \varepsilon^{-4})$ if $M$ is general metric space.*

- *$\tilde{O}(\text{ddim}(\mathcal{X}) \cdot \varepsilon^{-2})$ if $M$ is a doubling metric with doubling dimension $\text{ddim}(\mathcal{X})$.*

- *$\tilde{O}(t \cdot \varepsilon^{-4})$ if $M$ is a shortest-path metric of a graph with bounded treewidth $t$.*

- *$\tilde{O}(|H| \cdot \varepsilon^{-4})$ if $M$ is a shortest-path metric of a graph that excludes a fixed minor $H$.*

*Then $S$ is an $\varepsilon$-coreset for robust geometric median on $X$.*

This theorem illustrates the coreset size with respect to different metric spaces, improving upon previous results for robust coresets by eliminating the $O(m)$ dependency.

*Proof of Theorem E.12.* If $M = (\mathcal{X}, \text{dist})$ is a general metric space, then $d_{VC}(\mathcal{X}) = O(\log |\mathcal{X}|)$. Thus, we have $|S_O| = \tilde{O}(\log(|\mathcal{X}|) \cdot \varepsilon^{-2})$ by Lemma E.8 and $|S_I| = \tilde{O}(\log(|\mathcal{X}|) \cdot \varepsilon^{-4})$ by Theorem E.11, leading to a coreset of size $\tilde{O}(\log(|\mathcal{X}|) \cdot \varepsilon^{-4})$.

If $M = (\mathcal{X}, \text{dist})$ is a doubling metric space, then by definition, $\text{ddim}(\mathcal{X})$ is bounded. Thus we have $|S_O| = \tilde{O}(\text{ddim}(\mathcal{X}) \cdot \varepsilon^{-2})$ by Lemma E.10 and $|S_I| = \tilde{O}(\text{ddim}(\mathcal{X}) \cdot \varepsilon^{-2})$ by Theorem E.11, leading to a coreset of size $\tilde{O}(\text{ddim}(\mathcal{X}) \cdot \varepsilon^{-2})$.

If $M = (\mathcal{X}, \text{dist})$ is a shortest-path metric of a graph with bounded treewidth $t$, we have $d_{VC}(\mathcal{X}) = O(t)$ by [8]. Thus, we have $|S_O| = \tilde{O}(t \cdot \varepsilon^{-2})$ by Lemma E.8 and $|S_I| = \tilde{O}(t \cdot \varepsilon^{-4})$ by Theorem E.11, leading to a coreset of size $\tilde{O}(t \cdot \varepsilon^{-4})$.

If $M = (\mathcal{X}, \text{dist})$ is a shortest-path metric of a graph that excludes a fixed minor $H$, we have $d_{VC}(\mathcal{X}) = O(|H|)$ by [8]. Thus, we have $|S_O| = \tilde{O}(|H| \cdot \varepsilon^{-2})$ by Lemma E.8 and $|S_I| = \tilde{O}(|H| \cdot \varepsilon^{-4})$ by Theorem E.11, leading to a coreset of size $\tilde{O}(|H| \cdot \varepsilon^{-4})$.

$\square$

# F  Proof of Theorem 1.5: robust $(k, z)$-clustering

In this section, we provide a coreset construction for robust $(k, z)$-clustering when $d \geq 1$. We first extend the definition of the robust geometric median and its coreset to the robust $(k, z)$-clustering and its corresponding coreset.

**Definition F.1** (Robust $(k, z)$-clustering). Given a dataset $P \subset \mathbb{R}^d$ of size $n \geq 1$ and an integer $m \geq 0$, the robust $(k, z)$-clustering problem is to find a center set $C \subset \mathbb{R}^d$, $|C| = k$ that minimizes the objective function below:

$$\text{cost}_z^{(m)}(P, C) := \min_{L \subset P : |L| = m} \sum_{p \in P \setminus L} (\text{dist}(p, C))^z,$$

where $L$ represents the set of $m$ outliers w.r.t. $C$ and $\text{dist}(p, C) = \min_{c \in C} \text{dist}(p, c)$ denotes the Euclidean distance from $p$ to the closest center among $C$.

Before we define the coreset for robust $(k, z)$-clustering, we introduce the generalized cost function in the context of the robust $(k, z)$-clustering.

**Definition F.2** (Generalized cost function for robust $(k, z)$-clustering). Let $m$ be an integer. Let $S \subseteq \mathbb{R}^d$ be a weighted dataset with weights $w(p)$ for each point $p \in S$. Let $w(S) := \sum_{p \in S} w(p)$. Define a collection of weight functions $\mathcal{W} := \{w' : S \to \mathbb{R}^+ \mid \sum_{p \in S} w'(p) = w(S) - m \wedge \forall p \in S, w'(p) \leq w(p)\}$. Moreover, we define the following cost function on $S$:

$$\forall C \subset \mathbb{R}^d, |C| = k \quad \text{cost}_z^{(m)}(S, C) := \min_{w' \in \mathcal{W}} \sum_{p \in S} w'(p) \cdot (\text{dist}(p, C))^z.$$

With this definition of the generalized cost function, we now define the notion of a coreset for robust $(k, z)$-clustering.

**Definition F.3** (Coreset for robust $(k, z)$-clustering). Given a point set $P \subset \mathbb{R}^d$ of size $n \geq 1$, integer $m \geq 1$ and $\varepsilon \in (0, 1)$, we say a weighted subset $S \subseteq P$ together with a weight function $w : S \to \mathbb{R}^+$ is an $\varepsilon$-coreset of $P$ for robust $(k, z)$-clustering if $w(S) = n$ and for any center set $C \subset \mathbb{R}^d$, $|C| = k$, $\text{cost}_z^{(m)}(S, C) \in (1 \pm \varepsilon) \cdot \text{cost}_z^{(m)}(P, C)$.

Finally, we extend the concepts of inliers, outliers, and the ball range space.

**Inliers and outliers.** Throughout this section, we denote the approximate center set for robust $(k, z)$-clustering by $C^\star = \{c_1^\star, \ldots, c_k^\star\} \subset \mathbb{R}^d$, which is an $O(1)$-approximation of the optimal solution. We then define the inliers and outliers with respect to this approximation center set.

Let $L^\star := \arg\min_{L \subset P, |L| = m} \text{cost}_z(P - L, C^\star)$ denote the set of $m$ outliers with respect to $C^\star$, and define $P_I^\star := P \setminus L^\star$ as the corresponding inlier set. The inlier set $P_I^\star$ is naturally partitioned by $C^\star$ into $k$ clusters $\{P_1^\star, \ldots, P_k^\star\}$, where each cluster $P_i^\star$ contains the points in $P_I^\star$ closest to its corresponding center $c_i^\star$. Additionally, define $r_{\max} := \max_{p \in P_I^\star} \text{dist}(p, C^\star)$ which represents the maximum distance from the points in $P_I^\star$ to this center set $C^\star$. Let $\bar{r} := \sqrt[z]{\frac{\text{cost}_z(P_I^\star, C^\star)}{n - m}}$ denote the average distance from $P_I^\star$ to $C^\star$. Under these notations, the second condition of Assumption 1.4 can be rewritten as $(r_{\max})^z \leq 4k (\bar{r})^z$.

**Ball range space.** We introduce the concept of the $k$-balls range space, which is a direct extension of the ball range space defined in Definition E.1.

**Definition F.4** (Approximation of $k$-balls range space, Definition F.2 in [39]). Let $\text{Balls}(C, u) := \cup_{c \in C} \text{Ball}(c, u)$. For a given dataset $P \subset \mathbb{R}^d$, the $k$-balls range space on $P$ is $(P, \mathcal{P})$ where $\mathcal{P} := \{P \cap \text{Balls}(C, u) \mid C \subset \mathbb{R}^d, |C| = k, u \in \mathbb{R}^+\}$. A subset $Y \subset P$ is called an $\varepsilon$-approximation of the $k$-balls range space $(P, \mathcal{P})$ if for every $C \subset \mathbb{R}^d$, $|C| = k$, $u \in \mathbb{R}^+$,

$$\left| \frac{|P \cap \text{Balls}(C, u)|}{|P|} - \frac{|Y \cap \text{Balls}(C, u)|}{|Y|} \right| \leq \varepsilon.$$

Based on this definition, we have the following lemma that measures the performance of $S_O$ for robust $(k, z)$-clustering.

**Lemma F.5** (Refined from Lemma F.3 of [39]). *Given dataset $P_O \subset \mathbb{R}^d$. Let $S_O$ be a uniform sampling of size $\tilde{O}(\frac{kd}{\varepsilon^2})$ from $P_O$, then with probability at least $1 - \frac{1}{poly(k/\varepsilon)}$, $S_O$ is an $\varepsilon$-approximation of the $k$-balls range space on $P_O$. Define a weight function $w$: $w(p) = \frac{|P_O|}{|S_O|}$, for any $p \in S_O$. Then for any $C \subset \mathbb{R}^d$, $|C| = k$, $u \in \mathbb{R}^+$,*

$$||P_O \cap \text{Balls}(C, u)| - w(S_O \cap \text{Balls}(C, u))| \leq \varepsilon |P_O|.$$

Then we are ready to give out our main result.

## F.1 Result for robust $(k, z)$-clustering

We first recall the following theorem.

**Theorem F.6** (Restatement of Corollary 5.4 in [40]). *There exists a randomized algorithm $\mathcal{A}_{kd}$ that in $O(nkd)$ time constructs a weighted subset $S_I \subseteq P_I^\star$ of size $\tilde{O}(k^2 \varepsilon^{-2z} \min\{\varepsilon^{-2}, d\}))$, such that for every dataset $P_O$ of size $m$, every integer $0 \leq t \leq m$ and every center set $C \subset \mathbb{R}^d$, $|C| = k$, $|\mathrm{cost}_z^{(t)}(P_O \cup P_I^\star, C) - \mathrm{cost}_z^{(t)}(P_O \cup S_I, C)| \leq \varepsilon \cdot \mathrm{cost}_z^{(t)}(P_O \cup P_I^\star, C) + 2\varepsilon \cdot \mathrm{cost}_z(P_I^\star, C^\star)$.*

This theorem is a generalization of Theorem 3.1, describing the number of points that need to be sampled from $P_I^\star$ for robust $(k, z)$-clustering.

To adapt Algorithm 2 for Theorem 1.5, we only need to adjust the size of $S_O$ to $\tilde{O}(k\varepsilon^{-2z} \min\{\varepsilon^{-2}, d\})$ in Line 2 and modify the algorithm $\mathcal{A}_d$ to $\mathcal{A}_{kd}$ in Line 3 (see Algorithm 3). Consequently, the runtime remains dominated by Line 1 and Line 3, resulting in an overall complexity of $O(ndk)$ according to Theorem F.6.

---

**Algorithm 3** Coreset Construction for General $d$ and General $k$

---

**Input:** A dataset $P \subset \mathbb{R}^d$, $\varepsilon \in (0, 1)$ and an $O(1)$-approximate center set $C^\star \subset \mathbb{R}^d$
**Output:** An $\varepsilon$-coreset $S$

1: $L^\star \leftarrow \arg\min_{L \subset P, |L|=m} \mathrm{cost}_z(P - L, C^\star)$, $P_I^\star \leftarrow P - L^\star$
2: Uniformly sample $S_O \subseteq L^\star$ of size $\tilde{O}(k\varepsilon^{-2z} \min\{\varepsilon^{-2}, d\})$. Set $\forall p \in S_O$, $w_O(p) \leftarrow \frac{m}{|S_O|}$.
3: Construct $(S_I, w_I) \leftarrow \mathcal{A}_{kd}(P_I^\star)$ by Theorem F.6.
4: For any $p \in S_O$, define $w(p) = w_O(p)$ and for any $p \in S_I$, define $w(p) = w_I(p)$.
5: Return $S \leftarrow S_O \cup S_I$ and $w$;

---

Similar to the robust geometric median, we present the key lemma used to prove Theorem 1.5 below.

**Lemma F.7** (Induced error of $S_O$). *For any center set $C \subset \mathbb{R}^d$, $|C| = k$, we have $|\mathrm{cost}_z^{(m)}(P, C) - \mathrm{cost}_z^{(m)}(S_O \cup P_I^\star, C)| \leq O(\varepsilon) \cdot \mathrm{cost}_z^{(m)}(P, C)$.*

Theorem 1.5 follows as a corollary of Theorem F.6 and this lemma, with a proof similar to that of Theorem 1.3.

*Remark* F.8. The second condition in Assumption 1.4 can be replaced with $\mathrm{dist}(c_i^*, c_j^*)^z \geq \frac{m \cdot r_{\max}^z}{\min\{|P_i^*|, |P_j^*|\}}$ for any $c_i^*, c_j^* \in C^*$, $i \neq j$. Under this modified assumption, Theorem 1.5 becomes a direct generalization of Theorem 1.3, as the second condition vanishes when $k = 1$. This modification ensures that clusters are sufficiently well-separated, which explains why our algorithm performs well on the **Bank** dataset when $k = 3$, as this modified assumption is satisfied in this case.

Below we illustrate why this assumption also works. If points in each cluster are mostly assigned to distinct centers, then $\mathrm{cost}_z^{(m)}(P, C) \approx \mathrm{cost}_z^{(m)}(P, C^\star)$, making the error $|\mathrm{cost}_z^{(m)}(P, C) - \mathrm{cost}_z^{(m)}(P_I^\star \cup S_O, C)|$ easy to bound. Alternatively, if two clusters each have at least half of their points assigned to the same center, then by the modified assumption, $\mathrm{cost}_z^{(m)}(P, C) > mr_{\max}^z$, which is large enough to bound the error induced by $S_O$.

## F.2 Proof of Lemma F.7: Induced error of $S_O$

In this section, we fix a center set $C \subset \mathbb{R}^d$ of size $k$ and define $T_O := \min_{p \in L^\star}(\mathrm{dist}(p, C))^z$, $T_I := \max_{p \in P_I^\star}(\mathrm{dist}(p, C))^z$. We observe that Lemmas E.3 and E.4 can be directly extended to robust $(k, z)$-clustering. Therefore, the induced error of each point is at most $T_I - T_O$. Next, we show that $T_I - T_O \leq O(\mathrm{cost}_z^{(m)}(P, c)/m)$, supported by a detailed discussion of the sizes of $T_I$ and $T_O$ (see Lemma F.9). The remaining task is to ensure that the number of points that may induce error is $O(\varepsilon m)$. The result of Lemma E.6 can be extended to robust $(k, z)$-clustering when $S_O$ is an $\varepsilon$-approximation of the $k$-balls range space on $L^\star$. By Lemma F.5, we sample $\tilde{O}(k\varepsilon^{-2z} \min\{\varepsilon^{-2}, d\})$ points from $L^\star$ to ensure the approximation.

We then demonstrate how to prove that $T_I - T_O \leq O(\mathrm{cost}_z^{(m)}(P, c)/m)$.

**Lemma F.9** (Bounds for $T_I$ and $T_O$). *When $m_P < m$, we have $T_I - T_O \leq O(\text{cost}_z^{(m)}(P, c)/m)$*

*Proof.* Let $d_{\max} := \max_{c^\star \in C^\star} \text{dist}(c^\star, C)$ denote the maximum distance of points in $C^\star$ to $C$. Let $d'_{\max} := \max_{c \in C} \text{dist}(c, C^\star)$ denote the maximum distance of points in $C$ to $C^\star$.

We first claim that $T_I \leq (r_{\max} + d_{\max})^z$. Let $\widehat{p}$ be defined as $\max_{p \in P_I^\star} \text{dist}(p, C)$. Denote $c_p^\star$ as the closest approximate center to $\widehat{p}$, where $c_p^\star := \arg\min_{c^\star \in C^\star} \text{dist}(\widehat{p}, c^\star)$, and $c_p$ as the closest center to $c_p^\star$, where $c_p := \arg\min_{c \in C} \text{dist}(c_p^\star, c)$. This setup allows us to establish an upper bound for $T_I$:

$$
\begin{aligned}
\text{dist}(\widehat{p}, C) \leq \quad & \text{dist}(\widehat{p}, c_p) \\
\leq \quad & \text{dist}(\widehat{p}, c_p^\star) + \text{dist}(c_p^\star, c_p) \quad &\text{(Triangle Inequality)} \\
\leq \quad & r_{\max} + d_{\max},
\end{aligned}
$$

thus, $T_I \leq (r_{\max} + d_{\max})^z$.

Next, we discuss the relationship between $C$ and $C^\star$ in two cases: 1) $r_{\max} \leq d_{\max}$; 2) $r_{\max} > d_{\max}$.

**Case 1:** $d_{\max} \geq r_{\max}$ In this case, we have $T_I - T_O \leq 2^z d_{\max}^z$ since $T_O \geq 0$. Therefore, it suffices to prove $2^z d_{\max}^z \leq O(\text{cost}_z^{(m)}(P, C)/m)$. Suppose $\text{dist}(c_i^\star, C) = d_{\max}$, let $(\bar{r}_i)^z := \text{cost}_z(P_i^\star, C^\star)/|P_i^\star|$. Based on the scale of $d_{\max}$, we discuss in two cases.

**Case 1.1:** $d_{\max} \leq 8\bar{r}_i$ By definition of $\bar{r}_i$, we have

$$
\begin{aligned}
16^z \cdot \text{cost}_z^{(m)}(P_i^\star, C^\star) = \quad & 16^z \cdot |P_i^\star| \cdot (\bar{r}_i)^z \\
\geq \quad & 2^z \cdot |P_i^\star| \cdot d_{\max}^z \\
\geq \quad & 2^z \cdot m \cdot d_{\max}^z \quad &\text{(Item 1 of Assumption 1.4).}
\end{aligned}
$$

Since $C^\star$ is an $O(1)$-approximate solution for robust $(k, z)$-clustering, we have

$$
2^z \cdot (d_{\max})^z \leq 16^z \cdot \text{cost}_z^{(m)}(P, C^\star)/m \leq O(\text{cost}_z^{(m)}(P, C)/m).
$$

**Case 1.2:** $d_{\max} > 8\bar{r}_i$

Let $\widehat{P}_i := \{p \in P_i^\star | \text{dist}(p, C^\star) \leq 4\bar{r}_i\}$. By definition, we have $|\widehat{P}_i| \geq \frac{3}{4}|P_i^\star|$. For any point $p \in \widehat{p}_i$, suppose $c_p := \arg\min_{c \in C} \text{dist}(p, c)$, then we have

$$
\begin{aligned}
\text{dist}(p, c) \geq \quad & \text{dist}(c_i^\star, c_p) - \text{dist}(p, c_i^\star) \quad &\text{(Triangle Inequality)} \\
\geq \quad & \text{dist}(c_i^\star, C) - \text{dist}(p, c_i^\star) \\
= \quad & d_{\max} - \text{dist}(p, c_i^\star) \quad &(43) \\
\geq \quad & d_{\max} - 4\bar{r}_i \quad &\text{(Definition of } \widehat{P}_i)
\end{aligned}
$$

Since $d_{\max} > 8\bar{r}_i$, we obtain

$$
\begin{aligned}
8^z \cdot \text{cost}_z^{(m)}(P, c) \geq \quad & 8^z \cdot (\frac{3}{4}|P_i^\star| - m) \min_{p \in \widehat{P}_i}(\text{dist}(p, C))^z \\
\geq \quad & 8^z \cdot (\frac{3}{4}|P_i^\star| - m)(d_{\max} - 4\bar{r}_i)^z \quad &\text{(Inequality (43))} \\
\geq \quad & 4^z \cdot (\frac{3}{4}|P_i^\star| - m)d_{\max}^z \\
\geq \quad & 2^z \cdot m \cdot d_{\max} \quad &\text{(Item 1 of Assumption 1.4).}
\end{aligned}
$$

Overall, we have $T_I - T_O \leq O(\text{cost}_z^{(m)}(P, C)/m)$.

**Case 2:** $d_{\max} < r_{\max}$ We have $T_I - T_O < 2^z \cdot r_{\max}^z$ since $T_O \geq 0$. By Item 2 of Assumption 1.4, we have $m \cdot r_{\max}^z \leq n \cdot \bar{r}^z$, thus $T_I - T_O \leq 2^z \cdot \text{cost}_z^{(m)}(P, C)/m$.

Combine Case 1 and Case 2, we complete the proof.

$\square$

## F.3 Extension to other metric spaces

In this section, we explore the extension of Algorithm 3 for robust $(k, z)$-clustering in Euclidean space, to various other metric spaces.

Similar to the analysis for robust geometric median, we discuss the induced error of $S_I$ and $S_O$ separately. For the error induced by $S_O$, Lemma F.9 still holds, so it suffices for $S_O$ to be an $\varepsilon$-approximation of the $k$-balls range space.

**VC dimension**   We begin by introducing the concept of the VC dimension of the $k$-balls range space.

**Definition F.10** (VC dimension of $k$-balls range space). Let $(\mathcal{X}, \mathrm{dist})$ be the metric space and define $\mathrm{Balls}_k(\mathcal{X}) := \{\mathrm{Balls}(C, u) \mid C \subset \mathcal{X}, |C| = k, u > 0\}$. The VC dimension of the $k$-balls range space $(\mathcal{X}, \mathrm{Balls}_k(\mathcal{X}))$, denoted by $d_{VC}(\mathcal{X})$, is the maximum $|P|$, $P \subseteq \mathcal{X}$ such that $|P \cap \mathrm{Balls}_k(\mathcal{X})| = 2^{|P|}$, where $P \cap \mathrm{Balls}_k(\mathcal{X}) := \{P \cap \mathrm{Balls}(C, u) \mid \mathrm{Balls}(C, u) \in \mathrm{Balls}_k(\mathcal{X})\}$.

The VC dimension of $(\mathcal{X}, \mathrm{Balls}_k(\mathcal{X}))$ aligns to the pseudo-dimension of $(\mathcal{X}, \mathrm{Balls}_k(\mathcal{X}))$ used by [45]. Based on this observation, we give out a refined lemma from [45].

**Lemma F.11** (Refined from [45]). *Let $(\mathcal{X}, \mathrm{dist})$ be the metric space. Given dataset $P_O \subseteq \mathcal{X}$, assume $S_O$ is a uniform sample of size $\tilde{O}(\frac{k \cdot d_{VC}(\mathcal{X})}{\varepsilon^2})$ from $P_O$, then with probability $1 - 1/poly(k/\varepsilon)$, $S_O$ is a $\varepsilon$-approximation of the $k$-balls range space on $P_O$.*

This lemma illustrates the number of points needed to be sampled from $L^\star$.

**Doubling dimension**   Similar to the Lemma E.10, we give out the Lemma F.12 that illustrates the number of points needed to be sampled in order to bound the induced error of $S_O$.

**Lemma F.12** (Balls range space approximation for the doubling dimension). *Let $M = (\mathcal{X}, \mathrm{dist})$ be the metric space with doubling dimension $\mathrm{ddim}(\mathcal{X})$. Given $P_O \subseteq \mathcal{X}$, assume $S_O$ is a uniform sample of size $\tilde{O}(\mathrm{ddim}(\mathcal{X}) \cdot \varepsilon^{-2z} \cdot k)$ from $P_O$, then with probability $1 - 1/poly(k, 1/\varepsilon)$, $S_O$ is an $\varepsilon$-approximation of the $k$-balls range space on $P_O$.*

The remaining problem is to bound the induced error of $S_I$. We give out a generalized version of Theorem E.11 as below.

**Theorem F.13** (Refined from Corollary 5.4 in [40]). *Let $(\mathcal{X}, \mathrm{dist})$ be the metric space. There exists a randomized algorithm that in $O(nk)$ time constructs a weighted subset $S_I \subseteq P_I^\star$ of size:*

- *$\tilde{O}(k^2 \cdot d_{VC}(\mathcal{X}) \cdot \varepsilon^{-2z-2})$, w.r.t. VC dimension $d_{VC}(\mathcal{X})$,*

- *$\tilde{O}(k^2 \cdot \mathrm{ddim}(\mathcal{X}) \cdot \varepsilon^{-2z})$, w.r.t. doubling dimension $\mathrm{ddim}(\mathcal{X})$,*

*, such that for every dataset $P_O$ of size $m$, every integer $0 \leq t \leq m$ and every center set $C \subset \mathbb{R}^d$, $|C| = k$, $|\mathrm{cost}_z^{(t)}(P_O \cup P_I^\star, C) - \mathrm{cost}_z^{(t)}(P_O \cup S_I, C)| \leq \varepsilon \cdot \mathrm{cost}_z^{(t)}(P_O \cup P_I^\star, C) + 2\varepsilon \cdot \mathrm{cost}_z(P_I^\star, C^\star)$.*

By combining the discussions of the errors induced by $S_O$ and $S_I$, we present the main theorem for constructing a coreset for the robust $(k, z)$-clustering across various metric spaces.

**Theorem F.14** (Coreset size for robust $(k, z)$-clustering in various metric spaces). *Let $\varepsilon \in (0, 1)$. For a metric space $M = (\mathcal{X}, \mathrm{dist})$ and a dataset $X \subset \mathcal{X}$ satisfying Assumption 1.4, let $S = S_O \cup S_I$ be a sampled set of size*

- *$\tilde{O}(k^2 \cdot \log(|\mathcal{X}|) \cdot \varepsilon^{-2z-2})$ if $M$ is general metric space.*

- *$\tilde{O}(k^2 \cdot \mathrm{ddim}(\mathcal{X}) \cdot \varepsilon^{-2z})$ if $M$ is a doubling metric with doubling dimension $\mathrm{ddim}(\mathcal{X})$.*

- *$\tilde{O}(k^2 \cdot t \cdot \varepsilon^{-2z-2})$ if $M$ is a shortest-path metric of a graph with bounded treewidth $t$.*

- *$\tilde{O}(k^2 \cdot |H| \cdot \varepsilon^{-2z-2})$ if $M$ is a shortest-path metric of a graph that excludes a fixed minor $H$.*

*Then $S$ is an $\varepsilon$-coreset for robust $(k, z)$-clustering on $X$.*

Table 4: Datasets used in our experiments. For each dataset, we report its size, dimension (DIM), number of outliers ($m$). The number of centers ($k$) is used in robust $(k, z)$-clustering. We also provide the values of $\min_i |P_i^\star|$ and $(r_{\max}/\bar{r})^z$ for robust $(k, z)$-clustering. Our Assumption 1.4 requires that $\min_i |P_i^\star| \geq 4m$ and $(r_{\max}/\bar{r})^z \leq 4k$. The value $Y$ (resp. $N$) indicates these assumptions are satisfied or not. Note that the **Athlete** dataset is sourced from Kaggle, while the other datasets are from the UCI repository.

| DATASET | SIZE | DIM. | $m$ | $k$ | $\min_i |P_i^\star|$ | | $\left(\frac{r_{\max}}{\bar{r}}\right)^z$ | |
|---|---|---|---|---|---|---|---|---|
| | | | | | $z = 1$ | $z = 2$ | $z = 1$ | $z = 2$ |
| TWITTER[11] | $10^5$ | 2 | 2000 | 5 | 8968,Y | 588,N | 3.819,Y | 1.083,Y |
| CENSUS1990 [46] | $10^5$ | 68 | 2000 | 5 | 5927,N | 5927,N | 1.873,Y | 3.252,Y |
| BANK[47] | 41188 | 10 | 824 | 5 | 1361,N | 1361,N | 4.674,Y | 15.731,Y |
| ADULT[6] | 48842 | 6 | 977 | 5 | 4508,Y | 186,N | 5.151,Y | 5.834,Y |
| ATHLETE[36] | 10000 | 4 | 200 | 5 | 1018,Y | 1018,Y | 2.923,Y | 7.062,Y |
| DIABETES[43] | 10000 | 10 | 200 | 5 | 1415,Y | 1459,Y | 2.551,Y | 2.527,Y |

This theorem illustrates the coreset size with respect to different metric spaces for robust $(k, z)$-clustering, improving upon previous results by eliminating the $O(m)$ dependency.

*Proof of Theorem F.14.* If $M = (\mathcal{X}, \text{dist})$ is a general metric space, then $d_{VC}(\mathcal{X}) = O(\log |\mathcal{X}|)$. Thus, we have $|S_O| = \tilde{O}(k \cdot \log(|\mathcal{X}|) \cdot \varepsilon^{-2})$ by Lemma E.8 and $|S_I| = \tilde{O}(k^2 \cdot \log(|\mathcal{X}|) \cdot \varepsilon^{-4})$ by Theorem E.11, leading to a coreset of size $\tilde{O}(k^2 \cdot \log(|\mathcal{X}|) \cdot \varepsilon^{-4})$.

If $M = (\mathcal{X}, \text{dist})$ is a doubling metric space, then by definition, $\text{ddim}(\mathcal{X})$ is bounded. Thus we have $|S_O| = \tilde{O}(k \cdot \text{ddim}(\mathcal{X}) \cdot \varepsilon^{-2})$ by Lemma E.10 and $|S_I| = \tilde{O}(k^2 \cdot \text{ddim}(\mathcal{X}) \cdot \varepsilon^{-2})$ by Theorem E.11, leading to a coreset of size $\tilde{O}(k^2 \cdot \text{ddim}(\mathcal{X}) \cdot \varepsilon^{-2})$.

If $M = (\mathcal{X}, \text{dist})$ is a shortest-path metric of a graph with bounded treewidth $t$, we have $d_{VC}(\mathcal{X}) = O(t)$ by [8]. Thus, we have $|S_O| = \tilde{O}(k \cdot t \cdot \varepsilon^{-2})$ by Lemma E.8 and $|S_I| = \tilde{O}(k^2 \cdot t \cdot \varepsilon^{-4})$ by Theorem E.11, leading to a coreset of size $\tilde{O}(k^2 \cdot t \cdot \varepsilon^{-4})$.

If $M = (\mathcal{X}, \text{dist})$ is a shortest-path metric of a graph that excludes a fixed minor $H$, we have $d_{VC}(\mathcal{X}) = O(|H|)$ by [8]. Thus, we have $|S_O| = \tilde{O}(k \cdot |H| \cdot \varepsilon^{-2})$ by Lemma E.8 and $|S_I| = \tilde{O}(k^2 \cdot |H| \cdot \varepsilon^{-4})$ by Theorem E.11, leading to a coreset of size $\tilde{O}(k^2 \cdot |H| \cdot \varepsilon^{-4})$.

$\square$

## G  Additional empirical results

### G.1  Additional empirical results for robust geometric median

We first present Table 4, which lists the parameters of the datasets used in our experiments.

We show the missing result of Section 4 on **Bank**, **Athlete** and **Diabetes** dataset in Figure 4. This figure demonstrates that our coreset construction algorithm consistently outperforms the baselines. For instance, on the **Diabetes** dataset, our method produces a coreset of size 180 with an empirical error of 0.057, while the best baseline, **HLLW25**, results in a coreset size of 280 with an empirical error of 0.065.

**Analysis of our algorithm when $n < 4m$.** For robust geometric median, we set $m = n/2$ in the six dataset, violating the assumption $n \geq 4m$, and report the corresponding size-error tradeoff in Figure 5. The results show that our method still consistently outperforms the baselines even when $n < 4m$. For instance, in **Census1990** dataset, our method produces a coreset of size 50200 with an empirical error of 0.004, while the best baseline produces a coreset of size 51200 with a much larger empirical error of 0.012.

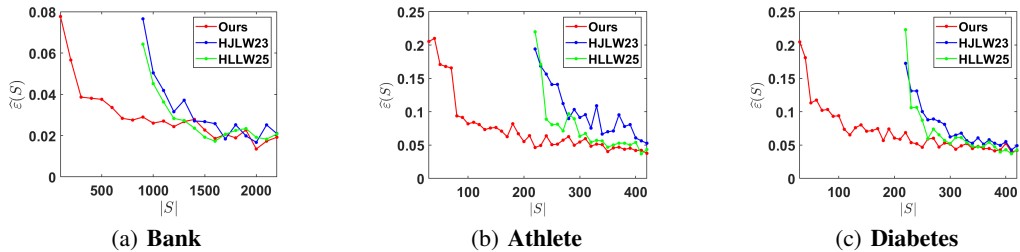

(a) **Bank**  (b) **Athlete**  (c) **Diabetes**

Figure 4: Tradeoff between coreset size $|S|$ and empirical error $\widehat{\varepsilon}(S)$. The horizontal axis is $|S|$ and the vertical axis is $\widehat{\varepsilon}(S)$.

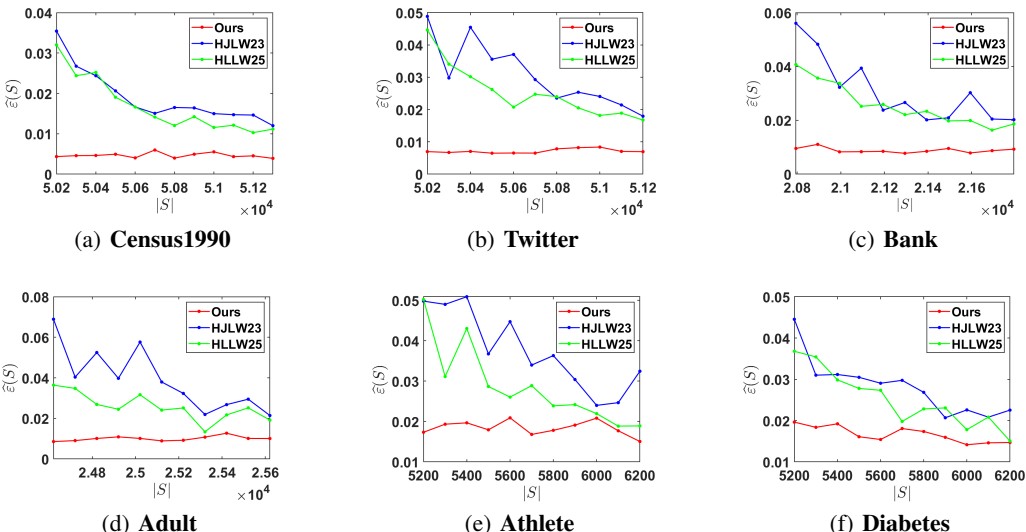

(a) **Census1990**  (b) **Twitter**  (c) **Bank**

(d) **Adult**  (e) **Athlete**  (f) **Diabetes**

Figure 5: Tradoff between coreset size $|S|$ and empirical error $\widehat{\varepsilon}(S)$ for robust geometric median when we set $m = n/2$. In this scenario, the assumption $n \geq 4m$ is violated.

**Analysis under heavy-tailed contamination.** For each dataset, we randomly perturb 10% of the points by adding independent $\mathrm{Cauchy}(0, 1)$ [4] noise to every dimension, thereby simulating a heavy-tailed data environment. As shown in Figure 6, our method consistently outperforms the baselines on the robust geometric median task, demonstrating that its performance advantage remains stable even under heavy-tailed contamination. For example, on the perturbed **Twitter** dataset, our method achieves a coreset of size $1500$ with an empirical error of $0.031$, whereas the best baseline, **HLLW25**, requires a coreset twice as large ($3000$) to attain a higher error of $0.032$.

### G.2 Empirical results for robust 1D geometric median

We implement our 1D coreset construction algorithm and compare its performance to the previous baselines and our general dimension method. All experiments are conducted on a PC with an Intel Core i9 CPU and 16GB of memory, and the algorithms are implemented in C++ 11.

**Setup.** We select the first dimension of the **Twitter** dataset, the second dimension of the **Adult** dataset, and the second dimension of the **Bank** dataset to create the **Twitter1D**, **Adult1D**, and **Bank1D** datasets. We choose these dimensions because other dimensions in the four datasets listed in Table 4 contain no more than 130 distinct points, which would result in an overly small coreset. We conduct experiments on these 1D datasets. To compute an approximate center for each dataset, we use the $k$-means++ algorithm.

---

[4]The probability density function of $\mathrm{Cauchy}(0, 1)$ is $f(x) = \frac{1}{\pi(1+x^2)}$ for any $x \in \mathbb{R}$.

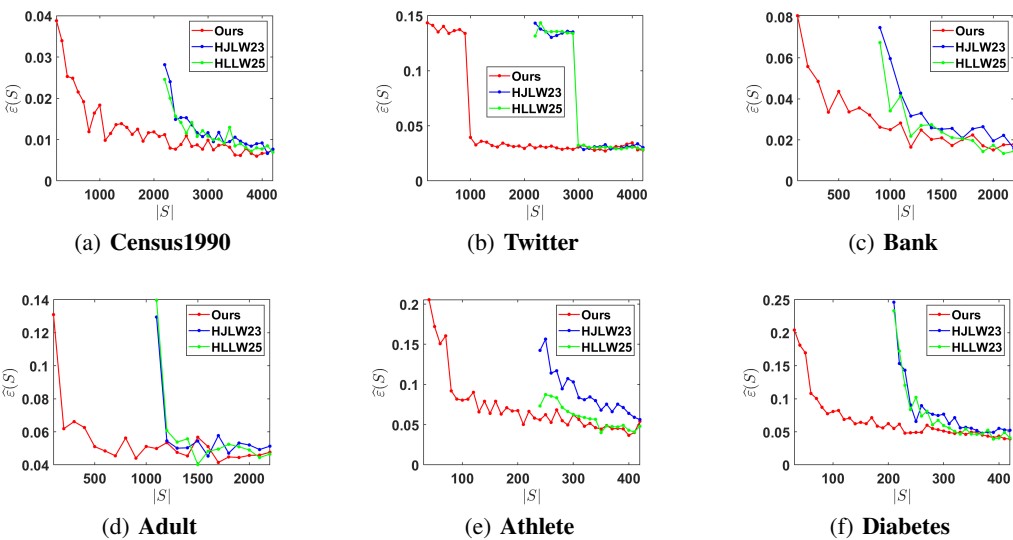

Figure 6: Tradoff between coreset size $|S|$ and empirical error $\widehat{\varepsilon}(S)$ for robust geometric median when we perturb 10% points.

**Experiment result.** We vary the coreset size from 200 to $m + 1000$, and compute the empirical error $\widehat{\varepsilon}(S)$ w.r.t. the coreset size $|S|$. For each size and each algorithm, we independently run the algorithm 10 times and obtain 10 coresets, compute their empirical errors $\widehat{\varepsilon}(S)$, and report the average of 10 empirical errors. Figure 7 presents our results. This figure shows that our 1D coreset (denoted by **Our1D**) outperforms the previous baselines and our coreset for general dimension. For example, with the **Twitter1D** dataset, our 1D method provides a coreset of size 320 with an empirical error 0.013. The best empirical error achieved by our general dimension method for a coreset size 3500 is much larger, 0.087. Note that the coreset size of **Our1D** method does not grow uniformly, since the number of buckets does not increase linearly with $\varepsilon^{-1}$ or $\varepsilon^{-1/2}$.

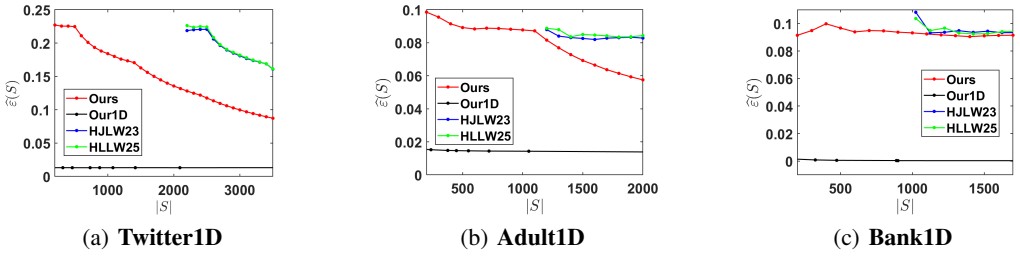

Figure 7: Tradeoff between coreset size $|S|$ and empirical error $\widehat{\varepsilon}(S)$ in 1D datasets.

### G.3 Empirical results for robust $k$-median

We implement Algorithm 3 for robust $k$-median and compare its performance to the previous baselines.

**Setup.** We do experiments on the six datasets listed in Table 4, and set the number of centers $k$ to be 5. We use Lloyd version $k$-means++ to compute an approximate center $C^\star$.

**Coreset size and empirical error tradeoff for robust $k$-median.** We vary the coreset size from $m$ to $2m$, and compute the empirical error $\widehat{\varepsilon}(S)$ w.r.t. the coreset size $|S|$. For each size and each algorithm, we independently run the algorithm 10 times and obtain 10 coresets, compute their empirical errors $\widehat{\varepsilon}(S)$, and report the average of 10 empirical errors. Figure 8 presents our results.

This figure shows that our algorithm (**Ours**) outperforms the previous baselines. For example, with the **Census1990** dataset, our method provides a coreset of size 2200 with empirical error 0.012. The best empirical error achieved by baselines for a coreset size 3600 is larger, 0.013.

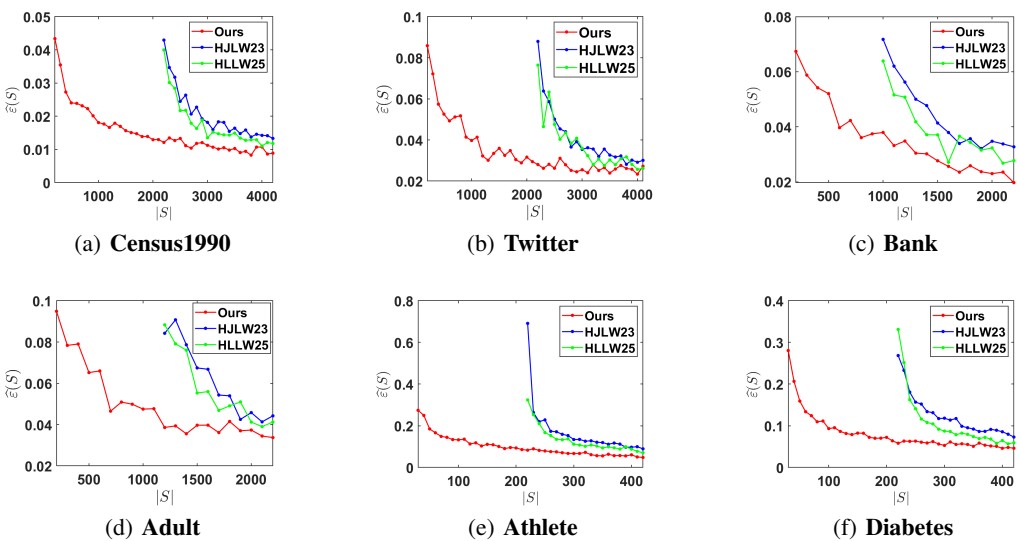

|   |   |   |
|---|---|---|
| (a) **Census1990** | (b) **Twitter** | (c) **Bank** |
| (d) **Adult** | (e) **Athlete** | (f) **Diabetes** |

Figure 8: Tradeoff between coreset size $|S|$ and empirical error $\widehat{\varepsilon}(S)$ for robust $k$-median on real-world datasets.

**Statistical test.**   Similar to the robust geometric median, we evaluate the statistical performance between our method and baselines for robust $k$-median on all six real-world datasets. The results, listed in Table 5, further demonstrate that our algorithm consistently outperforms the baselines.

Table 5: Statistical comparison of different coreset construction methods for robust $k$-median. The coreset $S_1$ represents our coreset, $S_2$ represents the coreset constructed by the baseline **HJLW23**, and $S_3$ the coreset constructed by baseline **HLLW25**. For each empirical error ratio $\widehat{\varepsilon}(S_2)/\widehat{\varepsilon}(S_1)$ and $\widehat{\varepsilon}(S_3)/\widehat{\varepsilon}(S_1)$, we report the mean value over 20 runs, with the subscript indicating the standard deviation.

| Coreset Size | Census1990 | | Twitter | |
|---|---|---|---|---|
| | $\widehat{\varepsilon}(S_2)/\widehat{\varepsilon}(S_1)$ | $\widehat{\varepsilon}(S_3)/\widehat{\varepsilon}(S_1)$ | $\widehat{\varepsilon}(S_2)/\widehat{\varepsilon}(S_1)$ | $\widehat{\varepsilon}(S_3)/\widehat{\varepsilon}(S_1)$ |
| 2200 | $3.374_{0.818}$ | $3.800_{1.262}$ | $2.958_{0.785}$ | $3.149_{1.477}$ |
| 3200 | $1.800_{0.659}$ | $1.483_{0.534}$ | $1.654_{0.788}$ | $1.574_{0.710}$ |
| 4200 | $1.543_{0.559}$ | $1.316_{0.373}$ | $1.408_{0.439}$ | $1.379_{0.369}$ |

| Coreset Size | Bank | | Adult | |
|---|---|---|---|---|
| | $\widehat{\varepsilon}(S_2)/\widehat{\varepsilon}(S_1)$ | $\widehat{\varepsilon}(S_3)/\widehat{\varepsilon}(S_1)$ | $\widehat{\varepsilon}(S_2)/\widehat{\varepsilon}(S_1)$ | $\widehat{\varepsilon}(S_3)/\widehat{\varepsilon}(S_1)$ |
| 1200 | $1.657_{0.415}$ | $2.019_{0.740}$ | $2.098_{0.551}$ | $2.153_{0.772}$ |
| 1700 | $1.548_{0.582}$ | $1.257_{0.543}$ | $1.440_{0.619}$ | $1.702_{0.584}$ |
| 2200 | $1.393_{0.558}$ | $1.460_{0.737}$ | $1.450_{0.677}$ | $1.373_{0.545}$ |

| Coreset Size | Athlete | | Diabetes | |
|---|---|---|---|---|
| | $\widehat{\varepsilon}(S_2)/\widehat{\varepsilon}(S_1)$ | $\widehat{\varepsilon}(S_3)/\widehat{\varepsilon}(S_1)$ | $\widehat{\varepsilon}(S_2)/\widehat{\varepsilon}(S_1)$ | $\widehat{\varepsilon}(S_3)/\widehat{\varepsilon}(S_1)$ |
| 210 | $11.481_{3.633}$ | $8.851_{0.740}$ | $12.688_{3.443}$ | $8.080_{1.652}$ |
| 310 | $2.142_{0.525}$ | $1.798_{0.553}$ | $1.902_{0.512}$ | $1.402_{0.395}$ |
| 410 | $2.104_{0.570}$ | $1.523_{0.481}$ | $1.691_{0.546}$ | $1.257_{0.470}$ |

**Speed-up baselines.**   We compare the coreset of size $2m$ constructed by the **HLLW25** baselines and coreset of size $m$ conducted by Algorithm 3. We repeat the experiment 10 times and report the aver-

ages. The result is listed in Table 6. Our algorithm achieves a speed-up over **HLLW25**—specifically, a $2\times$ reduction in the running time on the coreset—while maintaining the same level of empirical error.

**Validity of Assumption 1.4 for robust $k$-median.** We evaluate the validity of Assumption 1.4 for robust $k$-median across six datasets. The results in Table 4 show that the assumptions are satisfied by the **Twitter**, **Adult**, **Athlete**, and **Diabetes** datasets, while the condition $r_{\max} \le 4k\bar{r}$ holds for all six datasets. These results demonstrate that our assumptions are practical in real-world scenarios, and our algorithm performs well even when the assumption $\min_i |P_i^\star| \ge 4m$ is violated. Note that the condition $r_{\max} \le 4k\bar{r}$ is satisfied by all six real-world datasets, even across different choices of $k$ and $m$.

**Analysis under heavy-tailed contamination.** For each dataset, we randomly perturb 10% of the points by adding independent $\mathrm{Cauchy}(0,1)$ noise to every dimension, thereby simulating a heavy-tailed data environment. As shown in Figure 9, our method consistently outperforms the baselines on the robust $k$-median task, confirming that its superior performance remains stable even under heavy-tailed contamination. For example, on the perturbed **Bank** dataset, our method achieves a coreset of size 600 with an empirical error of 0.044, whereas the best baseline, **HLLW25**, requires a coreset of size 1300 to reach a higher error of 0.046.

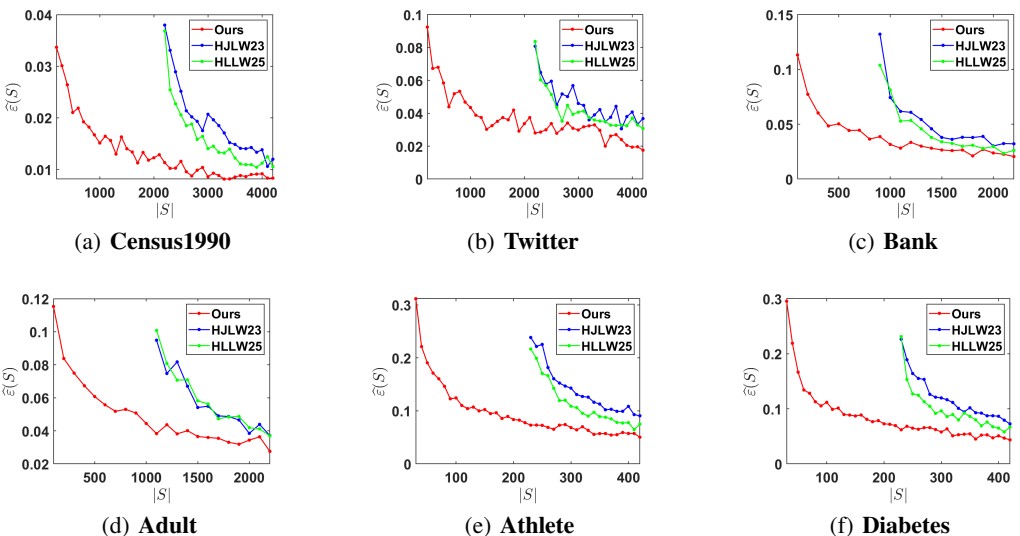

Figure 9: Tradoff between coreset size $|S|$ and empirical error $\widehat{\varepsilon}(S)$ for robust $k$-median when we perturb 10% points.

### G.4 Empirical results for robust $k$-means

We implement Algorithm 3 for robust $k$-means and compare its performance to the previous baselines.

**Setup.** We do experiments on the six datasets listed in Table 4, and set the number of centers $k$ to be 5. We use $k$-means++ to compute an approximate center $C^\star$.

**Coreset size and empirical error tradeoff for Robust $k$-means.** We vary the coreset size from $m$ to $2m$, and compute the empirical error $\widehat{\varepsilon}(S)$ w.r.t. the coreset size $|S|$. For each size and each algorithm, we independently run the algorithm 10 times and obtain 10 coresets, compute their empirical errors $\widehat{\varepsilon}(S)$, and report the average of 10 empirical errors. Figure 10 presents our results. This figure shows that our algorithm (**Ours**) outperforms the previous baselines. For example, with the **Twitter** dataset, our method provides a coreset of size 2200 with an empirical error 0.039. The best empirical error achieved by our general dimension method for a coreset size 4000 is much larger, 0.056.

Table 6: Comparison of runtime between our Algorithm 3 and baseline **HLLW25** for robust $k$-median. For each dataset, the coreset size of baseline **HLLW25** is $2m$ and the coreset size of ours is $m$. We use Lloyd algorithm given by [7] to compute approximate solutions $C_P$ and $C_S$ for both the original dataset $P$ and coreset $S$, respectively. "COST$_P$" denotes $\text{cost}_1^{(m)}(P, C_P)$ on the original dataset $P$. "COST$_S$" denotes $\text{cost}_1^{(m)}(P, C_S)$ on the coreset constructed by METHOD. $T_X$ is the running time on the original dataset. $T_S$ is the running time on coreset. $T_C$ is the construction time of the coreset.

| DATASET | COST$_P$ | METHOD | COST$_S$ | $T_X$ | $T_C$ | $T_S$ |
|---|---|---|---|---|---|---|
| CENSUS1990 | $1.032 \times 10^6$ | OURS | $1.030 \times 10^6$ | 312.606 | 38.862 | 5.532 |
| | | HLLW25 | $1.040 \times 10^6$ | | 38.703 | 10.815 |
| TWITTER | $1.328 \times 10^6$ | OURS | $1.364 \times 10^6$ | 96.024 | 12.220 | 1.621 |
| | | HLLW25 | $1.347 \times 10^6$ | | 12.452 | 2.995 |
| BANK | $3.179 \times 10^6$ | OURS | $3.194 \times 10^6$ | 147.324 | 9.021 | 1.484 |
| | | HLLW25 | $3.207 \times 10^6$ | | 9.210 | 3.445 |
| ADULT | $9.221 \times 10^8$ | OURS | $9.153 \times 10^8$ | 144.185 | 9.324 | 1.854 |
| | | HLLW25 | $9.166 \times 10^8$ | | 9.266 | 3.517 |
| ATHLETE | $7.251 \times 10^4$ | OURS | $7.503 \times 10^4$ | 41.541 | 1.825 | 0.206 |
| | | HLLW25 | $7.389 \times 10^4$ | | 1.872 | 0.421 |
| DIABETES | $8.571 \times 10^4$ | OURS | $8.791 \times 10^4$ | 44.733 | 1.829 | 0.226 |
| | | HLLW25 | $8.811 \times 10^4$ | | 1.850 | 0.473 |

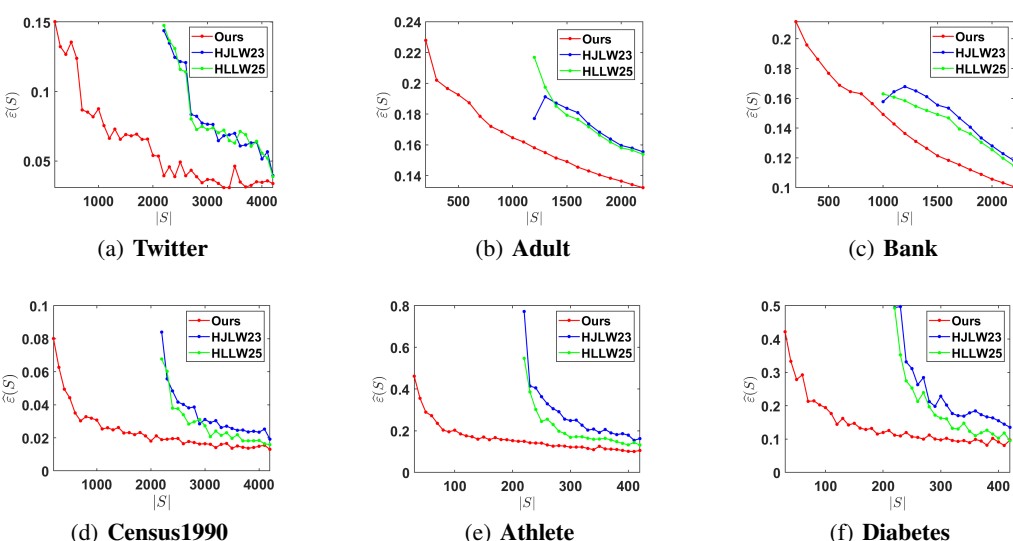

Figure 10: Tradeoff between coreset size $|S|$ and empirical error $\widehat{\varepsilon}(S)$ for robust $k$-means.

**Statistical test.** Similar to the previous cases, we evaluate the statistical performance between our method and baselines for robust $k$-means on all six real-world datasets. The results, listed in Table 7, further demonstrate that our algorithm consistently outperforms the baselines.

**Speed-up baselines.** We compare the coreset of size $2m$ constructed by the **HLLW25** baselines and coreset of size $m$ conducted by Algorithm 3. We repeat the experiment 10 times and report the averages. The result is listed in Table 8. Our algorithm achieves a speed-up over **HLLW25**—specifically, a $2\times$ reduction in the running time on the coreset—while maintaining the same level of empirical error.

**Validity of Assumption 1.4 for robust $k$-means.** We evaluate the validity of Assumption 1.4 for robust $k$-means across six datasets. The results in Table 4 show that our assumptions are satisfied by datasets **Athlete** and **Diabetes**, while the assumption $r_{\max}^2 \leq 4k\bar{r}^2$ is satisfied by all six datasets.

Table 7: Statistical comparison of different coreset construction methods for robust geometric median. The coreset $S_1$ represents our coreset, $S_2$ represents the coreset constructed by the baseline **HJLW23**, and $S_3$ the coreset constructed by baseline **HLLW25**. For each empirical error ratio $\widehat{\varepsilon}(S_2)/\widehat{\varepsilon}(S_1)$ and $\widehat{\varepsilon}(S_3)/\widehat{\varepsilon}(S_1)$, we report the mean value over 20 runs, with the subscript indicating the standard deviation.

| Coreset Size | Census1990 | | Twitter | |
| | $\widehat{\varepsilon}(S_2)/\widehat{\varepsilon}(S_1)$ | $\widehat{\varepsilon}(S_3)/\widehat{\varepsilon}(S_1)$ | $\widehat{\varepsilon}(S_2)/\widehat{\varepsilon}(S_1)$ | $\widehat{\varepsilon}(S_3)/\widehat{\varepsilon}(S_1)$ |
|---|---|---|---|---|
| 2200 | $3.253_{2.063}$ | $2.645_{1.458}$ | $1.793_{0.644}$ | $1.667_{0.479}$ |
| 3200 | $1.257_{0.842}$ | $1.251_{0.632}$ | $1.343_{0.234}$ | $1.283_{0.197}$ |
| 4200 | $1.303_{0.692}$ | $1.168_{0.739}$ | $1.244_{0.152}$ | $1.246_{0.148}$ |

| Coreset Size | Bank | | Adult | |
| | $\widehat{\varepsilon}(S_2)/\widehat{\varepsilon}(S_1)$ | $\widehat{\varepsilon}(S_3)/\widehat{\varepsilon}(S_1)$ | $\widehat{\varepsilon}(S_2)/\widehat{\varepsilon}(S_1)$ | $\widehat{\varepsilon}(S_3)/\widehat{\varepsilon}(S_1)$ |
|---|---|---|---|---|
| 1200 | $1.647_{0.972}$ | $1.360_{1.018}$ | $1.467_{0.287}$ | $1.094_{0.542}$ |
| 1700 | $1.010_{0.654}$ | $1.028_{0.574}$ | $2.149_{0.884}$ | $2.416_{1.002}$ |
| 2200 | $1.010_{0.654}$ | $1.026_{0.674}$ | $1.089_{0.360}$ | $1.172_{0.537}$ |

| Coreset Size | Athlete | | Diabetes | |
| | $\widehat{\varepsilon}(S_2)/\widehat{\varepsilon}(S_1)$ | $\widehat{\varepsilon}(S_3)/\widehat{\varepsilon}(S_1)$ | $\widehat{\varepsilon}(S_2)/\widehat{\varepsilon}(S_1)$ | $\widehat{\varepsilon}(S_3)/\widehat{\varepsilon}(S_1)$ |
|---|---|---|---|---|
| 210 | $5.172_{3.634}$ | $4.200_{1.944}$ | $5.700_{3.303}$ | $5.868_{2.952}$ |
| 310 | $2.467_{1.564}$ | $1.427_{0.660}$ | $1.567_{0.800}$ | $1.332_{0.653}$ |
| 410 | $1.658_{0.881}$ | $1.045_{0.449}$ | $1.360_{1.103}$ | $1.216_{0.943}$ |

Table 8: Comparison of runtime between our Algorithm 3 and baseline **HLLW25** for robust $k$-means. For each dataset, the coreset size of baseline **HLLW25** is $2m$ and the coreset size of ours is $m$. We use Lloyd algorithm given by [7] to compute approximate solutions $C_P$ and $C_S$ for both the original dataset $P$ and coreset $S$, respectively. "COST$_P$" denotes $\text{cost}_2^{(m)}(P, C_P)$ on the original dataset $P$. "COST$_S$" denotes $\text{cost}_2^{(m)}(P, C_S)$ on the coreset constructed by METHOD. $T_X$ is the running time on the original dataset. $T_S$ is the running time on coreset. $T_C$ is the construction time of the coreset.

| DATASET | COST$_P$ | METHOD | COST$_S$ | $T_X$ | $T_C$ | $T_S$ |
|---|---|---|---|---|---|---|
| CENSUS1990 | $1.172\times10^7$ | OURS | $1.170\times10^7$ | 358.218 | 35.513 | 5.770 |
| | | HLLW25 | $1.170\times10^7$ | | 35.399 | 11.043 |
| TWITTER | $2.657\times10^7$ | OURS | $2.664\times10^7$ | 106.327 | 14.084 | 1.806 |
| | | HLLW25 | $2.662\times10^7$ | | 13.330 | 3.494 |
| BANK | $3.477\times10^8$ | OURS | $3.531\times10^8$ | 133.978 | 7.478 | 1.283 |
| | | HLLW25 | $3.530\times10^8$ | | 7.225 | 2.725 |
| ADULT | $2.575\times10^{13}$ | OURS | $2.646\times10^{13}$ | 139.16 | 8.273 | 1.564 |
| | | HLLW25 | $2.652\times10^{13}$ | | 8.048 | 3.091 |
| ATHLETE | $8.630\times10^5$ | OURS | $9.089\times10^5$ | 44.208 | 1.890 | 0.251 |
| | | HLLW25 | $9.016\times10^5$ | | 1.909 | 0.466 |
| DIABETES | $6.490\times10^5$ | OURS | $6.814\times10^5$ | 39.570 | 1.622 | 0.196 |
| | | HLLW25 | $6.838\times10^5$ | | 1.602 | 0.413 |

These results show that our assumptions are practical in real-world datasets, and our algorithm performs well even when the assumption $\min_i |P_i^\star| \geq 4m$ is violated.

**Analysis of our algorithm when Assumption 1.4 violates.** We evaluate the applicability of our algorithm when both assumptions $\min_i |P_i^\star| \geq 4m$ and $(r_{\max}/\bar{r})^2 \leq 4k$ are violated. In **Bank** dataset, we let $k = 3$, then we have $(r_{\max}/\bar{r})^2 = 15.001 > 4k$ and $\min_i |P_i^\star| = 1432 < 4m$, violating the two assumptions. In **Adult** dataset, we let $k = 8$, $m = 500$, then we have $(r_{\max}/\bar{r})^2 = 36.278 > 4k$ and $\min_i |P_i^\star| = 1592 < 4m$, violating the two assumptions. We present the size-error tradeoff for robust $k$-means on these datasets in Figure 11. Our results show that our method still outperforms the baselines even when both assumptions are violated.

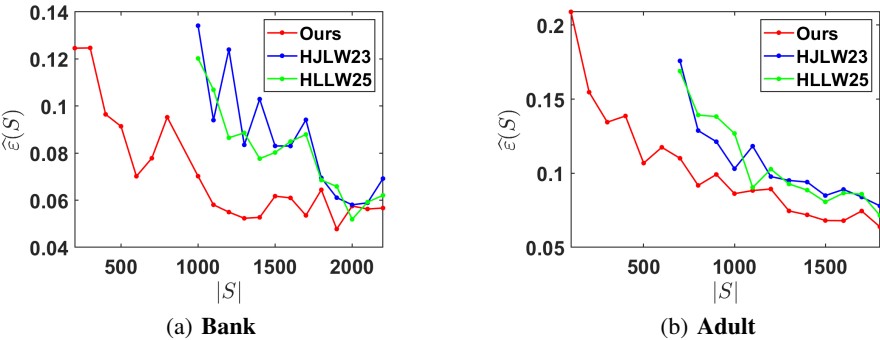

(a) **Bank**  (b) **Adult**

Figure 11: Tradeoff between coreset size $|S|$ and empirical error $\widehat{\varepsilon}(S)$ for robust $k$-means. We set $k = 3$ in the **Bank** dataset and set $k = 8$, $m = 500$ in the **Adult** dataset. In these cases, the values $(r_{\max}/\bar{r})^2$ and $\min_{i \in [k]} |P_i^\star|$ violate both of our assumptions. The goal of this figure is to examine whether our algorithm remains applicable when the assumptions are not satisfied.

**Analysis under heavy-tailed contamination.** For each dataset, we randomly perturb 10% of the points by adding independent $\mathrm{Cauchy}(0, 1)$ noise to every dimension, thereby simulating a heavy-tailed data environment. As shown in Figure 12, our method consistently outperforms the baselines on the robust $k$-means task. For example, on the perturbed **Athlete** dataset, our method yields a coreset of size 110 with an empirical error of 0.181, whereas the best baseline, **HLLW25**, attains a higher error of 0.196 even with a larger coreset of size 290.

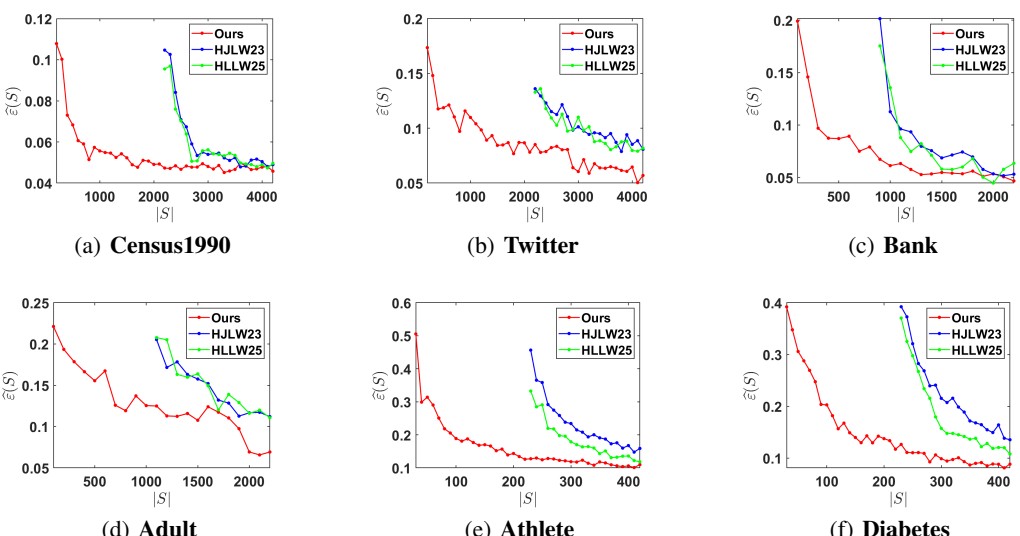

(a) **Census1990**  (b) **Twitter**  (c) **Bank**

(d) **Adult**  (e) **Athlete**  (f) **Diabetes**

Figure 12: Tradoff between coreset size $|S|$ and empirical error $\widehat{\varepsilon}(S)$ for robust $k$-means when we perturb 10% points.

