# OpenReview forum: "Coreset for Robust Geometric Median: Eliminating Size Dependency on Outliers"
_NeurIPS.cc/2025/Conference — NeurIPS 2025 poster_

### Official Review · Reviewer_sFeV · 2025-06-24

**Clarity:** 3
**Significance:** 3
**Originality:** 3
**Rating:** 5
**Confidence:** 2

**Summary:**

The paper proposes improved coreset constructions for the robust geometric median problem that remove the dependency on the number of outliers. It also extends to robust $(k, z)$-clustering and shows strong empirical performance.

**Questions:**

1. **Presentation Alignment:** Several figures, tables, and algorithm blocks are not placed near the corresponding explanatory text. Reorganizing the layout so that each element appears closer to the relevant discussion would significantly improve readability and reader comprehension.

2. **Structure of Experimental Results:** Key experimental results are currently not sufficiently highlighted in the main text. Could the authors restructure the paper to better emphasize these results in the main body, rather than relegating them to the appendix or supplementary material?

**Ethical Concerns:**

["NO or VERY MINOR ethics concerns only"]

**Final Justification:**

The authors have adequately addressed my concerns. I will maintain my current recommendation.

**Limitations:**

yes

**Paper Formatting Concerns:**

Only minor formatting issues (e.g., placement of figures/tables/algorithms) were observed and have already been noted in the Questions section above.

**Quality:**

4

**Strengths And Weaknesses:**

**Quality:**
The paper appears technically sound, with well-defined theoretical contributions and empirical validation across multiple datasets. The improvements over prior work, particularly in terms of coreset size and runtime efficiency, are clearly articulated and seem well-supported by both analysis and experiments.

**Clarity:**
The manuscript is generally clear and well-organized. However, there are some presentation issues—specifically, the placement of figures, tables, and algorithm blocks could be more closely aligned with the corresponding narrative. Additionally, the structure of the paper could be improved to highlight key experimental results more prominently in the main body, rather than deferring them to the appendix or supplementary materials.

**Significance:**
The problem of robust coreset construction for geometric median and $(k, z)$-clustering is important for scalable learning. The elimination of outlier dependency and the extension to general metric spaces indicate potential for meaningful impact in both theory and practice.

**Originality:**
The work introduces a novel non-component-wise error analysis technique and achieves theoretical and empirical improvements over existing methods. These contributions appear to be a non-trivial and original step forward in the field.

---

> ### Author Rebuttal · Authors · 2025-07-31
>
> Thank you for your positive assessment of our work’s quality, clarity, significance, and originality. In response to your feedback, we will improve the paper’s organization for better readability and move key experimental results from the appendix into the main text to enhance accessibility.
>
> > Presentation Alignment: Several figures, tables, and algorithm blocks are not placed near the corresponding explanatory text. Reorganizing the layout so that each element appears closer to the relevant discussion would significantly improve readability and reader comprehension.
>
> Thanks for your comment. We will do this to improve readability in the final version.
>
> > Structure of Experimental Results: Key experimental results are currently not sufficiently highlighted in the main text. Could the authors restructure the paper to better emphasize these results in the main body, rather than relegating them to the appendix or supplementary material?
>
> Thank you for your question. Yes, we will move the experimental results on the robust geometric median to the main text, including:
>
> * **Statistical test (Table 4):** Results show that our coreset consistently achieves lower empirical error than the baselines across six datasets, demonstrating the consistency of our method’s outperformance.
> * **Analysis for $n < 4m$ (Figure 5):** Results indicate that our coreset remains effective even when the assumption $n \geq 4m$ is not satisfied, highlighting the robustness of our method.
> * **Size-error tradeoff for 1D case (Figure 6):** Results show that our 1D method consistently outperforms the baselines for the robust 1D geometric median.
>
> Incorporating these results into the main text will enhance the integrity of the experiments presented in the main text.
>
> Thank you again for your comments. To further improve the paper’s presentation, we will also move Assumption F.6 to the main text and add figures to illustrate the pseudo-code.

---

> > ### Comment · Reviewer_sFeV · 2025-08-05
> >
> > The authors have adequately addressed my concerns. I will maintain my current recommendation.

---

### Official Review · Reviewer_NhxZ · 2025-07-02

**Clarity:** 3
**Significance:** 3
**Originality:** 3
**Rating:** 5
**Confidence:** 4

**Summary:**

This paper studies the robust geometric median problem in the d=1 and general d case, and provides improved results for coresets in these settings.  For d=1 their algorithm polynomially reduces the dependence on the error term eps, removes the dependence on the number of outliers m, and is tight up to poly-logarithmic factors.  For the general d case, they use the insight from the d=1 case to make a simple and clever adaptation of previous results, and this also improves the previous results by removing the dependence on the number of outliers m.
These results extend to provide improved results for robust (k,z)-clustering.
Experiments are provided to show on real data the improvement in size/error trade-off on previous robust-median coresets algorithms.

**Questions:**

N/A

**Ethical Concerns:**

["NO or VERY MINOR ethics concerns only"]

**Final Justification:**

Clear and nice theoretical results on a problem central to NeurIPS (clustering for coresets).  The new methods are simple -- this is a good thing.  Shows real improvement in simple experiments, so seems a real improvment, not just theoretical.

**Limitations:**

yes

**Quality:**

4

**Strengths And Weaknesses:**

I found the new methods and analysis relatively simple, but it improves the results which appeared in STOC, SODA, ICALP, FOCS in the last few years -- in that it removes the dependence on the number of outliers.  I appreciate the simple analysis, it is a positive.  And removing the dependence on the number of outliers is an important improvement.

The topic of coresets, and clustering, and robust estimators are all very relevant for NeurIPS, so since this is an important improvement, I think the paper should be accepted.

I found the writing to be formal, and no errors or concerned, but a bit dense.  I appreciate the Figure 1, but I did not find it very well labeled.

I was glad to see the experiments, even though it is a theory paper.  They convinced me that the improvements were meaningful.

---

> ### Author Rebuttal · Authors · 2025-07-31
>
> Thank you for your positive feedback and for recognizing both the significance of our contributions and the strength of our experimental results. We also appreciate your comment regarding Figure 1's labeling; in the final version we will enhance the axis titles, legend descriptions, and in-figure annotations to ensure it is fully self-explanatory.

---

> > ### Comment · Reviewer_NhxZ · 2025-08-03
> >
> > I believe the authors have a plan to address all concerns among reviews.  I think doing so will make the paper better.

---

### Official Review · Reviewer_n7bA · 2025-07-02

**Clarity:** 3
**Significance:** 1
**Originality:** 1
**Rating:** 4
**Confidence:** 4

**Summary:**

This paper addresses the problem of constructing coresets for the robust geometric median in Euclidean space $\mathbb{R}^d$. The key innovation is a method that removes the dependence on the number of outliers $m$ in the coreset size, which was a central limitation in prior works. The authors achieve this through a novel non-component-wise error analysis, leading to a coreset of size $\tilde{O}(\varepsilon^{-2} \cdot \min\{\varepsilon^{-2}, d\})$ when $n \geq 4m$. For $d = 1$, they obtain an optimal bound of $\Theta(\varepsilon^{-1/2} + \frac{m}{n} \varepsilon^{-1})$. The paper extends the results to robust $(k,z)$-clustering in various metric spaces. In terms of theoretical novelty, the paper introduces a novel non-component-wise error analysis, which enables a substantial reduction in the influence of outliers. The authors also provide empirical results on real-world datasets showing improved size-accuracy trade-offs and runtime.

**Questions:**

1. Does the proposed non-component-wise error analysis can be generalized to other robust learning tasks, such as robust regression or PCA, or is it fundamentally specific to clustering-like cost functions?

2. How does the proposed method perform when the assumption $n \geq 4m$ is violated in a controlled way?

3. How does the coreset behave under adversarial or heavy-tailed conditions for the datasets?

4. Is it possible to construct an $\varepsilon$-coreset for the robust geometric median (or more generally, robust $(k,z)$-clustering) in high-dimensional Euclidean space $\mathbb{R}^d$ with optimal or near-optimal size, without relying on uniform sampling of outliers or the ball range space framework—perhaps via alternative geometric or combinatorial constructions?

**Ethical Concerns:**

["NO or VERY MINOR ethics concerns only"]

**Final Justification:**

Thanks for your responses. My concerns for all the experiments have been addressed. The theoretical bottlenecks are still there and that makes the contributions somewhat limited. I am happy to raise the score to 4.

**Limitations:**

The main theoretical results rely on the assumption $n \geq 4m$, which may not hold in many practical scenarios, and the coreset bounds for $d > 1$ are not tight. The analysis depends on structural properties such as bounded VC-dimension, but the sensitivity of the method to violations of these assumptions is not studied. Computational aspects, including the overhead of constructing the coreset and its scalability to large-scale or streaming settings, are not addressed. The generalization to robust $(k, z)$-clustering is stated but not formally supported. Overall, the novelty of this paper is incremental and lacks thorough and extensive theoretical comparison, empirical validation, and contextualization within the broader literature.

**Paper Formatting Concerns:**

None.

**Quality:**

2

**Strengths And Weaknesses:**

$\mathbf{Strengths}$:
1. The paper is technically sound and presents significant theoretical advancements. The proposed non-component-wise error analysis is an elegant departure from prior component-wise methods, and the theoretical claims are backed with detailed proofs and matching lower bounds.

2. The paper is well-organized and offers a detailed exposition of the techniques. However, clarity suffers due to occasional overuse of notation and a lack of high-level intuition in the main text. The manuscript would benefit from more concrete examples or illustrations, especially to help interpret the new error analysis approach.

3. The elimination of $O(m)$ dependency is a notable contribution to the literature on robust coreset construction. However, the improvement is restricted to the regime where $n \geq 4m$. Moreover, the practical impact is limited by the modest empirical gains over strong baselines.


$\mathbf{Weakness}$:

1. The paper lacks a comparison with recent state-of-the-art literature such as Ding, H. and Wang, Z., 2020; Paul, D. et. al., 2021; Cohen-Addad, V. et. al., 2022 and related coreset literature. These should be included to better contextualize the contribution.

2. The empirical section is limited in scope. While results are presented across 6 datasets, there is insufficient analysis on the robustness of the proposed methods under adversarial or heavy-tailed distributions.

3. The paper makes some strong assumptions (e.g., $n \geq 4m$, existence of VC-dimension bounds) but does not clearly discuss their practical validity or limitations.

4. While the paper claims that the proposed methodology generalizes to broader robust clustering applications, this generalization is not rigorously substantiated—neither through theoretical guarantees nor through empirical demonstrations on such tasks.


$\mathbf{References:}$

Ding, H. and Wang, Z., 2020, November. Layered sampling for robust optimization problems. In International Conference on Machine Learning (pp. 2556-2566). PMLR.

Paul, D., Chakraborty, S., Das, S. and Xu, J., 2021. Uniform concentration bounds toward a unified framework for robust clustering. Advances in Neural Information Processing Systems, 34, pp.8307-8319.

Cohen-Addad, V., Larsen, K.G., Saulpic, D. and Schwiegelshohn, C., 2022, June. Towards optimal lower bounds for k-median and k-means coresets. In Proceedings of the 54th Annual ACM SIGACT Symposium on Theory of Computing (pp. 1038-1051).

---

> ### Author Rebuttal · Authors · 2025-07-31
>
> Thank you for your feedbacks and detailed suggestions. We appreciate your recognition of the soundness, presentation, and contribution of our work.
>
> > Q1: Does the proposed non-component-wise error analysis can be generalized to other robust learning tasks, such as robust regression or PCA, or is it fundamentally specific to clustering-like cost functions?
>
> Thank you for the insightful question. To the best of our knowledge, existing coreset methods for robust PCA and robust regression do not partition the dataset into components but take samples from the entity (e.g., [26]). This makes it challenging to directly apply our non-component-wise error analysis to robust regression or PCA.
>
>
> > Q2: How does the proposed method perform when the assumption $n\ge 4m$ is violated in a controlled way?
>
> Thank you for this valuable question. Suppose $n< 4m$.
> * Theoretically, the proposed method does not guarantee to produce a coreset. This is because when $n - m = o(n)$, the coreset size for the robust geometric median is $\Omega(m)$ (Theorem 1.1). Thus, the assumption $n\ge 4m$ is necessary (up to a constant factor) for our coreset construction.
> * In practice, our method continues to outperform the baselines when $n =2m$ (Figure 5, Section G.1), demonstrating the robustness of our method.
>
>
> > Q3: How does the coreset behave under adversarial or heavy-tailed conditions?
>
> Thank you for raising this question. We do the following experiment accordingly:
>
> On the **Census1990** dataset, we fix outlier number $m = 2000$ and randomly perturb 10% of the points by adding i.i.d. Cauchy(0,1) noise, simulating heavy-tailed contamination. Our method outperforms baselines on this perturbed dataset, demonstrating the robustness of our method’s outperformance. For example, our coreset of **size 900** achieves an empirical error of **0.018**, while the best baseline error is **0.020** when the size is **2200**.
>
> We will include this experiment in the final version.
>
> > Q4: Is it possible to construct an $\varepsilon$-coreset for the robust geometric median (or more generally, robust (k,z)-clustering) in high-dimensional Euclidean space $\mathbb{R}^d$  with optimal or near-optimal size, without relying on uniform sampling of outliers or the ball range space framework—perhaps via alternative geometric or combinatorial constructions?
>
> Thank you for raising this point. Achieving near-optimal robust coreset sizes in high-dimensional Euclidean spaces via geometric or combinatorial constructions is indeed challenging. This difficulty arises primarily due to the increased complexity in the distribution of candidate centers, which complicates geometric partitioning strategies. For example, [15] employs an $\varepsilon$-net to partition the space, but this approach incurs an exponential factor of $O(\varepsilon^{-d})$ in the coreset size (see Lines 212–215). To the best of our knowledge, the current state-of-the-art coresets for both vanilla and robust geometric median problems are derived through sampling-based methods.
>
>
>
> > concrete examples or illustrations ... new error analysis approach.
>
> Thank you for this insightful comment. We agree that a concrete example can largely improve the clarity of our new error analysis approach. Below we specify one: Let $P=P_1\cup P_2$ where $P_1=\\{p_1,\ldots,p_{n-m}\\}$ with $p_i = \frac{i}{n}$ and $P_2=\\{p_{n-m+1},\ldots,p_n\\}$ with $p_i = n^{3(i-n+m)}$ (see also Appendix A.3).
>
> * Algorithm 1 decomposes $P_2$ into at most $O(\varepsilon^{-1})$ buckets $B_i$, where each bucket contains $\varepsilon n/16$ points, and collects their mean points $\mu(B_i)$ in the coreset $S$.
>
> * Let $B=\\{p_l,\ldots,p_r\\}$ be a bucket within $P_2$. Let center $c=p_l=n^{3j}$. Since $p_l$ is an inlier point of $P$  w.r.t. $c$, the outlier number of $B$ w.r.t. $c$ must be $|B|-1$.
>
> * In the component-wise analysis, the induced error of bucket $B$ is $\left|\text{cost}^{(|B| - 1)}(B, c) - |\mu(B) - c|\right|$, which is at least $\text{dist}(\mu(B), c) = \Omega(n^{3(j+1)})$. However, $\text{cost}^{(m)}(P,c)$ is approximately $(n-m)\cdot 3^j \ll \text{dist}(\mu(B), c)$, which implies that the induced error substantially exceed the coreset error bound.
>
> * In contrast, in our non-component-wise analysis, the induced error of bucket $B$ is 0. This is because we allow the misaligned outlier numbers in each bucket. Since $\text{dist}(\mu(B),c) \gg \text{dist}(p,c)$ for any point $p \in P_1$, $\mu(B)$ will be totally regarded as an outlier w.r.t. $c$. As a result, $B$ contributes zero error.
>
> * Thus, non-component-wise analysis significantly reduces bucket-wise errors in this example, which is crucial for proving $S$ is a coreset.
>
> We will add this example to the appendix in the final version.
>
>
>
> > modest empirical gains over strong baselines.
>
> Thank you for raising this point. Below we summarize the empirical gains of our method over baselines:
>
> * **Smaller coreset size at same error**: Our method reduces the coreset size by **25%–45%** compared to baselines when achieving the same empirical error. For instance, on the **Census1990** dataset, our method achieves an empirical error of **0.012** while saving **28.6%** of the data relative to the baselines (see Lines 324–329).
>
> * **Consistent improvement across datasets**:  This size improvement consistently holds across diverse datasets across different data scales ($10^4$–$10^5$), dimensions ($2$–$64$), and diverse domains (including social network, demographic, and disease data)
>
> * **Smaller than $m$ coreset size**:
>   Unlike baselines, which require a coreset of size at least $m$, our method can achieve the same level of error with a coreset size **smaller than $m$**.
>   For example, on the **Census1990** dataset with $m = 2000$:
>   * Our method yields a coreset of **size 600** with an empirical error of **0.021**
>   * The best baseline needs **size 2300** to achieve a worse error of **0.022**
>
>   This highlights the strength of our method, especially in settings with **many outliers**.
>
> We will include this new empirical result in the final version.
>
>
> > lacks a comparison with recent state-of-the-art literature ...
>
> Thank you for suggesting these references.
> * [Ding and Wang (2020)] propose a layered sampling framework that constructs a weak coreset of size $\tilde{O}(kd\varepsilon^{-2}) + (1+\varepsilon^{-1})m$ in $O(nkd)$ time.
> * [Paul et al. (2021)] propose an iterative algorithm for robust clustering.
> * [Cohen-Addad et al. (2022)] has been discussed in Lines 46–47.
>
> In comparison, our paper focuses on (strong) coreset construction for robust clustering, which differs in objective from their works. We will cite these references in the Section 1 of the final version.
>
>
>
> > The paper makes some strong assumptions (e.g., , existence of VC-dimension bounds) but does not clearly discuss their practical validity or limitations.
>
> We apologize for any confusion.
>
> * Theorems E.11, E.12 work for any VC-dimension, just the coreset size is proportional to the VC dimension. We will remove the sentence "if the VC dimension $d_{VC}(\mathcal{X})$ is bounded" in Theorem E.11.
>
> * VC-dimension bounds are often known [9, 18, 40]:
>   * For general metric space $\mathcal{X}$, its VC-dimension is $O(\log|\mathcal{X}|)$.
>   * For shortest-path metric of a graph with bounded treewidth $t$, its VC-dimension is $O(t)$.
>   * For shortest-path metric of a graph that excludes a fixed minor $H$, its VC-dimension is $O(H)$.
>
>
> We will clarify this in the final version.
>
>
> > Computational aspects, including the overhead of constructing the coreset and its scalability to large-scale or streaming settings, are not addressed.
>
> Thank you for raising these important points.
>
> **Computational overhead.**
> The overhead for constructing our robust coresets is efficient.
>
> * **Robust 1D geometric median (Algorithm 1):** The construction takes $O(n)$ time, as detailed in Section D (Lines 1020–1026).
> * **Robust geometric median (Algorithm 2):** The overall complexity is dominated by one step that takes $O(nd)$ time, making the total overhead $O(nd)$.
> * **Robust $(k,z)$-clustering (Algorithm 3):** The total complexity is **$O(nkd)$**, with the dominant step detailed in Theorem F.7.
>
> **Scalability to large-scale data.**
> We measure scalability by the practical speed-up achieved when running downstream algorithms on the coreset versus the full dataset. Our empirical results in Table 1 show that using our robust coresets provides a 7x-10x speed-up in computation time while maintaining high accuracy, demonstrating its practical scalability.
>
> **Streaming Settings**
> Extending robust coresets to a streaming model is a valuable but challenging future direction. The primary difficulty is that our robust coreset may not satisfy the mergeability property—a key requirement for most streaming algorithms—due to the complex ways noise and outliers can interact across different data chunks.
>
> We will add this discussion on computational overhead and scalability to the main paper and explicitly list the streaming setting as an important direction for future work.
>
> > The generalization to robust (k,z)-clustering is stated but not formally supported.
>
> The formal definition and supporting proofs for robust $(k,z)$-clustering can be found in Appendix F.
>
>
> Thank you for your thoughtful comments, which have helped us improve the clarity and empirical reach of our work.

---

### Official Review · Reviewer_y1bN · 2025-07-03

**Clarity:** 2
**Significance:** 4
**Originality:** 3
**Rating:** 5
**Confidence:** 5

**Summary:**

Let P be a set of n points in R^d, m>1 and epsilon>0. Given a center c in R^d, we denote by cost^m(P,c) the sum of Euclidean distances over the points of P to c, excluding the m farthest points.

The main result is a randomized algorithm that gets P, m>1 and epsilon>0, and return a subset of size ~d/epsilon^2 that approximates cost^m(P,c) to every given c in R^d, up to a multiplicative factor of 1+epsilon. It is independent of $m$.
For d=1, the size is ~1/sqrt(eps)+m/(epsilon * n).
There are also some generalization to k-means with more assumptions,

**Questions:**

- Can you commit to an open code upon acceptance?
- What is the definition of \tilde{O}?

**Ethical Concerns:**

["NO or VERY MINOR ethics concerns only"]

**Limitations:**

yes

**Quality:**

3

**Strengths And Weaknesses:**

Strong:
- This is a very fundamental problem that would probably affect many other types of related coresets in many areas.
- Unlike previous coresets, the size is independent of m.
- The theory seems correct and interesting.
- The technique is novel, at least to me.
- Code in C++.

Weak:
- \tilde{O} is undefined, although it is used everywhere, including the main theorem.
- Assumption F6 is critical but appears only in the appendix. It is not clear, since most of the terms there (i,r,Pi) are undefined.
- A single center seems easy and less interesting then k-median/mean.
- Paper [1] is not cited, although it is one of the few papers that handle this problem.
- It was very hard to understand the pseudo-code. I Some figures would have helped.
- Put points on the curves, when you actually made the experiment.
- Datasets: Does it make sense to run your algorithm on these datasets? For what purposes?

[1] article{DBLP:journals/algorithms/StatmanRF20,
  author       = {Adiel Statman and
                  Liat Rozenberg and
                  Dan Feldman},
  title        = {k-Means: Outliers-Resistant Clustering+++},
  journal      = {Algorithms},
  volume       = {13},
  number       = {12},
  pages        = {311},
  year         = {2020},
  url          = {https://doi.org/10.3390/a13120311},
  doi          = {10.3390/A13120311},
  timestamp    = {Sat, 09 Jan 2021 14:14:06 +0100},
  biburl       = {https://dblp.org/rec/journals/algorithms/StatmanRF20.bib},
  bibsource    = {dblp computer science bibliography, https://dblp.org}
}

---

> ### Author Rebuttal · Authors · 2025-07-31
>
> Thank you for your positive feedback and for acknowledging the significance, correctness, and contributions of our work. We truly appreciate your detailed suggestions.
>
> In response to your concerns regarding paper presentation and code availability, we will (i) move Assumption F.6 to the main body and include the definition of $(i,r,P_i)$, (ii) add illustrative figures to accompany the pseudo-code, (iii) include markers on the curves for clarity, and (iv) release the code publicly upon acceptance.
>
> Below we address your specific questions.
>
>
> > What is the definition of $\tilde{O}$?
>
>
> Thank you for your question. The notation $\tilde{O}(n)$ denotes $O(n \cdot \text{polylog}(n))$. For example, our coreset size upper bound $\tilde{O}(\varepsilon^{-2}\cdot \min  \\{\varepsilon^{-2}, d \\})$ in Theorem 1.3 (Line 105) hides a polylog$(\frac{1}{\varepsilon},d)$ factor.
>
> We will add a footnote in the final version to clarify this notation.
>
>
> > Paper [1] is not cited, although it is one of the few papers that handle this problem.
>
> Thank you for pointing this out. We apologize for missing this citation.
>
> * [1] provides an $O(\log (k+m))$-approximation for robust $k$-median or $k$-means in $O(n)$ time by constructing a weak coreset of size $O(k+m)$. This is a time-efficient algorithm for robust clustering, while its approximation ratio is not constant.
>
> * [1] also proposes an $\varepsilon$-coreset for robust $k$-median or $k$-means of size $O((1/\varepsilon)^d (k+m)\log n)$. This size contains exponential dependence on $d$, much larger than our size $\tilde{O}(k^2 \varepsilon^{-2} \cdot \min\\{\varepsilon^{-2}, d\\})$.
>
> We will cite this paper in Section 1 in the final version.
>
>
> > Datasets: Does it make sense to run your algorithm on these datasets? For what purposes?
>
> Thank you for raising this question. The choice of these datasets aim to show our proposed methods consistently outperform the baselines [39, 40], across a range of scales ($10^4$–$10^5$), dimensions (2–68), and diverse domains (such as social network data, demographic data, and disease data). Moreover, these datasets cover those used in baseline [39], increasing the fairness in the comparison. Empirical results demonstrate the robustness of our methods' outperformance relative to the baselines.
>
> We will clarify the criterion for selecting these datasets in the final version.
>
>
> Thank you again for your comments, which have helped us enhance the clarity of our results and the quality of our references.

---

### Decision · Program_Chairs · 2025-09-17

**Decision:**

Accept (poster)

**Comment:**

This paper proposes ways to construct improved coresets for the robust geometric median problem, which also applies to the general problem of (k,z) clustering and to general metric spaces. The reviewers largely agree on the positive aspects of the paper, and find the empirical examples a helpful validation of a largely theoretical work. The paper focuses on a analysis that allows for the coreset size to be independent of the outlier count, and to the best of my knowledge, the analysis that is not component-based is novel and leads to tighter bounds than existing techniques. The contributions address a timely topic and reviewers agree on the merits of the work, though it will be improved with consideration for their constructive criticisms, including comments on notation, missing references, etc.